# Finite-Sample Convergence Bounds for Trust Region Policy Optimization in Mean-Field Games

**Antonio Ocello** [1]   **Daniil Tiapkin** [1 2]   **Lorenzo Mancini** [1]   **Mathieu Laurière** [3]   **Eric Moulines** [1 4]

## Abstract

We introduce Mean-Field Trust Region Policy Optimization (MF-TRPO), a novel algorithm designed to compute approximate Nash equilibria for ergodic Mean-Field Games (MFG) in finite state-action spaces. Building on the well-established performance of TRPO in the reinforcement learning (RL) setting, we extend its methodology to the MFG framework, leveraging its stability and robustness in policy optimization. Under standard assumptions in the MFG literature, we provide a rigorous analysis of MF-TRPO, establishing theoretical guarantees on its convergence. Our results cover both the exact formulation of the algorithm and its sample-based counterpart, where we derive high-probability guarantees and finite sample complexity. This work advances MFG optimization by bridging RL techniques with mean-field decision-making, offering a theoretically grounded approach to solving complex multi-agent problems.

## 1. Introduction

In an increasingly interconnected world, autonomous systems capable of adaptive decision-making have become indispensable. Their range of applications is vast and continuously expanding, spanning from autonomous driving (see, *e.g.*, Shalev-Shwartz et al., 2016) to energy market control (Samvelyan et al., 2019), and from traffic light optimization (Wiering et al., 2000) to advanced robotic systems (Matignon et al., 2007; Kober et al., 2013).

A powerful framework to model these adaptive decision agents is Multi-Agent Reinforcement Learning (MARL), which enables agents to learn optimal strategies through interaction with both the environment and other agents (see, *e.g.*, Zhang et al., 2021; Gronauer & Diepold, 2022). However, MARL faces two major challenges: *scalability* and *non-stationarity*. As the number of agents increases, the joint state-action space grows exponentially, making learning computationally prohibitive. Additionally, because all agents are simultaneously updating their policies, the environment becomes non-stationary from the perspective of each individual agent, severely hindering convergence and stability. To address these issues, many techniques have been considered, like Centralized Training with Decentralized Execution (CTDE) (Foerster et al., 2018) and attention approaches (Iqbal & Sha, 2019). While these methods improve stability and learning efficiency, it comes at a significant computational cost, limiting its scalability in large-scale systems.

Under the assumptions of *homogeneity* and *anonymity*, complex multi-agent games can be effectively approximated using Mean-Field Games (MFG). MFG provide an asymptotic approximation of exchangeable particle systems as the number of agents grows. Originally introduced by Lasry & Lions (2006a;b; 2007) and Huang et al. (2003; 2005; 2006a;b), MFG replace direct agent-to-agent interactions with a representative agent interacting with the statistical distribution of the population. Due to their analytical tractability and broad applicability, MFG have been widely adopted across various domains, including economic modeling (Bassière et al., 2024), finance (Lavigne & Tankov, 2023; Carmona et al., 2015), public health dynamics (Doncel et al., 2022), and energy storage (Alasseur et al., 2020).

In particular, Mean-Field Reinforcement Learning (MFRL) arises as the scaling limit of many MARL problems, positioning itself at the intersection of MFG and Reinforcement Learning (RL). In this setting, RL techniques are employed to solve equilibrium problems in large-scale multi-agent systems. At the core of this framework, each agent optimizes its objective while treating the mean-field distribution as fixed. This structure closely resembles CTDE in the MARL setting, but its computational cost is significantly reduced due to the mean-field approximation. In turn, the distribu-

---

[1]CMAP, CNRS, École polytechnique, Institut Polytechnique de Paris, 91120 Palaiseau, France [2]Université Paris-Saclay, CNRS, LMO, 91405, Orsay, France [3]Shanghai Frontiers Science Center of Artificial Intelligence and Deep Learning at NYU Shanghai and NYU-ECNU Institute of Mathematical Sciences, 567 West Yangsi Road, Pudong New District, Shanghai, China 200124 [4]Mohamed bin Zayed University of Artificial Intelligence, UAE. Correspondence to: Antonio Ocello .

*Proceedings of the 42$^{nd}$ International Conference on Machine Learning*, Vancouver, Canada. PMLR 267, 2025. Copyright 2025 by the author(s).

tion evolves dynamically based on the collective behavior of all agents. This formulation extends the classical notion of Nash equilibrium to the Mean-Field Nash Equilibrium (MFNE), where equilibrium emerges from the interaction between individual decision-making and population-wide updates.

MFNE provide an adequate approximation for large-scale multi-agent interactions, achieving an $\widetilde{O}(1/\sqrt{N})$-approximate Nash equilibrium in the corresponding $N$-player game (Cardaliaguet et al., 2019; Fischer & Silva, 2021; Flandoli et al., 2022). This approximation drastically reduces the complexity of analyzing strategic interactions in large populations, establishing MFG as a scalable and computationally efficient framework for real-world applications.

**Related works.** Recent work in MFG has explored the use of proximal methods due to their stability and empirical performance. Notably, Pérolat et al. (2022); Perrin et al. (2022) analyze Online Mirror Descent (OMD) from a model-specific perspective, while Yardim et al. (2023) extend this direction with a model-free approach. However, their analysis does not address population updates, operating in a restricted no-manipulation regime. These works have demonstrated the effectiveness of proximal methods in MFG, which is consistent with our research direction.

We extend this line of research by establishing finite-sample complexity guarantees providing a rigorous theoretical framework that ensures provable efficiency in solving MFG. Moreover, we explicitly incorporates the role of monotonicity in stabilizing population dynamics, a key aspect of the ergodic structure in MFG, and relax overly restrictive assumptions—such as uniformly bounded-away-from-zero policies or absolute continuity with respect to the uniform distribution. With a more flexible framework, we get refined learning guarantees under weaker assumptions, achieving an $\widetilde{O}(1/L)$ convergence rate in the optimization problem with improved sample efficiency, thus broadening the applicability of these methods. For an additional discussion on related work, we refer to Appendix A.

**Contributions.** We propose `Exact MF-TRPO` and `Sample-Based MF-TRPO`, trust-region-based algorithms for computing approximate MFNE in the MFRL setting. Our method combines the structure of ergodic MFG with the stability of trust-region optimization to enable theoretically grounded learning in multi-agent environments. Our key contributions are:

1. Theoretical analysis of `Exact MF-TRPO` with a bound on the exploitability of the learned policies, quantifying the achieved $\varepsilon_K$-MFNE after $K$ iterations.

2. A sample-based variant, `Sample-Based MF-TRPO`, with finite-sample complexity guarantees under the $\nu$-paradigm, requiring at most $\widetilde{O}(1/\varepsilon^6)$ environment interactions to reach an $\varepsilon$-MFNE.

3. Numerical experiments validating the efficiency and effectiveness of our approach in representative MFRL settings.

## 2. Framework

We use the discounted formulation as an approximation to the ergodic setting, a standard approach in the literature (see, *e.g.*, Laurière et al., 2022). This methodology allows us to build on the well-established theoretical framework of discounted RL while capturing the long-term behavior of the ergodic formulation in a stepwise fashion. In particular, we adopt the Mean-Field Markov Decision Process (MF-MDP) framework, a natural extension of the infinite-horizon discounted MDP commonly studied in RL (Sutton & Barto, 2018a). This adaptation preserves the computational tractability and convergence properties of the discounted problem while approximating the stationary dynamics of the ergodic setting. The MF-MDP framework thus serves as a bridge between classical RL and MFG models and provides a structured approach to policy optimization that is valid for both finite horizons and stationary regimes. A detailed discussion of this approach and its theoretical foundations can be found in the Appendix B.

**Notations.** For a finite set $\mathcal{X}$, let $\mathcal{P}(\mathcal{X})$ denote the set of probability distributions over $\mathcal{X}$. A finite MF-MDP is a tuple $(\mathcal{S}, \mathcal{A}, \mathsf{P}, \mathsf{r}, \gamma)$, where $\mathcal{S}$ is a finite state space, $\mathcal{A}$ is a finite action space, $\mathsf{P} : \mathcal{S} \times \mathcal{A} \times \mathcal{P}(\mathcal{S}) \to \mathcal{P}(\mathcal{S})$ is the transition function, $\mathsf{r} : \mathcal{S} \times \mathcal{A} \times \mathcal{P}(\mathcal{S}) \to \mathbb{R}$ is the reward function, $\gamma \in [0, 1)$ is the discount factor. Since we consider probability distributions over a finite state space of size $|\mathcal{S}|$, they can be identified as vectors in $\mathbb{R}^{|\mathcal{S}|}$; thus, we define the inner product $\langle \cdot, \cdot \rangle$ and the Euclidean norm $\|\cdot\|_2$ accordingly.

We assume that $\mathsf{r}$ is continuous, thus bounded, and denote by $\|\mathsf{r}\|_\infty$ its upper bound, *i.e.*, $\mathsf{r}(s, a, \mu) \in [0, \|\mathsf{r}\|_\infty]$, for $(s, a, \mu) \in \mathcal{S} \times \mathcal{A} \times \mathcal{P}(\mathcal{S})$. Given a policy $\pi : \mathcal{S} \to \mathcal{P}(\mathcal{A})$ and a population profile $\mu \in \mathcal{P}(\mathcal{S})$, we define the transition operator $\mathsf{P}_\mu^\pi$ as the transition matrix induced by the probability kernel, where actions are sampled as $a \sim \pi(s)$ under the mean-field parameter $\mu$, *i.e.*,

$$\mathsf{P}_\mu^\pi(s, s') = \sum_{a \in \mathcal{A}} \pi(a|s) \mathsf{P}(s'|s, a, \mu), \quad \text{for } s, s' \in \mathcal{S},$$

$$\mu \mathsf{P}_\mu^\pi(s) = \sum_{s_0 \in \mathcal{S}} \mu(s_0) \mathsf{P}_\mu^\pi(s_0, s), \quad \text{for } s \in \mathcal{S}.$$

$$(1)$$

Denote $\lambda_{\pi,\mu}$ the stationary distribution of the Markov chain

$P_\mu^\pi$. Let $\Pi$ to be set the policies, *i.e.*, the set of functions from $\mathcal{S}$ to $\mathcal{P}(\mathcal{A})$. For $s \in \mathcal{S}$, and $\pi, \pi' \in \Pi$, the Kullback–Leibler (KL) divergence between the two distributions $\pi(\cdot|s)$ and $\pi'(\cdot|s)$ is defined as $\mathrm{KL}(\pi(\cdot|s)\|\pi'(\cdot|s)) = \sum_{a \in \mathcal{A}} \pi(a|s) \log(\pi(a|s)/\pi'(a|s))$, if $\pi$ is absolutely continuous to $\pi'$, $\mathrm{KL}(\pi(\cdot|s)\|\pi'(\cdot|s)) = \infty$ otherwise. We use the notation $\widetilde{O}(\cdot)$ to hide polylogarithmic factors in the asymptotic complexity.

**Problem formulation.** As in Laurière et al. (2022), the discounted stationary problem within a mean-field interaction setting is designed to approximate the $N$-player game in the ergodic regime, as $\gamma \to 1$. In this setting, a representative agent in the mean-field approximation seeks to maximize the expected discounted sum of rewards while interacting with the population distribution $\mu$ under a policy $\pi$. The objective function is given by

$$
\begin{aligned}
&J^{\mathrm{MFG}}(\pi, \mu, \xi) \\
&:= \mathbb{E}\left[\sum_{t=0}^{\infty} \gamma^t \left[\mathsf{r}(s_t, a_t, \mu) + \eta \log(\pi(a_t|s_t))\right]\right],
\end{aligned} \quad (2)
$$

Given an initial state $s_0 \sim \xi$, actions are sampled at each time step as $a_t \sim \pi(\cdot \mid s_t)$, with state transitions governed by the kernel $P$, *i.e.*, $s_{t+1} \sim P(\cdot \mid s_t, a_t, \mu)$. We consider an entropy-regularized variant of the classical MFG problem, where $\eta$ denotes the entropy regularization parameter.

As MFG are an $\widetilde{O}(1/\sqrt{N})$-approximation of the $N$-player game, it is natural to introduce an additional regularization term. If this extra bias remains within the order of the existing approximation error, model fidelity is thus preserved. Regularization enhances solution stability, which is particularly beneficial given the inherent nonlinearity of MFG optimization. This is especially advantageous in RL settings, where small perturbations in the value function or policy updates can otherwise lead to erratic behavior.

The value function is defined as

$$
V^{\mathrm{MFG}}(\mu, \xi) := \max_{\pi \in \Pi} J^{\mathrm{MFG}}(\pi, \mu, \xi) . \quad (3)
$$

With a slight abuse of notation, we use $V^{\mathrm{MFG}}(\mu, s)$ (resp. $V^{\mathrm{MFG}}(\mu)$ and $J^{\mathrm{MFG}}(\pi, \mu)$) to denote $V^{\mathrm{MFG}}(\mu, \delta_s)$ for $s \in \mathcal{S}$ (resp. $V^{\mathrm{MFG}}(\mu, \mu)$ and $J^{\mathrm{MFG}}(\pi, \mu, \mu)$).

Furthermore, let $Q_\mu^\pi$ denote the regularized $Q$-function, defined as

$$
\begin{aligned}
&Q_\mu^\pi(s, a) \\
&= \mathsf{r}(s, a, \mu) + \gamma \sum_{s' \in \mathcal{S}} P(s'|s, a, \mu) \cdot J^{\mathrm{MFG}}(\pi, \mu, s') .
\end{aligned} \quad (4)
$$

We denote $\pi_\mu$ the optimal policy of optimization problem $V^{\mathrm{MFG}}(\mu)$, which is unique in the entropy-regularized RL (see, *e.g.* Haarnoja et al., 2017; Geist et al., 2019).

Define $\mathsf{d}_{\xi,\mu}^\pi$ (resp. $\mathsf{d}_{s,\mu}^\pi$ and $\mathsf{d}_\mu^\pi$) the occupation measure of the process $(s_t)_t$ induced by this policy, under the population distribution $\mu$ and initial distribution $\xi$ (resp. $\delta_s$ and $\mu$), *i.e.*,

$$
\mathsf{d}_{\xi,\mu}^\pi(s, a) = (1 - \gamma) \sum_{t=0}^{\infty} \gamma^t P((s_t, a_t) = (s, a)) , \quad (5)
$$

with $s_0 \sim \xi$, $a_t \sim \pi_t(\cdot|s_t)$, and $s_{t+1} \sim P(\cdot|s_t, a_t, \mu)$, and $\overline{\mathsf{d}}_{\mu,\xi}^\pi$ its spatial marginal, *i.e.*,

$$
\overline{\mathsf{d}}_{\mu,\xi}^\pi(s) = \sum_{a \in \mathcal{A}} \mathsf{d}_{\xi,\mu}^\pi(s, a) . \quad (6)
$$

**Interactions with the environment.** In the RL literature, various paradigms have been proposed to structure the interaction between an agent and its environment, influencing how data is collected and utilized for policy updates. In our setting, two fundamental actions can be performed: **reset** and **step**. The **reset** action initializes the environment by sampling a new state from the distribution $\nu$, effectively allowing the agent to restart from a fresh initial condition. The **step** action, on the other hand, takes a chosen action $a$ and the mean-field distribution profile $\mu$, and progresses the environment based on the current state $s$. Specifically, it samples the next state from the transition kernel $P(\cdot|s, a, \mu)$ and updates the environment accordingly. This interaction model aligns with previous works (Kakade, 2003; Shani et al., 2020) and provides an intermediate assumption in the spectrum of data access models in RL. It is therefore weaker than assuming full access to the true model or a generative model, where arbitrary state-action pairs can be queried, but stronger than the setting where no restarts are allowed, restricting exploration to trajectories induced by the current policy. This ensures reliable state exploration and promotes stable convergence.

**Nash Equilibrium.** Our work aims to compute a MFNE. A *Nash equilibrium* (NE), a fundamental concept in game theory, represents a stable state in which no player can improve her payoff by unilaterally changing her strategy, *i.e.*, each player reacts optimally to the strategies of the others.

In the MFG framework, as the number of agents approaches infinity, this concept extends to a situation in which each agent optimally adapts its strategy to the behavior of the collective population. This leads to a mean-field *fixed-point condition* in which individual decisions shape and are shaped by the evolving population distribution. This formulation makes MFG a powerful tool for analyzing large-scale strategic interactions.

**Definition 2.1** (MFNE). A pair $(\pi_\star, \mu_\star)$ is said to be a MFNE if it satisfies the following two conditions:

1. **(Optimality under the population dynamics)** The policy $\pi_\star$ is an optimal solution to (3), given the population evolution $\mu_\star$, *i.e.*,

$$J^{\mathrm{MFG}}(\pi_\star, \mu_\star, \mu_\star) = \max_{\pi \in \Pi} J^{\mathrm{MFG}}(\pi, \mu_\star, \mu_\star) .$$

2. **(Consistency of the population evolution)** The population distribution $\mu_\star$ is a fixed point of the mean-field evolution equation

$$\mu_\star = \mu_\star \mathsf{P}_{\pi_\star}^{\mu_\star} .$$

In this definition, the first condition ensures that no individual agent can improve their long-term objective by unilaterally deviating from the equilibrium policy $\pi_\star$. Given the equilibrium population distribution $\mu_\star$, this optimality condition guarantees that the policy $\pi_\star$ remains the best possible strategy for each agent, thereby ensuring individual rationality.

The second condition enforces consistency in population dynamics: when all agents follow the equilibrium policy $\pi_\star$, the resulting population distribution remains stable over time. This fixed-point property ensures the system's long-term evolution does not deviate from the equilibrium state.

This notion of equilibrium, introduced by Nash (1950) and extended to the mean-field setting by Lasry & Lions (2006a;b; 2007); Huang et al. (2003; 2005; 2006a), is fundamental in game theory. In these works, the authors generalize the results of Nash (1951), proving the existence of at least one NE using the Brouwer Fixed-Point Theorem. This non-constructive result holds under mild assumptions, such as the continuity of payoff functions and the compactness of strategy spaces, forming a cornerstone for analyzing strategic interactions. However, explicitly computing such equilibria remains challenging, particularly in the multiplayer regime (Austrin et al., 2011).

**Exploitability.** A key metric for evaluating the deviation from a NE is exploitability (Laurière et al., 2022). Formally, the mean-field setting, it is defined as:

$$\phi(\pi, \mu) :=$$
$$\max_{\pi' \in \Pi} J^{\mathrm{MFG}}(\pi', \lambda_{\pi,\mu}, \lambda_{\pi,\mu}) - J^{\mathrm{MFG}}(\pi, \lambda_{\pi,\mu}, \lambda_{\pi,\mu}) . \tag{7}$$

This quantity measures the potential improvement an individual agent could achieve by deviating unilaterally from the learned policy $\pi$, given as mean-field parameter the stationary distribution $\lambda_{\pi,\mu}$.

**Definition 2.2.** A pair $(\pi_\varepsilon, \mu_\varepsilon)$ is said to be a $\varepsilon$-MFNE, if its exploitability is bounded by $\varepsilon$, *i.e.*, $\phi(\pi_\varepsilon, \mu_\varepsilon) \leq \varepsilon$.

## 3. Assumptions

We outline the key assumptions that guarantee the well-posedness and stability of the MFG problem, providing a foundation for deriving finite-sample complexity bounds for the proposed algorithm.

***A*-1.** Let $\mathsf{r}$ (resp. $\mathsf{P}$) be Lipschitz continuous with respect to $\mu$, with Lipschitz constant $L_\mu^{\mathsf{r}}$ (resp. $L_\mu^{\mathsf{P}}$), *i.e.*, for $s, s' \in \mathcal{S}$, $a \in \mathcal{A}$, and $\mu, \mu' \in \mathcal{P}(\mathcal{S})$, we have

$$|\mathsf{r}(s, a, \mu) - \mathsf{r}(s, a, \mu')| \leq L_{\mathsf{r}} \|\mu - \mu'\|_2 ,$$
$$|\mathsf{P}(s'|s, a, \mu) - \mathsf{P}(s'|s, a', \mu')| \leq L_{\mathsf{P}} \|\mu - \mu'\|_2 .$$

Lipschitz continuity for the MDP parameters ensures that small perturbations in the state or action lead to proportionally small changes in the transition dynamics. This assumption is well established in the RL literature (Asadi et al., 2018; Le Lan et al., 2021) and it facilitates the derivation of meaningful error bounds and convergence rates in policy optimization. From this, we establish Lipschitz continuity of the optimal policies w.r.t. mean-field parameter as follows (cf. Corollary E.4).

**Proposition 3.1.** *Suppose Assumption 1 holds. Then, there exists a constant $C_{\pi,\mu} \geq 0$ such that, for $\mu, \mu' \in \mathcal{P}(\mathcal{S})$,*

$$\sup_{s \in \mathcal{S}} \|\pi_\mu(\cdot|s) - \pi_{\mu'}(\cdot|s)\|_{\mathrm{TV}}^2 \leq C_{\pi,\mu} \|\mu - \mu'\|_2 ,$$

*where $\pi_\mu$ is the optimal policy associated with the mean-field distribution $\mu$.*

This result connects the structural properties of the MFG framework with the behavior of the associated optimal policies, forming a key theoretical foundation for the analysis of the proposed algorithms.

***A*-2.** There exist an integer $M \geq 1$ and a real number $C_{\mathrm{op,MFG}} < 1$ such that, for $\mu \in \mathcal{P}(\mathcal{S})$, we have

$$\left\langle \mu - \mu', \mu\left(\mathsf{P}_\mu^{\pi_\mu}\right)^M - \mu'\left(\mathsf{P}_{\mu'}^{\pi_{\mu'}}\right)^M \right\rangle$$
$$\leq C_{\mathrm{op,MFG}} \|\mu - \mu'\|^2 .$$

This monotonicity condition of the mean-field update is employed in various forms throughout the MFG literature (Angiuli et al., 2022; 2023; Yardim et al., 2023). This condition, originally introduced in the foundational works by Huang et al. (2003; 2005; 2006a;b), ensures that iterative updates to the population distribution progressively aligns the system with the NE.

This condition extends the classical contractivity condition, which corresponds to $M = 1$ (see, *e.g.*, Guo et al., 2019). The condition might fail for $M = 1$ but hold for some

$M > 1$, reflecting the combined effect of the Lipschitz continuity of the regularized best response and the ergodicity of the Markov reward process $\mathsf{P}_\mu^{\pi_\mu}$. A full in-depth discussion of this condition and its implications is provided in Appendix E.4. As the regularized best response admits a unique optimizer $\pi_\mu$, the previous condition ensures the uniqueness of the fixed point of the associated operator, *i.e.*,

$$\mu_\star = \mu_\star \mathsf{P}_{\mu_\star}^{\pi_{\mu_\star}} \ , \tag{8}$$

with $\mu_\star$ the unique fixed point of this operator. This implies that, in this setting, we obtain the *uniqueness of the MFNE*, corresponding to the pair $(\pi_{\mu_\star}, \mu_\star)$.

*A*-3. The MF-MDP $(\mathcal{S}, \mathcal{A}, \mathsf{P}, \mathsf{r}, \gamma)$ is *unichain*: for every fixed $\mu \in \mathcal{P}(\mathcal{S})$, every stationary policy $\pi$ induces a unique stationary distribution $\lambda_{\pi,\mu}$ over the state space, satisfying, for all $s \in \mathcal{S}$, the following equation:

$$\lambda_{\pi,\mu}(s) = \sum_{(s',a') \in \mathcal{S} \times \mathcal{A}} \mathsf{P}(s \mid s', a', \mu)\pi(a' \mid s')\lambda_{\pi,\mu}(s') \ .$$

Moreover, there exist constants $\rho \in (0, 1]$ and $C_{\text{Erg}} > 0$ such that the following uniform mixing property holds for all $\xi, \mu \in \mathcal{P}(\mathcal{S})$ and $t \geq 0$:

$$\left\| \xi \left( \mathsf{P}_\mu^{\pi_\mu} \right)^t - \lambda_{\pi_\mu,\mu} \right\|_{\text{TV}} \leq C_{\text{Erg}} \rho^t \ . \tag{9}$$

The unichain property eliminates ambiguities in the initial population distribution caused by multiple recurrent classes, which could otherwise complicate value function evaluation and policy improvement steps. As highlighted in Shani et al. (2020), optimal policies are defined only within the recurrent states of the Markov reward process at equilibrium. Moreover, the ergodicity property naturally aligns with the regularization framework, as it promotes exploration within the Markov chain, leading to a more robust and stable optimization landscape.

*A*-4. We have that

$$\sup_{\mu \in \mathcal{P}(\mathcal{S})} \sup_{s \in \mathcal{S}} \left| \frac{\overline{\mathsf{d}}_{\mu,\mu}^{\pi_\mu}(s)}{\nu(s)} \right| < \infty \ .$$

The structured interaction with the environment in the considered RL paradigm requires this concentration property of the occupation measure with respect to the reset distribution $\nu$. This property enhances sample efficiency and aids in the accurate estimation of key quantities, such as the value function.

## 4. Exact algorithm: `Exact MF-TRPO`

We now introduce `Exact MF-TRPO` to solve the optimization problem (3). Note that the entropic regularization

---

**Algorithm 1** `Exact TRPO`$(\mu)$

---
1: **Initialize:** $\pi_0$ is the uniform policy.
2: **Input:** $L$.
3: **for** $\ell \in [L]$ **do**
4:    $J^{\text{MFG}}(\pi_\ell, \mu, \mu) \leftarrow \mu(\mathsf{I} - \gamma \mathsf{P}_\mu^{\pi_\ell})^{-1} \mathsf{r}_\mu^{\pi_k}$
5:    $\mathcal{S}_{\mathsf{d}_{\nu,\mu}^{\pi_\ell}} := \{s \in \mathcal{S} : \mathsf{d}_{\nu,\mu}^{\pi_\ell}(s) > 0\}$
6:    **for** $s \in \mathcal{S}_{\mathsf{d}_{\nu,\mu}^{\pi_\ell}}$ **do**
7:      **for** $a \in \mathcal{A}$ **do**
8:        $Q_{\pi_\ell,\mu}(s, a) \leftarrow \mathsf{r}(s, a, \mu)$
                $+ \gamma \sum_{s'} \mathsf{P}(s'|s, a, \mu)J^{\text{MFG}}(\pi_\ell, \mu, s')$
9:      **end for**
10:      $\pi_{\ell+1}(a|s) \leftarrow$ `PolicyUpdate`$(\pi_\ell, Q_{\pi_\ell,\mu}; \ell)(a, s)$
11:    **end for**
12: **end for**
13: **Output:** $\pi_L$.

---

used in `Exact MF-TRPO` is particularly well-suited for TRPO (see, *e.g.*, Shani et al., 2020), as the proximal policy update fully exploits the entropy term, enabling a soft-max closed-form updates in terms of the $Q$-functions. This property enhances convergence by ensuring smoother and more stable policy iterations.

Although (3) is not a convex optimization problem, the adaptive nature of the regularization term allows us to use mirror descent techniques from convex analysis to establish strong convergence guarantees. In particular, we derive finite-sample complexity bounds showing that show that `Exact MF-TRPO` converges at a rate of $\widetilde{O}(1/L)$.

**TRPO.** For a fixed mean-field population distribution $\mu$, `Exact TRPO`$(\mu)$ provides a reliable approximation of the value function. The convergence of the algorithm is explicitly influenced by the regularization parameter $\eta$, which determines the optimal step size $1/(\eta(\ell + 2))$.

The proposed algorithm relies on the subroutine `PolicyUpdate`, a key step in refining the policy at each iteration. This update mechanism is inherently tied to the regularization scheme employed in the mirror ascent formulation of TRPO. Specifically, with entropic regularization, as shown in Beck (2017), the policy update admits a closed-form solution in the form of a softmax function. This structure inherently ensures that the updated policy remains within the probabilistic simplex without requiring additional projections. This fosters smooth and stable learning while preventing overly aggressive updates that could destabilize the optimization process.

This policy update leverages a function $Q: \mathcal{S} \times \mathcal{A} \rightarrow \mathbb{R}$ and a policy $\pi \in \Pi$ at iteration $\ell$, leading to an explicit update

**Algorithm 2** `Exact MF-TRPO`

---

1: **Input:** Initial distribution $\mu_0$, $K$.
2: **Initialize:** Initial policy $\pi_0$ is the uniform policy.
3: **for** $k \in [K]$ **do**
4:     $\pi_k \leftarrow$ `Exact TRPO`$(\mu_{k-1})$
5:     $\mu_k \leftarrow \mu_{k-1} + \beta_k \left( \mu_{k-1} \left( \mathsf{P}_{\mu_{k-1}}^{\pi_k} \right)^M - \mu_{k-1} \right)$
                     # population update
6: **end for**
7: **Output:** $\mu_K$.

---

rule given by

$$
\texttt{PolicyUpdate}(\pi, Q; \ell)(a, s)
$$
$$
= \frac{\pi(a|s) \exp\left(\alpha_\ell \left(Q(s,a) - \eta \log \pi(a|s)\right)\right)}{\sum_{a' \in \mathcal{A}} \pi(a'|s) \exp\left(\alpha_\ell \left(Q(s,a') - \eta \log \pi(a'|s)\right)\right)} ,
\tag{10}
$$

with learning rate $\alpha_\ell = 1/(\eta(\ell + 2))$.

Shani et al. (2020) establishes error bounds for the approximation of value functions that can be directly generalized to our setting. Building on these results, we quantify the gap between the value function of the computed policy and the optimal value function under the given mean-field distribution. This allows to control the policy itself, and derive policy guarantees.

**Proposition 4.1.** *Suppose that Assumptions 1 and 3 hold. Let $\{\pi_\ell\}_{\ell=0}^L$ be the sequence generated by the* `Exact TRPO`$(\mu)$ *algorithm. Then, there exists a constant $C > 0$ such that, for all $L \geq 1$, we have*

$$
\mathbb{E}_{s \sim \mu} \left[ \|\pi_L(\cdot|s) - \pi_\mu(\cdot|s)\|_{\mathrm{TV}}^2 \right] \leq C \frac{\log L}{L} .
$$

**MF-TRPO.** We now present `Exact MF-TRPO`, which iteratively solves the MFG problem (3) by updating the population distribution $\mu$ using the output of `Exact TRPO`$(\mu)$. This approach assumes direct access to the transition kernel and cost function, eliminating the need for stochastic approximation in policy updates.

The analysis of `Exact MF-TRPO` is instrumental to understand the performances of its sample-based counterpart. We do this providing precise theoretical guarantees on convergence rates without the additional complexity of sampling-induced errors. Building on this deterministic setup, we then focus on the convergence behavior of the broader `Sample-Based MF-TRPO` framework. We use the label *"informal"* to avoid overloading the main text with technical assumptions (*e.g.*, on learning rates), rigorously stated in Appendix C.

**Proposition 4.2** (informal). *Suppose that Assumptions 1, 2, and 3 hold. Then, there exists a constant $C > 0$ such that the sequence $\{\mu_k\}_{k \geq 0}$ generated by* `Exact MF-TRPO` *satisfies*

$$
\|\mu_k - \mu_\star\|_2^2 \leq 2 \exp\left( -\frac{\tau}{2} \sum_{j=1}^k \beta_j \right) + C \frac{\log(L)}{L} ,
$$

*for $k \geq 1$, with $\tau := 1 - C_{\mathrm{op,MFG}}$.*

All constants appearing in our results are explicitly defined and detailed in Appendix C, where we also provide complete proofs supporting our theoretical guarantees.

This convergence result proves its effectiveness in tackling the challenges inherent in the non-linear and non-gradient structure of MFNE. Below, we summarize the key insights derived from this result:

- **Convergence.** We establish an exponential rate of convergence in the *first term* of the bound to the equilibrium population distribution $\mu_\star$, while the *second one* captures the finite-sample bias in the best-response computation.

- **Learning Rates constraints.** The theorem provides explicit constraints on the step size $\beta_k$ (cf. condition (17)), ensuring stability and preventing oscillations or divergence in the optimization process.

- **Explicit Dependence on Model Parameters.** All constants in the convergence bound are fully characterized in terms of the structural parameters of the model (cf. Appendix C).

- **Controlled Policy Learning Bias.** The bias introduced by policy updates in `Exact TRPO`, bounded by $\log(L)/L$ (Proposition 4.2), remains controlled throughout the iterative process. This ensures algorithmic stability, even with approximations in policy optimization.

With these results, we can now explicitly quantify the parameter $\varepsilon$ corresponding to the proximity of our obtained solution with respect to the MFNE.

**Corollary 4.3.** *Suppose that Assumptions 1, 2, and 3 hold. Let $\mu_K$ (resp. $\pi_{K+1}$) the output of* `Exact MF-TRPO` *(resp.* `Exact TRPO`$(\mu_K)$*). Then, there exists a constant $C > 0$, such that $(\pi_{K+1}, \mu_K)$ is $\varepsilon_K$-MFNE, with*

$$
\varepsilon_K = C \exp\left( -\frac{\tau}{4} \sum_{j=1}^K \beta_j \right) + C \sqrt{\frac{\log(L)}{L}} .
$$

# 5. Stochastic approximation: `Sample-Based MF-TRPO`

We provide `Sample-Based MF-TRPO`, a model-free variant of the previous algorithm designed to operate without explicit knowledge of the environment's dynamics, nor

**Algorithm 3** `Sample-Based TRPO`$(\mu)$

1: **Initialize:** $\pi_0(\cdot|s) = \mathcal{U}(\mathcal{A})$ for $s \in \mathcal{S}$.
2: **Input:** $I_\ell, T_\ell, \epsilon, \delta > 0, L$.
3: **for** $\ell \in [L]$ **do**
4:     $S_\ell^{I_\ell} = \{\}$
5:     $Q_{\hat{\pi}_\ell, \mu}(s, a) = 0, n_\ell(s, a) = 0$, for any $(s, a)$
6:     **for** $i = 1, \ldots, I_\ell$ **do**
7:         Sample $s_i \sim \bar{\mathsf{d}}_{\nu,\mu}^{\hat{\pi}_\ell}(\cdot), a_i \sim \mathcal{U}(\mathcal{A})$
8:         $Q_{\hat{\pi}_\ell, \mu}(s_i, a_i, i) \leftarrow \mathsf{r}(s_i, a_i, \mu) +$
            $\sum_{t=1}^{T_\ell} \gamma^t \mathbb{E}_{s_t \sim \delta_{s_i}(\mathsf{P}_\mu^{\hat{\pi}_\ell})^t, a_t \sim \hat{\pi}_\ell(\cdot|s_t)} \big[ \mathsf{r}(s_t, a_t, \mu)$
                                        $+ \eta \log(\hat{\pi}_\ell(a_t|s_t)) \big]$
9:         $Q_{\hat{\pi}_\ell, \mu}(s_i, a_i) \leftarrow Q_{\hat{\pi}_\ell, \mu}(s_i, a_i) + Q_{\hat{\pi}_\ell, \mu}(s_i, a_i, i)$
10:        $n_\ell(s_i, a_i) \leftarrow n_\ell(s_i, a_i) + 1$
11:        $S_\ell^{I_\ell} = S_\ell^{I_\ell} \cup \{s_i\}$
12:    **end for**
13:    **for** $s \in S_\ell^{I_\ell}$ **do**
14:        **for** $a \in \mathcal{A}$ **do**
15:            $Q_{\hat{\pi}_\ell, \mu}(s, a) \leftarrow \frac{|\mathcal{A}|Q_{\hat{\pi}_\ell, \mu}(s, a)}{\sum_{a' \in \mathcal{A}} n_\ell(s, a')}$
16:        **end for**
17:        $\hat{\pi}_{\ell+1}(a|s) \leftarrow$ `PolicyUpdate`$(\hat{\pi}_\ell, Q_{\hat{\pi}_\ell, \mu}; \ell)(a, s)$
18:    **end for**
19: **end for**
20: **Output:** $\hat{\pi}_L^{\texttt{Unif}, \hat{\mu}_k}$.

the reward function. `Sample-Based MF-TRPO` utilizes sampled trajectories to estimate these updates, making it more applicable to real-world scenarios.

**TRPO.** First, we adapt `Exact TRPO` to estimate policy updates in a data-driven manner. This approach leverages sampled trajectories to approximate the policy gradient, providing quantitative bounds on the proximity of the best response. Additionally, this TRPO formulation is particularly well-suited to the considered oracle-based framework, incorporating the $\nu$-restart modeling. This structure ensures robust exploration while maintaining stability in policy updates, aligning naturally with the trust-region optimization paradigm.

The inherent stochasticity in the updates prevents us from establishing sample complexity bounds on the last iterate of the algorithm. However, by leveraging an averaging scheme, implemented through a dedicated subroutine (cf. Remark D.1), we mitigate this variability and provide clear quantitative bounds on the desired gap. Specifically, the policy we focus on is the *uniform mixture* $\hat{\pi}_L^{\texttt{Unif}, \mu}$ over the first $L + 1$ policies. This averaging scheme is standard in the RL literature and, in the unregularized case, satisfies the identity

$$\frac{1}{L+1} \sum_{\ell=0}^{L} J^{\text{MFG}}(\hat{\pi}_\ell, \mu, \mu) = J^{\text{MFG}}(\hat{\pi}_L^{\texttt{Unif}, \mu}, \mu, \mu) \,.$$
(11)

While we do not have a statistically feasible expression for this mixture policy, it is straightforward to sample from $\hat{\pi}_L^{\texttt{Unif}, \mu}$, using `Uniform-Mixture`$(\{\hat{\pi}_\ell\}_{\ell=0,\ldots,L})$, as discussed in Remark D.1.

**Proposition 5.1.** *Suppose Assumptions 1, 2, 3, and 4 hold. Fix $\epsilon, \delta > 0$. Let $\hat{\pi}_L^{\texttt{Unif}, \hat{\mu}_k}$ be the output of* `Sample-Based TRPO`$(\mu)$*, over L iterations.*

*Then, there exists $C > 0$ such that the following holds with probability greater than $1 - \delta$*

$$\mathbb{E}_{s \sim \mu} \left[ \left\| \hat{\pi}_L^{\texttt{Unif}, \mu}(\cdot|s) - \pi_\mu(\cdot|s) \right\|_{\text{TV}}^2 \right] \leq C \left( \frac{\log L}{L} + \epsilon \right) \,.$$

All the constants appearing in our results are explicitly defined and detailed in Appendix D, where we also provide the complete proofs supporting our theoretical guarantees.

**MF-TRPO.** We present a version of `Sample-Based MF-TRPO`, with the full algorithm and detailed implementation provided in Appendix D.3.

One key aspect of the algorithm is the initialization step. Unlike in a generative model paradigm, access to a state $s \sim \mu$ is only available through a subroutine initialized at the restart distribution $\nu$. For further details on this procedure, we refer to Appendix D.2.

The theoretical foundation of the convergence guarantees of this algorithm relies on deriving high-probability estimates

**Algorithm 4** `Sample-Based MF-TRPO` (informal)

1: **Input:** Initial distribution $\mu_0, K$.
2: **Initialize:** Initial policy $\pi_0$ is the uniform policy.
3: **for** $k \in [K]$ **do**
4:     $\hat{\pi}_k \leftarrow$ `Sample-Based TRPO`$(\hat{\mu}_{k-1})$.
5:     **for** $p \in [P]$ **do**
6:         Initialize $s_{0,p,k} \sim \hat{\mu}_{k-1}$.
7:         **for** $m \in [M]$ **do**
8:             $s_{m,p,k} \sim \mathsf{P}_{\hat{\pi}_k}^{\hat{\mu}_{k-1}}(\cdot|s_{m-1,p,k})$
9:         **end for**
10:        $\hat{\zeta}_{k,p} \leftarrow \mathbb{1}_{\{s_{M,p,k}\}}(\cdot)$.
11:    **end for**
12:    $\hat{\zeta}_k \leftarrow \frac{1}{P} \sum_{p=1}^{P} \hat{\zeta}_{k,p}$.
13:    $\hat{\mu}_k \leftarrow \hat{\mu}_{k-1} + \beta_k \left( \hat{\zeta}_k - \hat{\mu}_{k-1} \right)$
14: **end for**
15: **Output:** $\mu_K$.

(cf. Proposition D.3). This is a crucial step in sample complexity bounds and is achieved using a martingale-based argument (Harvey et al., 2019). By leveraging concentration inequalities for martingales, the analysis ensures that the error in estimating key quantities, such as the policy value and state distributions, remains controlled with high probability throughout the learning process.

**Proposition 5.2** (informal). *Suppose Assumptions 1, 2, 3, and 4 hold. Fix $\epsilon, \delta > 0$. Then, there exists a constant $C > 0$ such that the sequence $\{\mu_k\}_{k \geq 0}$ generated by* `Sample-Based MF-TRPO` *satisfies, for $k \geq 1$,*

$$\|\mu_k - \mu_\star\|_2^2 \leq 2 \exp\left(-\frac{\tau}{2} \sum_{j=1}^{k} \beta_j\right) + C \frac{\log(L)}{L} + C\epsilon \, .$$

All the constants appearing in our results are explicitly defined and detailed in Appendix D, where we also provide the complete proofs supporting our theoretical guarantees.

The sample-based convergence analysis of `Sample-Based MF-TRPO` in the model-free setting demonstrates that the algorithm preserves the same desirata described in Section 4 driving the convergence of its exact counterpart `Exact MF-TRPO`—proximal updates, entropic regularization, and trust-region optimization—remain intact. While having a model-agnostic nature, its theoretical power of estimation of the MFNE is preserved, as explicit knowledge of the environment's transition dynamics or reward structure is required. It builds on trajectory sampling to iteratively refine policy updates while maintaining stability and efficiency.

This result entails that the stochastic error in the estimation of the mean-field distribution at each iteration does not compound throughout the iterative process. By leveraging concentration inequalities and high-probability guarantees, the cumulative impact of these estimation errors remains controlled, preventing divergence or instability. As a result, `Sample-Based MF-TRPO` retains strong convergence guarantees, making it a practical approach for solving MFG in a data-driven manner.

Moreover, we can also bound the exploitability as follows.

**Corollary 5.3.** *Suppose that Assumptions 1, 2, 3, and 4 hold. Fix $\epsilon, \delta > 0$. Let $\hat{\mu}_K$ (resp. $\hat{\pi}_L^{Unif,\hat{\mu}_K}$) the output of* `Sample-Based MF-TRPO` *(resp. the iterates of* `Sample-Based TRPO`$(\hat{\mu}_k)$*). Then, there exists a constant $C > 0$ such that*

$$\phi(\hat{\pi}_L^{Unif,\hat{\mu}_K}, \hat{\mu}_k)$$
$$\leq C \left( \exp\left(-\frac{\tau}{4} \sum_{j=1}^{K} \beta_j\right) + \sqrt{\frac{\log(L)}{L}} + \epsilon \right) \, .$$

The complexity analysis demonstrates that the proposed `Sample-Based MF-TRPO` algorithm achieves a computational cost scaling as $\widetilde{O}(1/\varepsilon^6)$ to reach an $\varepsilon$-MFNE (cf. Remark D.7). This scaling emerges naturally from two principal contributions: the inner loop performing the policy optimization via `Sample-Based TRPO`, and the outer population distribution update. These results align well with established convergence bounds in Shani et al. (2020), emphasizing a balanced trade-off between computational efficiency and the precision required to approximate the MFNE.

## 6. Numerical Experiments

We present here the results of the numerical experiments obtained with the `Sample-Based MF-TRPO` algorithm. The environment considered is a Grid-Based Crowd Modeling game where, from a given initial distribution, agents are tasked with moving through a grid, avoiding both static obstacles and potential overcrowding. A representative player's state corresponds to her position within the grid, and at every time step, she can choose to move in any direction or stay in place. The reward structure imposes a small penalty for movement, offers a slight incentive for staying, and discourages agents from entering overcrowded areas. In addition, agents are encouraged to move toward a designated target, that is,

$$\tilde{r}(s, a, \mu) = r(s, a, \mu) + \max\{0.3 - 0.1 \cdot d(s, s_{\text{target}}); \ 0\} \, ,$$

where $d(\cdot, \cdot)$ is a $\ell_1$-distance between the corresponding states, and the crowd reward is defined as

$$r(s, a, \mu) = -\kappa \log(\mu(s)) + \Gamma(a) \, ,$$

where $\Gamma(a) = 0.2 \cdot 1_{\{a = \text{Stay}\}} - 0.2 \cdot 1_{\{a \neq \text{Stay}\}}$, with 1 being the indicator function and $\kappa$ being a crowd-aversion parameter. We refer to Appendix F for additional details and experimental results related to the `Exact MF-TRPO` algorithm. The environment used here is a $5 \times 5$ grid featuring three walls located at coordinates $(1, 2)$, $(2, 2)$, and $(3, 2)$. The point of interest is located in the bottom-right corner of the grid, and all the players start clustered in the top-left corner. In Figure 1 it is possible to observe that the exploitability behavior of the `Sample-Based MF-TRPO` algorithm converges after a few iterations, matching the theoretical predictions, whereas Figure 2 illustrates the progression of the mean field distribution across three different time steps during the learning phase, thus demonstrating that the players progressively learn to distribute themselves around the point of interest, preserving spread over the whole state space.

## 7. Conclusion

In this work, we introduced `Exact MF-TRPO`, a novel algorithm for computing MFNE in ergodic MFG. By leveraging the trust-region policy optimization framework, we established explicit non-asymptotic convergence guarantees, demonstrating that `Exact MF-TRPO` inherits the $\widetilde{O}(1/L)$ rate from TRPO, ensuring efficient learning in structured multi-agent systems.

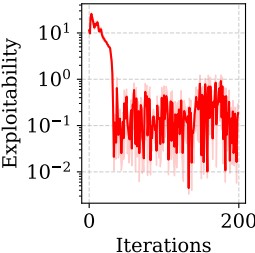 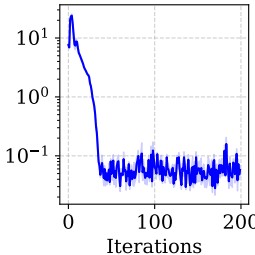

*Figure 1.* Exploitability achieved by the `Sample-Based MF-TRPO` algorithm in the $5 \times 5$ Grid-Based Crowd Modeling game with the bottom-right corner being a point of interest. The left plot corresponds to $\eta = 0.05$, and the right to $\eta = 0.3$ with results averaged over 10 and 3 random seeds, respectively.



*Figure 2.* Evolution of the mean field distribution for $\eta = 0.05$ in the $5 \times 5$ Grid-Based Crowd Modeling game with the bottom-right corner being a point of interest. From left to right: step 0, step 10 and step 200.

To bridge the gap between theoretical guarantees and practical applicability, we further developed `Sample-Based MF-TRPO`, a model-free variant that estimates policy updates solely from sampled trajectories, under the $\nu$-restart RL paradigm. Using concentration inequalities, we provided finite-sample complexity bounds for this algorithm, proving convergence under more relaxed assumptions compared to recent literature. Moreover, we show that a total number of calls to the environment that scales as $\widetilde{O}(1/\varepsilon^6)$, consistent with standard RL results (see, *e.g.*, Shani et al., 2020). This result highlights the potential of RL techniques for scalable and data-driven MFG solutions.

Overall, our work contributes to the growing intersection of MFG and RL, providing both theoretical insights and algorithmic advancements. Future directions include extending these methods to more general MFG settings, such as those with continuous state spaces, and exploring adaptive sampling techniques to further improve efficiency in real-world applications.

## Acknowledgements

The work of A.O. and E.M. was funded by the European Union (ERC-2022-SYG-OCEAN-101071601). Views and opinions expressed are however those of the author only and do not necessarily reflect those of the European Union or the European Research Council Executive Agency. Neither the European Union nor the granting authority can be held responsible for them. The work of L.M. and E.M. has been supported by Technology Innovation Institute (TII), project Fed2Learn. The work of D.T. has been supported by the Paris Île-de-France Région in the framework of DIM AI4IDF.

## Impact Statement

This paper presents work whose goal is to advance the field of Machine Learning. There are many potential societal consequences of our work, none which we feel must be specifically highlighted here.

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

# Appendix

In the appendix, we provide a detailed exposition of the theoretical foundations and technical results supporting our main contributions. In Appendix A, we give a broad description of related work in MFG and MFRL, and in Appendix B, we lay out the fundamental framework of the MFG problem we aim to study. This section introduces the ergodic MFG formulation and its connection to the discounted setting. We also discuss the role of entropic regularization in RL and MFG, emphasizing its impact on stability and the approximation of Nash equilibria. In Appendix B.4, we present key assumptions we use in the proofs. In Appendix C, we present the exact TRPO-based algorithm for solving the MFG problem. This section provides a precise formulation of the exact methods and establishes its theoretical convergence guarantees. The adaptation of TRPO to the MFG setting is examined in detail, leveraging the structure of entropy-regularized RL. We present formal results on convergence rates and error bounds, ensuring the effectiveness and reliability of these methods in computing approximate MFNE. In Appendix D, we extend our analysis to the sample-based version of the algorithm. Here, we derive global sample complexity results and analyze the statistical error introduced by sampling. We establish high-probability bounds on the approximation error at each iterations. Next, in Appendix E, we provide additional proofs and auxiliary results that support the theoretical analysis conducted in the previous sections. These supplementary results play a crucial role in rigorously validating the convergence and stability properties of our proposed algorithms. Finally, in Appendix F we provide additional experimental details as well as experiments on exact versions of the algorithm.

# A. Related works

Except for the well-known Linear-Quadratic (LQ) case, where explicit solutions can be derived analytically or through simple ordinary differential equations, computing MFNE numerically remains a challenging and active research area. A vast body of literature has focused on addressing the computational complexity of these models, leading to three major methodological approaches.

The first class of methods relies on PDE approximations, leveraging the classical formulation of MFG through the Hamilton–Jacobi–Bellman equation coupled with the Fokker–Planck–Kolmogorov equation (Achdou & Capuzzo-Dolcetta, 2010; Achdou et al., 2012; Achdou & Porretta, 2016; Achdou & Laurière, 2020). While mathematically elegant, these methods suffer from the well-documented curse of dimensionality, as solving PDEs numerically becomes intractable in high-dimensional state spaces.

The second approach leverages deep learning techniques with neural networks to approximate equilibrium solutions. These methods exploit function approximation to bypass explicit PDE resolution, making them a promising alternative in high-dimensional settings (Weinan et al., 2017; Chassagneux et al., 2019; Germain et al., 2022). However, they lack rigorous convergence guarantees. Additionally, these methods struggle to capture model specifications in a purely data-driven manner directly, limiting their adaptability in real-world applications.

The third category of numerical methods integrates RL techniques into the MFG framework, leading to two primary subcategories. The first subcategory employs RL as a solver for a given MFG model, using value-based or policy-based RL techniques to approximate Nash equilibria (Pérolat et al., 2022; Perrin et al., 2022). These approaches have achieved state-of-the-art performance in various settings and have been successfully deployed in large-scale simulations.

The second subcategory focuses on developing model-free RL algorithms for solving MFG, often incorporating regularization techniques to enhance stability (Cui & Koeppl, 2021; Perrin et al., 2021). While these methods show promising empirical performance, theoretical guarantees on finite-sample complexity remain limited, particularly for model-based RL approaches.

**RL for MFG.** Among model-free RL approaches, we identify three main categories: value function-based methods, actor-critic methods—which combine value-based and policy-based approaches—and policy-oriented methods. As in classical RL, proximal methods have emerged as state-of-the-art techniques across a wide range of tasks, both in model-specific and model-agnostic settings.

Several studies have explored $Q$-learning-based algorithms for MFG, establishing theoretical convergence guarantees (Angiuli et al., 2022; 2023; Guo et al., 2019; Anahtarci et al., 2023b). However, these approaches rely on stringent assumptions that are often difficult to verify in practice. Moreover, $Q$-learning operates within the *generative model* oracle framework of RL, which assumes full query access to state-action transitions. In contrast, the restart oracle assumption provides a significantly weaker paradigm, offering a more practical and adaptable alternative for real-world learning settings.

Two-timescale updates have been employed in, *e.g.*, Zaman et al. (2023) and Mao et al. (2022), where policy updates operate on a faster timescale, while the population distribution evolves more slowly based on model-based estimates of state dynamics. However, these approaches introduce significant challenges in both theoretical analysis and practical implementation. Their convergence often relies on strong assumptions that are difficult to verify, particularly in non-stationary environments.

In the MFG setting, proximal methods have been adopted for their stability properties and strong empirical performance. Among these approaches, we highlight the work of Pérolat et al. (2022), where the authors investigate Online Mirror Descent (OMD) from a model-specific perspective. Yardim et al. (2023) build up in this direction, developing the model-free analysis. Their approach, however, does not consider the crucial aspect of population updates, placing itself in the no-manipulation regime. In contrast, our work explicitly discusses how the monotonicity assumption can stabilize the population evolution up to a certain threshold, a perspective that aligns naturally with the ergodic nature of the Markov reward process once a policy has been selected. This additional consideration allows us to provide a more refined analysis of the learning dynamics in MFG, ensuring a more structured and well-posed approach to policy optimization.

Furthermore, Yardim et al. (2023) imposes stringent assumptions by requiring that policies remain uniformly bounded away from zero by a fixed constant, effectively enforcing an overly rigid form of regularization. This constraint exceeds the controlled bias typically introduced by entropic regularization. Notably, a similar assumption appears in Angiuli et al. (2022; 2023), where the Markov reward process is required to be absolutely continuous with respect to the uniform distribution over the state-action space—a significantly stronger condition than our unichain assumption.

By adopting a more flexible framework, we relax these restrictive conditions while improving sample complexity, particularly in terms of the number of required trajectories. Our analysis still achieves an $\widetilde{O}(1/L)$ error bound in trajectory estimates but under significantly milder assumptions.

An interesting direction for future work would be to connect our framework with that of Anand et al. (2024), which focuses on mean-field control. While their setting assumes cooperative agents optimizing a common objective, our work addresses non-cooperative mean-field games. Despite this key difference, it would be valuable to explore how our analytical tools—particularly for handling weak structural assumptions—could generalize their approach to broader settings.

## B. Framework

### B.1. Ergodic MFG problem

The ergodic problem focuses on optimizing the long-term average performance of a stochastic system over an infinite time horizon. In contrast to finite time horizon problems, the emphasis is on stability and efficiency over time, making it essential for operations such as energy systems, financial markets and resource management. The goal is to find stationary policies that balance short-term costs with long-term gains. This approach is robust to uncertainties and ensures consistent performance despite stochastic disturbances.

Ergodic MFG have been first studied in continuous time and space problems (see, *e.g.*, Bardi & Priuli, 2014; Arapostathis et al., 2017; Carmona & Laurière, 2021). In the context of discrete-time MFG, the ergodic setting has been studied by Saldi (2020) under the terminology of average-cost MFG. Anahtarci et al. (2023a) proposed a learning algorithm based on $Q$-learning and analyzed its convergence and sample complexity using a strict contraction argument.

Similarly, in Guo et al. (2023), the authors address the problem of evolving mean-field parameters, focusing on dynamically adjusting the population distribution over time. This differs from our approach, where we aim to learn policies without requiring explicit control or manipulation of the mean-field distribution at each step.

### B.2. From ergodic MFG to discounted formulation

In this article, we focus on an ergodic equilibrium problem within the framework of mean-field games. This problem is traditionally defined as the unique solution resulting from the optimization of a long-term average cost function

$$J_{\text{erg}}^{\text{MFG}}(\underline{\pi}, \mu, \xi) := \liminf_{T \to \infty} \frac{1}{T} \mathbb{E}\left[ \sum_{t=0}^{T} \mathsf{r}\left(s_t, a_t, \mu_t\right) \,\middle|\, \begin{array}{c} s_0 \sim \xi,\ a_t \sim \pi_t(\cdot|s_t),\ \mu_{t+1} = \mu_t \mathsf{P}_{\pi_t, \mu_t} \\ s_{t+1} \sim \mathsf{P}\left(\cdot|s_t, a_t, \mu_t\right) \end{array} \right],$$

for $\underline{\pi} = (\pi_t)_{t=0}^{\infty}$. We seek Nash equilibria with respect to this cost function, *i.e.*, a tuple $(\underline{\pi}^{\star}, \mu, \xi)$ such that

- $J_{\mathrm{erg}}^{\mathrm{MFG}}(\underline{\pi}^\star, \mu, \xi) = \max_{\underline{\pi}} J_{\mathrm{erg}}^{\mathrm{MFG}}(\underline{\pi}, \mu, \xi)$ ;

- $\mathcal{L}(s_t) = \mu_t$.

Such a problem is independent of the initial condition $\xi$ and is stationary. Therefore, we can consider only constant vectors $\underline{\pi} = (\pi_t)_{t=0}^\infty$ for $\pi_t = \pi$, for any $t \geq 0$, a$\pi \in \Pi$.

Moreover, given the stationarity of both the policy and the problem, we can further restrict ourselves to time-invariant policies and reformulate the problem as follows.

$$J_{\mathrm{erg,stat}}^{\mathrm{MFG}}(\pi, \mu) := \liminf_{T \to \infty} \frac{1}{T} \mathbb{E}\left[ \sum_{t=0}^T \mathsf{r}(s_t, a_t, \mu) \middle| \substack{s_0 \sim \mu,\ a_t \sim \pi_t(\cdot|s_t), \\ s_{t+1} \sim \mathsf{P}(\cdot|s_t, a_t, \mu)} \right] ,$$

Moreover, as also noted in Carmona et al. (Chapter 7.1, Volume 1, 2018), we have that

$$J_{\mathrm{erg,stat}}^{\mathrm{MFG}}(\pi, \mu) := \lim_{\gamma \to 1} \frac{1}{1 - \gamma} J_\gamma^{\mathrm{MFG}}(\pi, \mu, \mu) ,$$

with

$$J_\gamma^{\mathrm{MFG}}(\pi, \mu, \xi) = \mathbb{E}\left[ \sum_{t=0}^\infty \gamma^t \mathsf{r}(s_t, a_t, \mu) \middle| \substack{s_0 \sim \xi,\ a_t \sim \pi_t(\cdot|s_t), \\ s_{t+1} \sim \mathsf{P}(\cdot|s_t, a_t, \mu)} \right] .$$

For this reason, for a $\gamma$ close to 1, we can consider the problem as formulated in Section 2. This is also the consideration behind the formulation of the ergodic problem presented in Laurière et al. (Remark 8, 2022).

### B.3. Regularization

**Entropy regularization in RL.** Entropy regularization has been a prominent concept across various fields, including RL (Sutton & Barto, 2018b; Szepesvári, 2022). In dynamic programming and RL contexts, entropy-regularized Bellman equations and corresponding algorithms have been extensively studied to address key challenges. These include inducing safe exploration (Fox et al., 2016) and designing risk-sensitive policies (Howard & Matheson, 1972; Marcus et al., 1997; Ruszczyński, 2010). Additionally, these methods have been employed to model behaviors of imperfect decision-makers, as demonstrated by Ziebart et al. (2010); Ziebart (2010); Braun et al. (2011).

Beyond dynamic programming approaches, direct policy search methods have emerged as a powerful alternative for optimizing entropy-regularized objectives. These methods, which aim to drive safe online exploration in unknown Markov decision processes, have been explored in works such as Williams & Peng (1991); Peters et al. (2010); Schulman et al. (2015); Mnih (2016); O'Donoghue et al. (2017). Notably, state-of-the-art RL methods, including those by Mnih (2016); Schulman et al. (2015), leverage entropy-regularized policy search to balance exploration and exploitation effectively, highlighting the central role of regularization in achieving robust and safe learning.

Regularization, particularly the entropic one, has been extensively studied in the theoretical literature. In Neu et al. (2017), the authors provide a comprehensive analysis of mirror descent methods for RL, highlighting how regularization influences policy optimization and convergence properties. Similarly, Geist et al. (2019) formalize the theoretical impact of entropy regularization, demonstrating its role in stabilizing policy updates and improving exploration.

**Regularization in RL for MFG.** In the inherently non-linear MFG setting, stabilizing policy updates is essential to ensuring convergence and preventing oscillatory behavior. In this setting, the underlying dynamics involve the interplay of numerous agents and necessitate a precise balance between individual and collective objectives. Regularization plays a key role by smoothing the cost landscape, mitigating instability, and creating well-conditioned optimization problems, ultimately leading to more reliable and efficient learning dynamics.

Moreover, MFG serve as approximations of the $N$-player game problem in MARL, leveraging assumptions like *anonymity* and *homogeneity* to simplify the otherwise intractable dynamics of joint policy updates in large-scale systems. Introducing regularization into the MFG framework not only enhances stability but is theoretically justified, as the additional

approximation error introduced by regularization is comparable to the inherent $\mathcal{O}(1/\sqrt{N})$ error of the MFG approximation itself.

Moreover, in the context of RL for MFG, regularize policies has been used to facilitates convergence of learning algorithms. Cui & Koeppl (2021) show that strict contraction property used by several other works fail to holds in general. The authors then studied a modified MFG with an entropy-regularized reward and showed that, for a sufficiently large degree of regularization, policy-iteration type RL algorithms can be shown to converge using contraction techniques. A similar approach has been used by Anahtarci et al. (2023b) to prove convergence of a $Q$-learning algorithm, and by Yardim et al. (2023) to prove the convergence of policies learned by independent learners in a regularized MFG. On the empirical side, policy regularization has also been used by Algumaei et al. (2023) through an algorithm relying on proximal policy optimization (PPO).

### B.4. Discussion on the assumptions

**Lipschitz property of the MDP parameters.** Assumption 1 of Lipschitz continuity on the parameters of the MDP, reward function r and transition probability matrix P, implies that the MDP does not change abruptly with respect to the state or action. This ensures that small perturbations in these variables lead to correspondingly small changes in the MDP's dynamics.

This assumption is well-established in the RL literature (see, *e.g.*, Asadi et al., 2018; Le Lan et al., 2021) and serves several critical purposes. First, it ensures the smoothness of value functions, essential for the stability of iterative optimization methods Second, it enables the derivation of meaningful error bounds and convergence rates, as shown in foundational works on policy optimization.

Moreover, as the reward function r is defined on a compact domain, since $\mathcal{S}$ and $\mathcal{A}$ are finite, it is guaranteed that $\|r\|_\infty < \infty$. A common practice in RL literature to normalize r, *i.e.*, $\|r\|_\infty = 1$, without loss of generality (Mei et al., 2020). This normalization simplifies expressions and makes algorithms scale-independent. However, in this work, we retain $\|r\|_\infty$ explicitly in our analysis to emphasize the clear dependence of all constants and bounds on the magnitude of the reward function. This approach ensures transparency in how the properties of r influence the theoretical results and practical performance.

**Unique recurrence property of the Markov reward process.** To ensure the well-posedness of the TRPO algorithm and derive meaningful performance bounds, we impose the unichain assumption 3. This assumption guarantees that the Markov chain induced by any admissible policy has a single recurrent class, potentially accompanied by a set of transient states. Such a property ensures that the long-term behavior of the Markov chain is well-defined, with a unique stationary distribution for each policy.

The unichain property plays a pivotal role in stabilizing the analysis of RL algorithms, particularly TRPO, as highlighted in Neu et al. (2017). It eliminates ambiguities on the initial population distribution arising from multiple recurrent classes, which could otherwise complicate the evaluation of value functions and policy improvement steps. As shown in in Puterman & Shin (1978), this condition is satisfied if all policies induce an irreducible and aperiodic Markov chain. Moreover, in the regularized setting, the regularization term helps in its satisfaction. Therefore, for any mean-field population profile $\mu$, the Markov chain $\mathsf{P}_\mu^{\pi_\mu}$ is irreducible and aperiodic, implying the existence of a unique stationary distribution $\lambda_{\pi_\mu, \mu}$ and establishes the foundation for the mixing property (9).

The ergodicity assumption has been explored in various forms by different authors in the MFG literature. Notably, Angiuli et al. (2022; 2023) impose the condition that the induced Markov chain is aperiodic and absolutely continuous with respect to the uniform distribution over the state space. While this guarantees strong mixing properties, it is a highly restrictive assumption, as it effectively enforces immediate communication between all states, which is often unrealistic in practical applications. Instead, we generalize this assumption by adopting a standard ergodicity condition widely used in RL literature (Mei et al., 2020). This approach maintains the necessary stability properties while allowing for more realistic transition dynamics, ensuring broader applicability in complex multi-agent systems.

**Finite concentration of the occupation measure.** In this paper, we adopt a RL paradigm where access to the environment is structured through a $\nu$-restart, ensuring that each learning episode begins from a well-defined initial state distribution. The concentration of the occupation measure assumption is crucial for the convergence of the `Sample-Based TRPO` algorithm, ensuring that the estimation of the policy update remains stable and accurate over successive iterations. To

compute this task, the algorithm operates in an episodic setting, where each episode begins by drawing an initial state $s_0$ from the restart distribution $\nu$, followed by collecting a trajectory $(s_0, r_0, s_1, r_1, \dots)$ under the current policy $\pi_k$. This episodic structure allows the algorithm to interact with the MDP in a controlled manner, facilitating the estimation of quantities like the value function $J^{\mathrm{MFG}}$.

This approach builds on the seminal work of Kakade (2003), which introduced the notion of a $\nu$-restart model as an intermediary assumption in RL. The $\nu$-restart model is weaker than having direct access to the true model or a generative model (Kearns & Singh, 1998; Azar et al., 2013; Sidford et al., 2018; Agarwal et al., 2020), as it does not require full knowledge of the transition kernel or the reward function. At the same time, it is stronger than the unrestricted case where no restarts are allowed, ensuring that the algorithm can sample states from a well-defined initial distribution at the start of each episode. This controlled interaction with the environment is crucial for accurately estimating value functions and gradients in the `Sample-Based TRPO` algorithm, ultimately enabling convergence guarantees.

The supremum in Assumption 4, often referred to as the concentrability coefficient, plays a critical role in the theoretical analysis of policy search algorithms. This concept was initially highlighted in the foundational work of Kakade & Langford (2002) and has since been extensively studied in the RL (RL) literature.

One of the reasons the concentrability coefficient has garnered attention is its frequent appearance in the analysis of approximate policy iteration schemes. Research by Scherrer & Geist (2014) and Bhandari & Russo (2024) shows that the concentrability coefficient often governs error propagation during learning. In essence, it provides bounds on how errors in approximating value functions or policies propagate through successive updates.

## C. Exact algorithms

Various algorithms have been proposed in the literature to address the exact MFG problem in scenarios where the MDP kernel and the reward function are fully accessible. In this case, the value function and a best response can be computed using dynamic programming and backward induction. This approach has been used, *e.g.*, by Perrin et al. (2020) and Pérolat et al. (2022) to implement Fictitious Play (FP) and Online Mirror Descent (OMD) respectively. Cui & Koeppl (2022) presented a exact fixed point algorithm for graphon games. Angiuli et al. (2023) analyzed the convergence of a model-specific multi-scale algorithm for MFG.

This line of research often stems from the classical control theory and optimization frameworks, tailored to solve specific MFG problems with high precision. These methods focus on the exact representation of the MFG dynamics, providing critical insights into the equilibrium behavior of large-agent systems. In this work, we propose a novel adaptation of the TRPO algorithm, building on the robust framework of Shani et al. (2020). Our adaptation incorporates key elements of the MFG structure, leveraging entropic regularization and mean-field population dynamics. Moreover, we establish finite sample complexity results for this algorithm, demonstrating its theoretical convergence properties and its practical applicability in solving the ergodic MFG problem under a finite state-action setting.

### C.1. TRPO - exact formulation

TRPO, inherently structured as a mirror descent method, proves particularly well-suited for entropy-regularized settings. This framework benefits from a significant simplification: the policy update admits a closed-form solution (Beck, 2017), expressed in terms of the $Q$-function associated with the current policy. By recasting the optimization problem in terms of $Q$-function computation, the algorithm focuses on the essential dynamics of the system, effectively bypassing the need for direct policy optimization over a high-dimensional space.

This closed-form update leverages the softmax form of the policy, a direct consequence of entropy regularization. The softmax structure ensures that the updated policies remain strictly in the interior of the probability simplex $\mathcal{P}(\mathcal{S})$, avoiding deterministic solutions. This property not only facilitates numerical stability but also aligns with the theoretical foundations of the regularized problem. The use of first-order conditions becomes feasible and efficient, as the regularization term enforces a smooth, convex optimization landscape.

Moreover, the reward function's linear dependence on the policy pairs seamlessly with the coercive nature of the entropy-regularized optimization problem. The coercivity guarantees that the optimal policies minimize the objective within the confines of the simplex, effectively balancing exploration and exploitation. This alignment between the problem structure and the algorithm's mechanics underscores the power of TRPO in achieving convergence while maintaining

theoretical guarantees in entropy-regularized MFG settings. By translating the original optimization problem into $Q$-function evaluations, the algorithm provides a practical yet robust pathway to finding approximate Nash equilibria in complex systems.

In the case where the mean-field population distribution parameter $\mu$ is fixed, the algorithm `Exact TRPO`$(\mu)$ provides a robust approximation to the value function. By iteratively updating the policy using trust region optimization techniques, the algorithm ensures convergence rates that explicitly depend on the regularization parameter $\eta$. Given a fixed $\eta$, the learning rate $1/(\eta(\ell+2))$ is optimally chosen to balance stability and efficiency in the policy updates. The following result establishes the error bounds for the value function approximation, highlighting the role of entropic regularization in convergence guarantees. Specifically, the bounds quantify the discrepancy between the value function induced by the computed policy and the optimal value function for the given mean-field population profile.

**Theorem C.1** (Theorem 16 in Shani et al. (2020))**.** *Fix* $\mu \in \mathcal{P}(\mathcal{S})$ *the initial distribution. Let* $\{\pi_\ell\}_{\ell=0}^L$ *be the sequence generated by the* `Exact TRPO`$(\mu)$ *algorithm. Then, there exists a constant* $C'_{\mathrm{TRPO},0} > 0$ *such that*

$$J^{\mathrm{MFG}}(\pi_\mu, \mu, \mu) - J^{\mathrm{MFG}}(\pi_L, \mu, \mu) \leq C'_{\mathrm{TRPO},0} \frac{\left(\|\mathsf{r}\|_\infty + \eta^2 \log^2 |\mathcal{A}|\right)}{\eta(1-\gamma)^3} \cdot \frac{\log L}{L} . \tag{12}$$

**Corollary C.2.** *Let* $\{\pi_\ell\}_{\ell=0}^L$ *be the sequence generated by the* `Exact TRPO`$(\mu)$ *algorithm. Then, we have that*

$$\sum_{s \in \mathcal{S}} \|\hat{\pi}_L(\cdot|s) - \pi_\mu(\cdot|s)\|_{\mathrm{TV}}^2 \, \mu(s) \leq C_{\mathrm{TRPO},0} \cdot \frac{\log L}{L} , \tag{13}$$

*with*

$$C_{\mathrm{TRPO},0} := C'_{\mathrm{TRPO},0} \frac{2}{\eta(1-\gamma)} \cdot \frac{\left(\|\mathsf{r}\|_\infty + \eta^2 \log^2 |\mathcal{A}|\right)}{\eta(1-\gamma)^3} .$$

*Proof.* Using Proposition E.2, we use the relationship between the total variation distance of a policy $\pi$ to the optimal policy $\pi_\mu$ and the corresponding difference in their value functions obtainine

$$\sum_{s \in \mathcal{S}} \|\hat{\pi}_L(\cdot|s) - \pi_\mu(\cdot|s)\|_{\mathrm{TV}}^2 \, \mu(s) \leq \frac{2}{\eta(1-\gamma)} \left( J^{\mathrm{MFG}}(\pi_\mu, \mu, \mu) - \frac{1}{L+1} \sum_{\ell=0}^L J^{\mathrm{MFG}}(\pi_\ell, \mu, \mu) \right)$$

Therefore, using (12), we get (13). $\qquad \square$

### C.2. Exact algorithm

We now analyze Algorithm 5, where no approximation is made. This algorithm is exact and does not involve any approximation. We provide a convergence result for this algorithm in the tabular setting.

---

**Algorithm 5** `ExactAlgo`

---

1: **Input:** $M$.
2: **Initialize:** $\mu_0$.
3: **for** $k \in [K]$ **do**
4:     $\mu_k := \mu_{k-1} + \beta_k \left( \mu_{k-1} \left( \mathsf{P}_{\mu_{k-1}}^{\pi_{\mu_{k-1}}} \right)^M - \mu_{k-1} \right).$       # Update population distribution
5: **end for**
6: **Output:** $\mu_K$.

---

**Proposition C.3.** *Suppose that Assumptions 1, 2, and 3 hold. Assume that, for any* $k \geq 0$,

$$\beta_k < \frac{\tau}{\tau - C_{\mathrm{op,MFG}} + C_{\pi,\mu}^2 C_{\mathrm{Erg,M}}^2} , \tag{14}$$

*with $\tau := 1 - C_{\mathrm{op,MFG}}$. Then, the sequence $\{\mu_k\}_{k \geq 0}$ defined in* `ExactAlgo` *satisfies*

$$\|\mu_k - \mu_\star\|_2^2 \leq (1 - \tau\beta_k) \|\mu_{k-1} - \mu_\star\|_2^2 . \tag{15}$$

*Moreover, if the step-sizes $\beta_k$ satisfy*

$$\sum_{k=0}^{\infty} \beta_k = \infty , \tag{16}$$

*we have that the exact algorithm* `ExactAlgo` *converges to the optimal policy in the tabular setting.*

*Proof.* We focus on the convergence of the sequence $\mu_k$ toward $\mu_\star$. Recall that $\mu_\star$ is the fixed point (8). From this condition, we then have that

$$\mu_\star = \mu_\star \mathsf{P}_{\mu_\star}^{\pi_{\mu_\star}} = \mu_\star \left(\mathsf{P}_{\mu_\star}^{\pi_{\mu_\star}}\right)^M .$$

We then have that

$$\|\mu_k - \mu_\star\|_2^2 = \left\|\mu_{k-1} - \mu_\star + \beta_k \left\{\mu_{k-1}\left(\mathsf{P}_{\mu_{k-1}}^{\pi_{\mu_{k-1}}}\right)^M - \mu_{k-1}\right\}\right\|_2^2$$

$$= \left\|\mu_{k-1} - \mu_\star + \beta_k \left\{\left(\mu_{k-1}\left(\mathsf{P}_{\mu_{k-1}}^{\pi_{\mu_{k-1}}}\right)^M - \mu_{k-1}\right) - \left(\mu_\star\left(\mathsf{P}_{\mu_\star}^{\pi_{\mu_\star}}\right)^M - \mu_\star\right)\right\}\right\|_2^2$$

$$= \left\|(1 - \beta_k)(\mu_{k-1} - \mu_\star) + \beta_k \left\{\mu_{k-1}\left(\mathsf{P}_{\mu_{k-1}}^{\pi_{\mu_{k-1}}}\right)^M - \mu_\star\left(\mathsf{P}_{\mu_\star}^{\pi_{\mu_\star}}\right)^M\right\}\right\|_2^2$$

$$= (1 - \beta_k)^2 \|\mu_{k-1} - \mu_\star\|_2^2 + 2(1 - \beta_k)\beta_k \left\langle \mu_{k-1} - \mu_\star, \mu_{k-1}\left(\mathsf{P}_{\mu_{k-1}}^{\pi_{\mu_{k-1}}}\right)^M - \mu_\star\left(\mathsf{P}_{\mu_\star}^{\pi_{\mu_\star}}\right)^M\right\rangle$$

$$+ \beta_k^2 \left\|\mu_{k-1}\left(\mathsf{P}_{\mu_{k-1}}^{\pi_{\mu_{k-1}}}\right)^M - \mu_\star\left(\mathsf{P}_{\mu_\star}^{\pi_{\mu_\star}}\right)^M\right\|_2^2 .$$

Applying Assumptions 2, and Corollary E.4, together with Lemma E.1, the previous equality implies that

$$\|\mu_k - \mu_\star\|_2^2 \leq \left[(1 - \beta_k)(1 + (2C_{\mathrm{op,MFG}} - 1)\beta_k) + \beta_k^2 C_{\pi,\mu}^2 C_{\mathrm{Erg,M}}^2\right] \|\mu_{k-1} - \mu_\star\|_2^2 .$$

Since $\beta_k$ satisfy (14), we then obtain 15. We see that 15 is a contraction inequality. Combining this with (16), it implies that the sequence $\mu_k$ converges to $\mu_\star$ exponentially fast, *i.e.*,

$$\|\mu_k - \mu_\star\|_2^2 \leq \prod_{j=1}^{k}(1 - \tau\beta_j) \|\mu_0 - \mu_\star\|_2^2 \leq \exp\left(-\tau \sum_{j=1}^{k}\beta_j\right) \|\mu_0 - \mu_\star\|_2^2 .$$

The rate of convergence is determined by the step-size $\beta_k$. This concludes the proof. $\qquad\square$

*Remark* C.4. The exact algorithm `ExactAlgo` is a simplified version of the algorithm we consider in this paper. The exact algorithm does not involve any approximation, thus is deterministic. This convergence is in line with deterinistic optimization. In fact, to get a precision of $\varepsilon$, we need $k$ to be of order

- $\log(\|\mu_0 - \mu_\star\|_2^2 / \varepsilon)(1 - 2C_{\mathrm{op,MFG}} + C_{\pi,\mu}^2 C_{\mathrm{Erg,M}}^2)$, if $\beta_k$ is constant equal to $\gamma > 0$ such that (14) is verified;

- $\|\mu_0 - \mu_\star\|_2^2 / \varepsilon \, \exp(1 - 2C_{\mathrm{op,MFG}} + C_{\pi,\mu}^2 C_{\mathrm{Erg,M}}^2)$, if $\beta_k = C/k$ for a certain $C > 0$ such that (14) is verified.

We now analyze the convergence of the algorithm with approximation, *i.e.*, Algorithm 2. We consider the following algorithm.

**Theorem C.5.** *Suppose that Assumptions 1 and 2 hold. Assume that, for any $k \geq 0$,*

$$\beta_k < b_0 := \frac{\tau}{4C_{\pi,\mu}^2 C_{\mathrm{Erg,M}}^2 + 2\tau} , \qquad for \, k \geq 1 , \tag{17}$$

*Then, the exact algorithm* `Exact MF-TRPO` *converges to the optimal policy in the tabular setting. In particular, we have that*

$$\|\mu_k - \mu_\star\|_2^2 \le \exp\left(-\frac{\tau}{2}\sum_{j=1}^{k}\beta_j\right)\|\mu_0 - \mu_\star\|_2^2 + \frac{2C_{\text{MF},0}}{\tau}\cdot\frac{\log(L)}{L}\ , \tag{18}$$

*with*

$$\tau := 1 - C_{\text{op,MFG}}\ ,$$
$$C_{\text{MF},0} := \frac{2+b_0}{\tau}\cdot C_{\text{Erg,M}}C_{\text{TRPO},0}\ .$$

*Proof.* We focus on the convergence of the sequence $\mu_k \to \mu_\star$, with $\mu_\star$ as in (8). Denote $\pi_k$ the output of `Exact TRPO`$(\mu_k)$ at each step. We then have that

$$
\begin{aligned}
\|\mu_k - \mu_\star\|_2^2 &= \left\|\mu_{k-1} - \mu_\star + \beta_k\left\{\mu_{k-1}\left(\mathsf{P}_{\mu_{k-1}}^{\pi_k}\right)^M - \mu_{k-1}\right\}\right\|_2^2 \\
&= \left\|\mu_{k-1} - \mu_\star + \beta_k\left(\mu_{k-1}\left(\mathsf{P}_{\mu_{k-1}}^{\pi_k}\right)^M - \mu_{k-1}\left(\mathsf{P}_{\mu_{k-1}}^{\pi_{\mu_{k-1}}}\right)^M\right) + \right. \\
&\qquad\qquad \left. +\beta_k\left(\mu_{k-1}\left(\mathsf{P}_{\mu_{k-1}}^{\pi_{\mu_{k-1}}}\right)^M - \mu_{k-1}\right)\right\|_2^2 \\
&= \left\|\mu_{k-1} - \mu_\star + \beta_k\left(\mu_{k-1}\left(\mathsf{P}_{\mu_{k-1}}^{\pi_k}\right)^M - \mu_{k-1}\left(\mathsf{P}_{\mu_{k-1}}^{\pi_{\mu_{k-1}}}\right)^M\right)\right. \\
&\qquad\qquad \left. +\beta_k\left(\mu_{k-1}\left(\mathsf{P}_{\mu_{k-1}}^{\pi_{\mu_{k-1}}}\right)^M - \mu_{k-1}\right) - \beta_k\left(\mu_\star\left(\mathsf{P}_{\mu_\star}^{\pi_{\mu_\star}}\right)^M - \mu_\star\right)\right\|_2^2 \\
&= \left\|(1-\beta_k)(\mu_{k-1} - \mu_\star) + \beta_k\left(\mu_{k-1}\left(\mathsf{P}_{\mu_{k-1}}^{\pi_k}\right)^M - \mu_{k-1}\left(\mathsf{P}_{\mu_{k-1}}^{\pi_{\mu_{k-1}}}\right)^M\right)\right. \\
&\qquad\qquad \left. +\beta_k\left(\mu_{k-1}\left(\mathsf{P}_{\mu_{k-1}}^{\pi_{\mu_{k-1}}}\right)^M - \mu_\star\left(\mathsf{P}_{\mu_\star}^{\pi_{\mu_\star}}\right)^M\right)\right\|_2^2 \\
&= (1-\beta_k)^2\|\mu_{k-1} - \mu_\star\|_2^2 + 2(1-\beta_k)\beta_k\left\langle\mu_{k-1} - \mu_\star, \mu_{k-1}\left(\mathsf{P}_{\mu_{k-1}}^{\pi_{\mu_{k-1}}}\right)^M - \mu_\star\left(\mathsf{P}_{\mu_\star}^{\pi_{\mu_\star}}\right)^M\right\rangle \\
&\quad + \beta_k^2\left\|\mu_{k-1}\left(\mathsf{P}_{\mu_{k-1}}^{\pi_{\mu_{k-1}}}\right)^M - \mu_\star\left(\mathsf{P}_{\mu_\star}^{\pi_{\mu_\star}}\right)^M\right\|_2^2 \\
&\quad + 2(1-\beta_k)\beta_k\left\langle\mu_{k-1} - \mu_\star, \mu_{k-1}\left(\mathsf{P}_{\mu_{k-1}}^{\pi_k}\right)^M - \mu_{k-1}\left(\mathsf{P}_{\mu_{k-1}}^{\pi_{\mu_{k-1}}}\right)^M\right\rangle \\
&\quad + \beta_k^2\left\|\mu_{k-1}\left(\mathsf{P}_{\mu_{k-1}}^{\pi_k}\right)^M - \mu_{k-1}\left(\mathsf{P}_{\mu_{k-1}}^{\pi_{\mu_{k-1}}}\right)^M\right\|_2^2 \\
&\quad + 2\beta_k^2\left\langle\mu_{k-1}\left(\mathsf{P}_{\mu_{k-1}}^{\pi_k}\right)^M - \mu_{k-1}\left(\mathsf{P}_{\mu_{k-1}}^{\pi_{\mu_{k-1}}}\right)^M, \mu_{k-1}\left(\mathsf{P}_{\mu_{k-1}}^{\pi_{\mu_{k-1}}}\right)^M - \mu_\star\left(\mathsf{P}_{\mu_\star}^{\pi_{\mu_\star}}\right)^M\right\rangle\ .
\end{aligned}
$$

Applying Assumptions 2 and Corollary E.4 together with Lemma E.1, and following the same lines as in the proof

of Proposition C.3, the previous equality implies that

$$
\begin{aligned}
\|\mu_k - \mu_\star\|_2^2 &\leq \left[(1-\beta_k)\left(1+(1-2\tau)\beta_k\right) + \beta_k^2 C_{\pi,\mu}^2 C_{\mathrm{Erg,M}}^2\right] \|\mu_{k-1} - \mu_\star\|_2^2 \\
&\quad + 2(1-\beta_k)\beta_k \left\langle \mu_{k-1} - \mu_\star, \mu_{k-1}\left(\mathsf{P}_{\mu_{k-1}}^{\pi_k}\right)^M - \mu_{k-1}\left(\mathsf{P}_{\mu_{k-1}}^{\pi_{\mu_{k-1}}}\right)^M \right\rangle \\
&\quad + \beta_k^2 \left\| \mu_{k-1}\left(\mathsf{P}_{\mu_{k-1}}^{\pi_k}\right)^M - \mu_{k-1}\left(\mathsf{P}_{\mu_{k-1}}^{\pi_{\mu_{k-1}}}\right)^M \right\|_2^2 \\
&\quad + 2\beta_k^2 \left\langle \mu_{k-1}\left(\mathsf{P}_{\mu_{k-1}}^{\pi_k}\right)^M - \mu_{k-1}\left(\mathsf{P}_{\mu_{k-1}}^{\pi_{\mu_{k-1}}}\right)^M, \mu_{k-1}\left(\mathsf{P}_{\mu_{k-1}}^{\pi_{\mu_{k-1}}}\right)^M - \mu_\star\left(\mathsf{P}_{\mu_\star}^{\pi_{\mu_\star}}\right)^M \right\rangle \\
&=: \left[(1-\beta_k)\left(1+(1-2\tau)\beta_k\right) + \beta_k^2 C_{\pi,\mu}^2 C_{\mathrm{Erg,M}}^2\right] \|\mu_{k-1} - \mu_\star\|_2^2 \\
&\quad + 2\beta_k(1-\beta_k)\mathbf{T}_1 + \beta_k^2 \mathbf{T}_2 + 2\beta_k^2 \mathbf{T}_3 \ .
\end{aligned}
$$

We now proceed in studying the terms $\mathbf{T}_1$, $\mathbf{T}_2$, and $\mathbf{T}_3$. Using Young's inequality, we get that

$$
\begin{aligned}
|\mathbf{T}_1| &\leq \frac{\tau}{2} \|\mu_{k-1} - \mu_\star\|_2^2 + \frac{1}{2\tau} \left\| \mu_{k-1}\left(\mathsf{P}_{\mu_{k-1}}^{\pi_k}\right)^M - \mu_{k-1}\left(\mathsf{P}_{\mu_{k-1}}^{\pi_{\mu_{k-1}}}\right)^M \right\|_2^2 \\
&\leq \frac{\tau}{2} \|\mu_{k-1} - \mu_\star\|_2^2 + \frac{1}{2\tau} \mathbf{T}_2 \ ,
\end{aligned}
$$

and, using Lemma E.1 and Corollary E.4,

$$
\begin{aligned}
|\mathbf{T}_3| &\leq \frac{1}{2} \left\| \mu_{k-1}\left(\mathsf{P}_{\mu_{k-1}}^{\pi_{\mu_{k-1}}}\right)^M - \mu_\star\left(\mathsf{P}_{\mu_\star}^{\pi_{\mu_\star}}\right)^M \right\|_2^2 + \frac{1}{2} \left\| \mu_{k-1}\left(\mathsf{P}_{\mu_{k-1}}^{\pi_k}\right)^M - \mu_{k-1}\left(\mathsf{P}_{\mu_{k-1}}^{\pi_{\mu_{k-1}}}\right)^M \right\|_2^2 \\
&\leq \frac{1}{2} C_{\pi,\mu}^2 C_{\mathrm{Erg,M}}^2 \|\mu_{k-1} - \mu_\star\|_2^2 + \frac{1}{2} \mathbf{T}_2 \ .
\end{aligned}
$$

Since $\tau < 1$ from Assumption 2 and $\beta_k$ satisfies (17), a straightforward computation shows that

$$
(1-\beta_k)\left(1+(1-\tau)\beta_k\right) + 2\beta_k^2 C_{\pi,\mu}^2 C_{\mathrm{Erg,M}}^2 - \frac{\tau}{2}\beta_k \leq \left(1 - \frac{\tau}{2}\beta_k\right) \ .
$$

Moreover, applying Lemma E.1, together with Theorem C.1 on the performances of TRPO, we have

$$
\begin{aligned}
\mathbf{T}_2 &\leq C_{\mathrm{Erg,M}} \left( \sum_{s\in\mathcal{S}} \mu_{k-1}^2(s) \|\pi_k(\cdot|s) - \pi_\mu(\cdot|s)\|_{\mathrm{TV}}^2 \right) \\
&\leq C_{\mathrm{Erg,M}} \left( \sum_{s\in\mathcal{S}} \mu_{k-1}(s) \|\pi_k(\cdot|s) - \pi_{\mu_{k-1}}(\cdot|s)\|_{\mathrm{TV}}^2 \right) \\
&\leq C_{\mathrm{Erg,M}} \left( J^{\mathrm{MFG}}(\pi_{\mu_{k-1}}, \mu_{k-1}, \mu_{k-1}) - J^{\mathrm{MFG}}(\pi_k, \mu_{k-1}, \mu_{k-1}) \right) \leq C_{\mathrm{Erg,M}} \, 2C_{\mathrm{TRPO,0}} \frac{\log(L)}{L} \ .
\end{aligned}
$$

Therefore, combining the previous inequalities, we have that

$$
\begin{aligned}
\|\mu_k - \mu_\star\|_2^2 &\leq \left(1 - \frac{\tau}{2}\beta_k\right) \|\mu_{k-1} - \mu_\star\|_2^2 + \beta_k \left(\frac{1-\beta_k}{\tau} + \frac{3\beta_k}{2}\right) \cdot 2C_{\mathrm{Erg,M}} C_{\mathrm{TRPO,0}} \frac{\log(L)}{L} \\
&\leq \left(1 - \frac{\tau}{2}\beta_k\right) \|\mu_{k-1} - \mu_\star\|_2^2 + \beta_k \frac{2+b_0}{\tau} \cdot C_{\mathrm{Erg,M}} C_{\mathrm{TRPO,0}} \frac{\log(L)}{L} \qquad (19)\\
&\leq \left(1 - \frac{\tau}{2}\beta_k\right) \|\mu_{k-1} - \mu_\star\|_2^2 + \beta_k C_{\mathrm{MF,0}} \frac{\log(L)}{L} \ .
\end{aligned}
$$

Developing the recursion (19), we obtain

$$
\begin{aligned}
\|\mu_k - \mu_\star\|_2^2 &\leq \prod_{j=1}^k \left(1 - \frac{\tau}{2}\beta_j\right) \|\mu_0 - \mu_\star\|_2^2 + C_{\mathrm{MF,0}} \sum_{j=1}^k \frac{\log(L)}{L} \beta_j \prod_{\ell=j+1}^k \left(1 - \frac{\tau}{2}\beta_\ell\right) \\
&\leq \exp\left(-\frac{\tau}{2} \sum_{j=1}^k \beta_j\right) \|\mu_0 - \mu_\star\|_2^2 + C_{\mathrm{MF,0}} \frac{\log(L)}{L} \sum_{j=1}^k \beta_j \prod_{\ell=j+1}^k \left(1 - \frac{\tau}{2}\beta_\ell\right) \ .
\end{aligned}
$$

Note that the second term of the r.h.s. of the previous inequality is a telescopic sum, as the central term can be rewritten as

$$\beta_j \prod_{\ell=j+1}^{k} \left(1 - \frac{\tau}{2}\beta_\ell\right) = \frac{2}{\tau}\left[\prod_{\ell=j+1}^{k} \left(1 - \frac{\tau}{2}\beta_\ell\right) - \prod_{\ell=j}^{k} \left(1 - \frac{\tau}{2}\beta_\ell\right)\right] .$$

Therefore, we get (18). $\qquad \square$

$\varepsilon$-**MFNE.** With the theoretical foundations established in Theorem C.1 and Theorem C.5, we can now derive a result on the closeness of the proposed algorithm to the MFNE. Specifically, we show that `Exact MF-TRPO` achieves an $\varepsilon$-MFNE, where the approximation error $\varepsilon$ is explicitly quantified as follows.

**Corollary C.6.** *Suppose that Assumptions 1, 2, and 3 hold. Assume that, for any $k \geq 0$, the learning rate $\beta_k$ satisfies (17). Let $\mu_k$ (resp. $\pi_{k+1}$) the output of* `Exact MF-TRPO` *(resp.* `Exact TRPO`$(\mu_k)$*). Then, $(\pi_{k+1}, \mu_k)$ is $\varepsilon_k$-MFNE, with*

$$\varepsilon_k := \delta_{NE,1,k} + C_\phi\left(\left(1 + C_{\mathrm{Erg},\infty}\left(1 + C_{\pi,\mu}\right)\right)\sqrt{\delta_{NE,2,k}} + \frac{2C_{\mathrm{Erg},\infty}\sqrt{|\mathcal{S}|}}{\eta(1-\gamma)} \cdot \sqrt{\delta_{NE,1,k}}\right) ,$$

$$\delta_{NE,1,k} = C'_{\mathrm{TRPO},0}\frac{\left(\|\mathsf{r}\|_\infty + \eta^2\log^2|\mathcal{A}|\right)}{\eta(1-\gamma)^3} \cdot \frac{\log L}{L} ,$$

$$\delta_{NE,2,k} = \exp\left(-\frac{\tau}{2}\sum_{j=1}^{k}\beta_j\right)\|\mu_0 - \mu_\star\|_2 + \frac{2C_{\mathrm{MF},0}}{\tau} \cdot \frac{\log(L)}{L} .$$

*Proof.* From Proposition E.5, we have that the exploitability of a policy $\pi$ and a mean-field parameter $\mu$ can be bound by the gap of optimality of the $\pi$ w.r.t. the value function $J^{\mathrm{MFG}}(\cdot, \mu, \mu)$ and the distance between $\mu$ and the stationary distribution $\lambda_{\pi,\mu}$.

From Theorem C.1, we have that

$$\max_{\pi \in \Pi} J^{\mathrm{MFG}}(\pi, \mu_k, \mu_k) - J^{\mathrm{MFG}}(\pi_k, \mu_k, \mu_k) \leq \delta_{\mathrm{NE},1,k} . \tag{20}$$

On the other hand, using the fact that $(\pi_{\mu_\star}, \mu_\star)$ is a MFNE, we have

$$\mu_k - \lambda_{\pi_{k+1},\mu_k} = (\mu_k - \mu_\star) + \left(\lambda_{\pi_{\mu_\star},\mu_\star} - \lambda_{\pi_{\mu_k},\mu_k}\right) + \left(\lambda_{\pi_{\mu_k},\mu_k} - \lambda_{\pi_{k+1},\mu_k}\right) .$$

Then, applying Lemma E.1, together with Corollary E.4, we obtain

$$\left\|\lambda_{\pi_{\mu_\star},\mu_\star} - \lambda_{\pi_{\mu_k},\mu_k}\right\|_2 \leq C_{\mathrm{Erg},\infty}\left(1 + C_{\pi,\mu}\right)\|\mu_\star - \mu_k\|_2 .$$

Moreover, from Theorem C.1 on the performances of `Exact TRPO`, together with Lemma E.1 and Proposition E.2, we have

$$\left\|\lambda_{\pi_{\mu_k},\mu_k} - \lambda_{\pi_{k+1},\mu_k}\right\|_2 \leq C_{\mathrm{Erg},\infty}\sum_{s \in \mathcal{S}}\mu_k(s)\|\pi_{k+1}(\cdot|s) - \pi_{\mu_k}(\cdot|s)\|_{\mathrm{TV}}$$

$$\leq \frac{2C_{\mathrm{Erg},\infty}\sqrt{|\mathcal{S}|}}{\eta(1-\gamma)}\sqrt{J^{\mathrm{MFG}}(\pi_{\mu_k}, \mu_k, \mu_k) - J^{\mathrm{MFG}}(\pi_{k+1}, \mu_k, \mu_k)} \leq \frac{2C_{\mathrm{Erg},\infty}\sqrt{|\mathcal{S}|}}{\eta(1-\gamma)} \cdot \sqrt{\delta_{\mathrm{NE},1,k}} .$$

Using the triangle inequality, together with Theorem C.5, we can bound $\left\|\mu_k - \lambda_{\pi_{k+1},\mu_k}\right\|_2$ as

$$\left\|\mu_k - \lambda_{\pi_{k+1},\mu_k}\right\|_2 \leq \left(1 + C_{\mathrm{Erg},\infty}\left(1 + C_{\pi,\mu}\right)\right)\|\mu_\star - \mu_k\|_2 + \frac{2C_{\mathrm{Erg},\infty}\sqrt{|\mathcal{S}|}}{\eta(1-\gamma)} \cdot \sqrt{\delta_{\mathrm{NE},1,k}}$$

$$\leq \left(1 + C_{\mathrm{Erg},\infty}\left(1 + C_{\pi,\mu}\right)\right)\sqrt{\delta_{\mathrm{NE},2,k}} + \frac{2C_{\mathrm{Erg},\infty}\sqrt{|\mathcal{S}|}}{\eta(1-\gamma)} \cdot \sqrt{\delta_{\mathrm{NE},1,k}} .$$

Using the last inequality and (20), together with Proposition E.5, we have that

$$\phi(\pi_{k+1}, \mu_k) \leq \delta_{\mathrm{NE},1,k} + C_\phi \left( \left(1 + C_{\mathrm{Erg},\infty}\left(1 + C_{\pi,\mu}\right)\right) \sqrt{\delta_{\mathrm{NE},2,k}} + \frac{2C_{\mathrm{Erg},\infty}\sqrt{|\mathcal{S}|}}{\eta(1-\gamma)} \cdot \sqrt{\delta_{\mathrm{NE},1,k}} \right) = \varepsilon_k \ .$$

Therefore, $(\mu_k, \pi_{k+1})$ is $\varepsilon_k$-MFNE as defined in Definition (2.1). $\qquad\square$

*Remark* C.7. From this corollary, it follows directly that to achieve an $\varepsilon$-MFNE, the required number of inner policy updates $L$ and outer population updates $K$ must satisfy the following scaling conditions:

$$L \in \widetilde{O}\left(1/\varepsilon^2\right) \quad \text{and} \quad K \in \widetilde{O}\left(\log(1/\varepsilon)\right).$$

This implies that the sample complexity of the proposed algorithm scales polynomially in $1/\varepsilon$ with respect to the inner optimization steps and logarithmically with respect to the outer mean-field updates. This confirms the efficiency of our approach, ensuring that even for small values of $\varepsilon$, convergence to an approximate MFNE remains computationally feasible.

# D. Model free algorithms

Model-free approaches play a fundamental role in developing model-agnostic algorithms capable of autonomously adapting to diverse and evolving environments. In the MFG context, various models have been proposed across different domains (Perrin et al., 2021; Yardim et al., 2023), and recent efforts have explored data-driven methodologies to enhance their applicability. Following this line of research, we introduce `Sample-Based MF-TRPO`, a model-free approach tailored for MFG problems. By leveraging RL techniques with scalable sample-based updates, our method contributes to the growing body of work on data-driven MFG solutions, providing finite-sample complexity guarantees in this setting. This framework further aligns MFG with model-free learning paradigms, broadening their potential for real-world deployment in complex decision-making environments.

## D.1. TRPO - sample-based formulation

In the framework established by Shani et al. (2020), it is important to note that the sample-based algorithm does not provide a last-iterate sample complexity guarantee. This contrasts with the exact algorithm, where the policy improvement lemma (Lemma 15, Shani et al., 2020) serves as a foundation for analyzing the convergence properties of the last iterate. In the exact update setting, this guarantee is analogous to Howard's lemma (Howard, 1960). However, the presence of sampling errors in the sample-based setting hinders the attainment of such guarantees, necessitating a more refined approach when designing and analyzing RL algorithms for MFG.

To address this limitation, it becomes essential to consider alternative strategies than Theorem 5 in Shani et al. (2020). This theorem, however, still provides a framework for analyzing the uniform mixture of the policies generated during the iterative procedure, rather than relying solely on the last iterate. By shifting focus to such policy, we can generalize the theoretical guarantees of the algorithm—a property inherent to the MFG setting.

Additionally, the connection between the value function and the policy space plays a crucial role. Proposition E.2 ensures that the gap in value functions directly bounds the differences between policies. This property provides a pathway to refine the policy improvement process and derive meaningful finite-sample complexity guarantees. By combining these insights, we can propose a robust methodology where the sample-based algorithm achieves convergence with high probability, utilizing uniform mixture policies to overcome the challenges posed by the lack of last-iterate guarantees.

Overall, this refinement introduces a smarter utilization of the sample-based algorithm, emphasizing the role of averaging in mitigating the variability and uncertainty inherent in sample-based methods. This approach not only aligns with the theoretical underpinnings of convex optimization but also strengthens the practical applicability of RL algorithms in MFG, delivering finite-sample complexity results with rigorous probabilistic guarantees.

*Remark* D.1. For a fixed $\mu$, the output of `Sample-Based TRPO`$(\mu)$ is the uniform mixture policy $\hat{\pi}_L^{\mathrm{Unif},\mu}$. This policy is such that, in the unregularized case, we have (11). It consists on a mixture of $\hat{\pi}_\ell$, for $\ell = 0, \ldots, L$. The following dedicated subroutine achieve the sampling process in a computationally efficient manner, without performing a direct mixture at every decision step.

---

**Algorithm 6** `Sample-Based TRPO`$(\mu)$

---

1: **Initialize:** $\pi_0(\cdot|s) = \mathcal{U}(\mathcal{A})$ for any $s \in \mathcal{S}$.
2: **Input:** $\epsilon, \delta > 0$, $L$.
3: **for** $\ell \in [L]$ **do**
4:      $S_\ell^{I_\ell} = \{\}, \forall s, a, Q_{\pi_\ell, \mu}(s, a) = 0, n_\ell(s, a) = 0$
5:      $I_\ell \geq \frac{|\mathcal{A}|^2\big(\|\mathsf{r}\|_\infty + \eta^2 \log^2 |\mathcal{A}|\big)\big(|\mathcal{S}| \log 2|\mathcal{A}| + \log \frac{1}{\delta}\big)}{(1-\gamma)^2 \epsilon^2}$                # Sample Trajectories
6:      $T_\ell \geq \frac{1}{1-\gamma} \log \left( \frac{\epsilon}{|\mathcal{A}|\big(\|\mathsf{r}\|_\infty + \eta \log |\mathcal{A}|\big)} \right)$                          # Rollout horizon
7:      **for** $p = 1, \ldots, I_\ell$ **do**
8:          Sample $s_i \sim \bar{\mathsf{d}}_{\nu, \mu}^{\pi_\ell}(\cdot)$, $a_i \sim \mathcal{U}(\mathcal{A})$
9:          $Q_{\hat{\pi}_\ell, \mu}(s_i, a_i, i) \leftarrow \mathsf{r}(s_i, a_i, \mu) + \sum_{t=1}^{T_\ell} \gamma^t \mathbb{E}_{s_t \sim \delta_{s_i}(\mathsf{P}_\mu^{\hat{\pi}_\ell})^t, a_t \sim \hat{\pi}_\ell(\cdot|s_t)} \big[ \mathsf{r}(s_t, a_t, \mu) + \eta \log\big(\hat{\pi}_\ell(a_t|s_t)\big) \big]$
                                                                           # Truncated rollout
10:         $Q_{\pi_\ell, \mu}(s_i, a_i) \leftarrow Q_{\pi_\ell, \mu}(s_i, a_i) + Q_{\pi_\ell, \mu}(s_i, a_i, i)$
11:        $n_\ell(s_i, a_i) \leftarrow n_\ell(s_i, a_i) + 1$
12:        $S_\ell^{I_\ell} = S_\ell^{I_\ell} \cup \{s_i\}$
13:      **end for**
14:      **for** $s \in S_\ell^{I_\ell}$ **do**
15:        **for** $a \in \mathcal{A}$ **do**
16:          $Q_{\hat{\pi}_\ell, \mu}(s, a) \leftarrow \frac{|\mathcal{A}| Q_{\hat{\pi}_\ell, \mu}(s, a)}{\sum_{a' \in \mathcal{A}} n_\ell(s, a')}$
17:        **end for**
18:        $\hat{\pi}_{\ell+1}(a|s) \leftarrow \frac{\hat{\pi}_\ell(a|s) \exp\big(\frac{1}{\eta(\ell+2)}\big(Q_{\hat{\pi}_\ell, \mu}(s, a) - \eta \log \hat{\pi}_\ell(a|s)\big)\big)}{\sum_{a' \in \mathcal{A}} \hat{\pi}_\ell(a'|s) \exp\big(\frac{1}{\eta(\ell+2)}\big(Q_{\hat{\pi}_\ell, \mu}(s, a') - \eta \log \hat{\pi}_\ell(a'|s)\big)\big)}$
19:      **end for**
20: **end for**
21: **Output:** $\hat{\pi}_L^{\mathtt{Unif}, \mu}$.

---

**Algorithm 7** `Uniform-Mixture`$(\{\hat{\pi}_\ell\}_{\ell=0,\ldots,L})$

---

1: **Input:** $\{\hat{\pi}_\ell\}_{\ell=0,\ldots,L}$.
2: Draw a random variable $\hat{\ell} \sim \mathcal{U}(\{0, 1, \ldots, L\})$.
3: **Output:** $\hat{\pi}_{\hat{\ell}}$

---

This policy is defined is to sample from this policy efficiently without explicitly computing an arithmetic average at the sampling level, particularly in its use within the inner loop of `Sample-Based TRPO`$(\mu)$. Moroever, given that the number of iterations $L$ is fixed beforehand, the procedure begins by drawing a random variable $\hat{\ell}$ uniformly from the set $\{0, 1, \ldots, L\}$. Once $\hat{\ell}$ is selected, the sampling step follows the policy $\hat{\pi}_{\hat{\ell}}$. This approach ensures that the selected action is drawn a policy $\hat{\pi}_L^{\mathtt{Unif}, \mu}$ without incurring unnecessary computational overhead during execution.

In particular, due to the regularization term, we have that the following inequality holds:

$$\frac{1}{L+1} \sum_{\ell=0}^{L} J^{\mathrm{MFG}}(\hat{\pi}_\ell, \mu, \mu) \leq J^{\mathrm{MFG}}(\hat{\pi}_L^{\mathtt{Unif}, \mu}, \mu, \mu) \,. \tag{21}$$

In the absence of regularization, the objective is linear in the occupancy measure, which allows for exact equalities when considering mixtures of policies. However, once the entropic regularization term is introduced, the objective becomes *concave* in the occupancy measure (see, e.g., Neu et al. (2017) for a proof). As a result, we only obtain the previous inequality rather than (11) when averaging over iterates, as in the relation involving the mixture policy.

**Theorem D.2** (Based on Theorem 5 in Shani et al. (2020)). *Suppose Assumption 4 holds. Fix $\epsilon, \delta > 0$. Let $\{\hat{\pi}_\ell\}_{\ell \geq 0}$ be the*

*sequence generated by* `Sample-Based TRPO`$(\mu)$, *using*

$$I_\ell \geq \frac{|\mathcal{A}|^2 \left( \|\mathsf{r}\|_\infty^2 + \eta^2 \log^2 |\mathcal{A}| \right) \left( |\mathcal{S}| \log 2|\mathcal{A}| + \log \frac{1}{\delta} \right)}{(1-\gamma)^2 \epsilon^2}$$

*trajectories in each iteration and a rollout up to time* $T_\ell$ *with*

$$T_\ell \geq \frac{1}{1-\gamma} \log \left( \frac{\epsilon}{|\mathcal{A}| \left( \|\mathsf{r}\|_\infty + \eta \log |\mathcal{A}| \right)} \right) \ .$$

*Then, there exists* $C'_{\mathrm{TRPO},1} > 0$ *such that for all* $L \geq 1$*, the following holds with probability greater than* $1 - \delta$

$$
\begin{aligned}
& J^{\mathrm{MFG}} \left( \pi_\mu, \mu, \mu \right) - J^{\mathrm{MFG}} (\hat{\pi}_L^{Unif,\mu}, \mu, \mu) \\
& \leq J^{\mathrm{MFG}} \left( \pi_\mu, \mu, \mu \right) - \frac{1}{L+1} \sum_{\ell=0}^{L} J^{\mathrm{MFG}} \left( \hat{\pi}_\ell, \mu, \mu \right) \\
& \leq C'_{\mathrm{TRPO},1} \left( \frac{\left( \|\mathsf{r}\|_\infty^2 + \eta^2 \log^2 |\mathcal{A}| \right) |\mathcal{A}|^2 \log L}{\eta(1-\gamma)^3 (L+1)} + \frac{\epsilon}{(1-\gamma)^2} \left\| \frac{\overline{\mathsf{d}}_{\mu,\mu}^{\pi_\mu}}{\nu} \right\|_\infty \right) \ .
\end{aligned}
$$

(22)

*Proof.* The proof of this result is based on the proof of Shani et al. (Theorem 5, 2020). The main difference is that we are considering the uniform mixture of the policies generated during the iterative procedure.

Applying Shani et al. (Lemma 19, 2020), we get

$$
\begin{aligned}
& \frac{1-\gamma}{\eta(\ell+2)} \left( J^{\mathrm{MFG}} \left( \pi_\mu, \mu, \mu \right) - J^{\mathrm{MFG}} \left( \hat{\pi}_\ell, \mu, \mu \right) \right) \\
& \leq \overline{\mathsf{d}}_{\mu,\mu}^{\pi_\mu} \left( \left( 1 - \frac{1}{\ell+2} \right) \mathbf{D}_\Omega(\pi_\mu, \hat{\pi}_\ell) - \mathbf{D}_\Omega(\pi_\mu, \hat{\pi}_{\ell+1}) \right) + \frac{h^2(\ell)}{2\eta^2(\ell+2)^2} + \overline{\mathsf{d}}_{\mu,\mu}^{\pi_\mu} \epsilon_k \\
& \leq \overline{\mathsf{d}}_{\mu,\mu}^{\pi_\mu} \left( \frac{\ell+1}{\ell+2} \mathbf{D}_\Omega(\pi_\mu, \hat{\pi}_\ell) - \mathbf{D}_\Omega(\pi_\mu, \hat{\pi}_{\ell+1}) \right) + \frac{h^2(L)}{2\eta^2(\ell+2)^2} + \overline{\mathsf{d}}_{\mu,\mu}^{\pi_\mu} \epsilon_k \ ,
\end{aligned}
$$

with

$$
\begin{aligned}
\overline{\mathsf{d}}_{\mu,\mu}^{\pi_\mu} \epsilon_k = & \sum_{s \in \mathcal{S}} \frac{\overline{\mathsf{d}}_{\mu,\mu}^{\pi_\mu}(s)}{I_\ell \overline{\mathsf{d}}_{\nu,\mu}^{\pi_\mu}(s)} \sum_{i=1}^{I_\ell} \mathbb{1}_{\{s=s_i\}} \sum_{a \in \mathcal{A}} \Bigg( \frac{1}{\eta(\ell+2)} \big( |\mathcal{A}| Q_{\hat{\pi}_\ell,\mu}(s,a,i) - \eta \left( 1 + \log \hat{\pi}_\ell(a|s) \right) \big) \\
& - \log \hat{\pi}_{\ell+1}(a|s) + \log \hat{\pi}_\ell(a|s) \Bigg) \left( \hat{\pi}_{\ell+1}(a|s) - \pi_\mu(a|s) \right) \\
h(\ell) = & (1 + 8\eta) \frac{\|\mathsf{r}\|_\infty + \eta \log |\mathcal{A}|}{1-\gamma} \log \ell
\end{aligned}
$$

using that $h$ is a non-decreasing function. Multiplying both sides by $\eta(\ell+2)$, summing from $\ell = 0$ to $L$, and using the linearity of expectation, we get

$$
\begin{aligned}
& (1-\gamma) \sum_{\ell=0}^{L} \left( J^{\mathrm{MFG}} \left( \pi_\mu, \mu, \mu \right) - J^{\mathrm{MFG}} \left( \hat{\pi}_\ell, \mu, \mu \right) \right) \\
& \leq \overline{\mathsf{d}}_{\mu,\mu}^{\pi_\mu} \left( \mathbf{D}_\Omega(\pi_\mu, \pi_0) - (L+2)\mathbf{D}_\Omega(\pi_\mu, \pi_{L+1}) \right) + \sum_{\ell=0}^{L} \frac{h^2(L)}{2\eta(\ell+2)} + \sum_{\ell=0}^{L} \eta(\ell+2)\overline{\mathsf{d}}_{\mu,\mu}^{\pi_\mu} \epsilon_\ell \\
& \leq \overline{\mathsf{d}}_{\mu,\mu}^{\pi_\mu} \mathbf{D}_\Omega(\pi_\mu, \pi_0) + \sum_{\ell=0}^{L} \frac{h^2(L)}{2\eta(\ell+2)} + \sum_{\ell=0}^{L} \eta(\ell+2)\overline{\mathsf{d}}_{\mu,\mu}^{\pi_\mu} \epsilon_\ell \\
& \leq \log |\mathcal{A}| + \sum_{\ell=0}^{L} \frac{h^2(L)}{2\eta(\ell+2)} + \sum_{\ell=0}^{L} \eta(\ell+2)\overline{\mathsf{d}}_{\mu,\mu}^{\pi_\mu} \epsilon_\ell \ ,
\end{aligned}
$$

with the occupancy measure $\overline{\mathsf{d}}_{\mu,\mu}^{\pi_\mu}$ defined as (6), where the second relation holds by the positivity of the Bregman distance, and the third relation by Shani et al. (Lemma 28, 2020) for uniformly initialized $\pi_0$.

$$\sum_{\ell=0}^{L} \left( J^{\mathrm{MFG}}\left(\pi_\mu, \mu, \mu\right) - J^{\mathrm{MFG}}\left(\hat{\pi}_\ell, \mu, \mu\right) \right) \leq \frac{\log |\mathcal{A}|}{1-\gamma} + C'_{\mathrm{TRPO},1} \frac{h^2(L) \log L}{\eta(1-\gamma)} + \frac{1}{1-\gamma} \sum_{\ell=0}^{L} \eta(\ell+2) \overline{\mathsf{d}}_{\mu,\mu}^{\pi_\mu} \epsilon_\ell \ .$$

Dividing by $(L+1)$, we obtain

$$\frac{1}{L+1} \sum_{\ell=0}^{L} \left( J^{\mathrm{MFG}}\left(\pi_\mu, \mu, \mu\right) - J^{\mathrm{MFG}}\left(\hat{\pi}_\ell, \mu, \mu\right) \right)$$

$$\leq \frac{\log |\mathcal{A}|}{(1-\gamma)(L+1)} + C'_{\mathrm{TRPO},1} \frac{h^2(L) \log L}{\eta(1-\gamma)(L+1)} + \frac{1}{(1-\gamma)(L+1)} \sum_{\ell=0}^{L} \eta(\ell+2) \overline{\mathsf{d}}_{\mu,\mu}^{\pi_\mu} \epsilon_\ell \ .$$

Plugging in Shani et al. (Lemma 22 and Lemma 23, 2020), we get that for any $(\epsilon, \delta)$, if the number of trajectories in the $\ell$-th iteration satisfies

$$I_\ell \geq \frac{8|\mathcal{A}|^2 \left( \|\mathsf{r}\|_\infty^2 + \eta^2 \log^2 |\mathcal{A}| \right)}{\epsilon^2 (1-\gamma)^2} \left( |\mathcal{S}| \log 2|\mathcal{A}| + \log \frac{\pi^2(\ell+1)^2}{6\delta} \right) \ ,$$

and the rollout is performed up to time $T_\ell$ with

$$T_\ell \geq \frac{1}{1-\gamma} \log \left( \frac{\epsilon}{|\mathcal{A}| \left( \|\mathsf{r}\|_\infty + \eta \log |\mathcal{A}| \right)} \right) \ ,$$

then with probability at least $1 - \delta$,

$$\frac{1}{L+1} \sum_{\ell=0}^{L} \left( J^{\mathrm{MFG}}\left(\pi_\mu, \mu, \mu\right) - J^{\mathrm{MFG}}\left(\hat{\pi}_\ell, \mu, \mu\right) \right)$$

$$\frac{\log |\mathcal{A}|}{(1-\gamma)(L+1)} + C'_{\mathrm{TRPO},1} \frac{h^2(L) \log L}{\eta(1-\gamma)(L+1)} + \frac{1}{(1-\gamma)(L+1)} \sum_{\ell=0}^{L} \eta(\ell+2) \overline{\mathsf{d}}_{\mu,\mu}^{\pi_\mu} \epsilon_\ell + C'_{\mathrm{TRPO},1} \frac{\epsilon}{(1-\gamma)^2} \left\| \frac{\overline{\mathsf{d}}_{\mu,\mu}^{\pi_\mu}}{\nu} \right\|_{\mathrm{TV}} ,$$

where we used Assumption 4 to bound the last term. Thus, combining this with (21), we obtain that

$$J^{\mathrm{MFG}}\left(\pi_\mu, \mu, \mu\right) - J^{\mathrm{MFG}}\left(\hat{\pi}_L^{\mathtt{Unif},\mu}, \mu, \mu\right)$$

$$\leq \frac{1}{L+1} \sum_{\ell=0}^{L} \left( J^{\mathrm{MFG}}\left(\pi_\mu, \mu, \mu\right) - J^{\mathrm{MFG}}\left(\hat{\pi}_\ell, \mu, \mu\right) \right)$$

$$\leq C'_{\mathrm{TRPO},1} \left( \frac{\left( \|\mathsf{r}\|_\infty^2 + \eta^2 \log^2 |\mathcal{A}| \right) |\mathcal{A}|^2 \log L}{\eta(1-\gamma)^3 (L+1)} + \frac{\epsilon}{(1-\gamma)^2} \left\| \frac{\overline{\mathsf{d}}_{\mu,\mu}^{\pi_\mu}}{\nu} \right\|_{\mathrm{TV}} \right) \ .$$

$\square$

### D.2. Initialization step in the sample-based algorithm

The initialization step of the `Sample-Based MF-TRPO` algorithm presents particular challenges due to the limited operations allowed, specifically the **reset** and **action** operations, as described in Section 2. During each iteration of the algorithm, the initial state must be sampled from the distribution $\hat{\mu}_{k-1}$.

As detailed in Section 5, the distribution update in the algorithm follows the iterative rule:

$$\hat{\mu}_k \leftarrow \hat{\mu}_{k-1} + \beta_k \left( \hat{\zeta}_k - \hat{\mu}_{k-1} \right),$$

where $\hat{\zeta}_k$ is the output of a single iteration of the `Sample-Based MF-TRPO` algorithm.

Consequently, at iteration $k$, the distribution $\hat{\mu}_{k-1}$ is an estimator of

$$\prod_{j=1}^{k-1}(1-\beta_j)\nu + \sum_{\ell=1}^{k-1}\beta_\ell\prod_{j=\ell+1}^{k-1}(1-\beta_j)\nu\left(\mathsf{P}_{\hat{\mu}_1}^{\hat{\pi}_1}\right)^M\cdots\left(\mathsf{P}_{\hat{\mu}_\ell}^{\hat{\pi}_\ell}\right)^M,$$

since the update $\hat{\zeta}_\ell$ at iteration $\ell$ of `Sample-Based MF-TRPO`, is an unbiased estimator of the product distribution $\nu(\mathsf{P}_{\hat{\mu}_1}^{\hat{\pi}_1})^M\cdots(\mathsf{P}_{\hat{\mu}_i}^{\hat{\pi}_i})^M$.

To correctly initialize the environment to a state s such that $s \sim \hat{\mu}_k$, the following subroutine is applied:

1. **Sampling a Level.** Define the categorical random variable $\mathrm{Cat}_k$ that takes value in the discrete space $\{0, 1, \ldots, k\}$ with probabilities given by

$$\left(\prod_{j=1}^k(1-\beta_j), \beta_1\prod_{j=2}^k(1-\beta_j), \ldots, \beta_{k-1}(1-\beta_k), \beta_k\right),$$

   *i.e.,*

$$\mathbb{P}\left(\mathrm{Cat}_k = \ell\right) = \beta_\ell\prod_{j=\ell+1}^k(1-\beta_j).$$

2. **Selecting a Level.** Draw a sample $\hat{\ell} \sim \mathrm{Cat}_k$.

3. **Rollout Procedure.** Starting from an initial state sampled as $s_{0,p,k}^{\mathrm{init}} \sim \nu$, execute a rollout of the Markov transition kernels up to level $\hat{\ell}$, having $\hat{\mu}_{k-1}$ to be the particle approximation of $\nu(\mathsf{P}_{\hat{\mu}_1}^{\hat{\pi}_1})^M\cdots(\mathsf{P}_{\hat{\mu}_\ell}^{\hat{\pi}_\ell})^M$.

### D.3. Sample based algorithm and High Probability Estimates

Transitioning from exact computations to a sample-based setting, we introduce estimators for the key quantities involved in the learning process. These estimators leverage sampled trajectories to approximate the necessary expectations while maintaining computational efficiency.

To ensure the reliability of these approximations, we establish high-probability error bounds by leveraging concentration inequalities. This allows us to rigorously assess the performance of the algorithm, providing quantitative guarantees on the estimation error and its impact on the overall convergence rate. Through this probabilistic framework, we ensure that the sample-based algorithm retains stability and efficiency despite the inherent stochasticity.

---

**Algorithm 8** `Sample-Based MF-TRPO`

---

1: **Input:** $K$.
2: **Initialize:** Initial policy $\pi_0(\cdot|s) = \mathcal{U}(\mathcal{A})$, for any $s \in \mathcal{S}$. Initial distribution $\mu_0 = \nu$.
3: **for** $k \in [K]$ **do**
4:     $\hat{\pi}_k \leftarrow$ `Sample-Based TRPO`$(\hat{\mu}_{k-1})$.                    # Update of the policy.
5:     **for** $p \in [P]$ **do**
6:         $\hat{\ell} \sim \mathrm{Cat}_k$                                                        # Sampling a level.
7:         Sample $s_{0,p,k}^{\mathrm{init}} \sim \nu$.
8:         **for** $\ell \in [\hat{\ell} - 1]$ **do**
9:             **for** $m \in [M]$ **do**
10:                 Sample $s_{m+(\ell-1)M,p,k}^{\mathrm{init}} \sim \mathsf{P}_{\hat{\mu}_\ell}^{\hat{\pi}_\ell}(\cdot|s_{(m-1)+(\ell-1)M,p,k}^{\mathrm{init}})$.   # Rollout from the MDP for level $\ell$.
11:             **end for**
12:         **end for**
13:         Initialize $s_{0,p,k} = s_{\hat{\ell}M,p,k}^{\mathrm{init}}$.                              # Initialization.
14:         **for** $m \in [M]$ **do**
15:             Sample $s_{m,p,k} \sim \sum_{a \in \mathcal{A}} \mathsf{P}(\cdot|s_{m-1,p,k}, a, \hat{\mu}_{k-1}) \hat{\pi}_k(a|s_{m-1,p,k})$.        # Rollout from the MDP.
16:         **end for**
17:         $\hat{\zeta}_{k,p} \leftarrow \mathbb{1}_{\{s_{M,p,k}\}}(\cdot)$.
18:     **end for**
19:     $\hat{\zeta}_k \leftarrow \frac{1}{P} \sum_{p=1}^{P} \hat{\zeta}_{k,p}$.
20:     $\hat{\mu}_k \leftarrow \hat{\mu}_{k-1} + \beta_k \left( \hat{\zeta}_k - \hat{\mu}_{k-1} \right)$.                    # Update population distribution.
21: **end for**
22: **Output:** $\mu_K$.

---

Examining the `Sample-Based MF-TRPO` algorithm, we observe that two key approximations are introduced in the learning process. First, the policy update is performed through `Sample-Based TRPO`, whose finite-sample analysis in high probability is established in Theorem D.2. This result ensures that the policy iterates remain well-controlled throughout the optimization process in high probability. Secondly, in order to analyze the evolution of the mean-field population distribution, we need to establish a similar high-probability bound on the estimation of the term $\hat{\mu}_{k-1} (\mathsf{P}_{\hat{\mu}_{k-1}}^{\hat{\pi}_k})^M$, which represents the transition dynamics under the estimated policy.

The unbiased estimator of this term uses the trajectories $\{s_{m,p,k-1}\}_{m=0}^{M}$ and is given by the empirical sum of $\hat{\zeta}_{k,p} = \mathbb{1}_{\{s_{M,p,k}\}}(\cdot)$. Note that each component of this vector is distributed according to a Bernoulli distribution and is centered in $\hat{\mu}_{k-1} (\mathsf{P}_{\hat{\mu}_{k-1}}^{\hat{\pi}_k})^M$. Therefore, define $\epsilon_k$ the following martingale difference term

$$\epsilon_k = \frac{1}{P} \sum_{p=1}^{P} \hat{\zeta}_{k,p} - \hat{\mu}_{k-1} (\mathsf{P}_{\hat{\mu}_{k-1}}^{\hat{\pi}_k})^M ,$$

with $s_{M,p,k-1}$ defined as in `Sample-Based MF-TRPO`.

To address this, we first derive Proposition D.3, which is a preliminary concentration result that quantifies the approximation error in the estimation of this key quantity. The first one is Proposition D.3, which provides guarantees on the deviation of the error incurred in a single iteration of the algorithm, with high probability. Specifically, it establishes that the error made while estimating the error at each iteration, which is a bounded increment of a martingale. This result is pivotal, as it ensures that the errors introduced in each iteration of the algorithm are controlled and do not diverge as the algorithm progresses, and lays the foundation for a rigorous convergence analysis of `Sample-Based MF-TRPO`.

**Proposition D.3.** *For any $\epsilon > 0$ and $\delta > 0$, if the number of trajectories in the $k$-th iteration satisfies:*

$$P \geq \frac{64}{\epsilon^2} \log \frac{2}{\delta} ,$$

*then, with probability at least $1 - \delta$, the following holds:*

$$\|\epsilon_k\|_2 \leq \epsilon .$$

*Proof.* From previous consideration, we have that $\hat{\zeta}_k$ is unbiased and $\epsilon_k$ is a martingale difference. Moreover, note that $\hat{\zeta}_{k,p} - \hat{\mu}_{k-1} \, (\mathsf{P}^{\hat{\pi}_k}_{\hat{\mu}_{k-1}})^M$ is a bounded vectore, *i.e.*

$$\left\| \hat{\zeta}_{k,p} - \hat{\mu}_{k-1} \, (\mathsf{P}^{\hat{\pi}_k}_{\hat{\mu}_{k-1}})^M \right\|_2 \leq 2 \left\| \hat{\zeta}_{k,p} - \hat{\mu}_{k-1} \, (\mathsf{P}^{\hat{\pi}_k}_{\hat{\mu}_{k-1}})^M \right\|_1$$

$$= 2 \sum_{s \in \mathcal{S}} \left| \hat{\zeta}_{k,p}(s) - \hat{\mu}_{k-1} \, (\mathsf{P}^{\hat{\pi}_k}_{\hat{\mu}_{k-1}})^M(s) \right|$$

$$\leq 2 \left( \sum_{s \in \mathcal{S}} \hat{\zeta}_{k,p}(s) + \sum_{s \in \mathcal{S}} \hat{\mu}_{k-1} \, (\mathsf{P}^{\hat{\pi}_k}_{\hat{\mu}_{k-1}})^M(s) \right) = 4 \ .$$

Moreover, using Jenses's inequality, we have that

$$\|\epsilon_k\|_2 = \left\| \frac{1}{P} \sum_{p=1}^{P} \hat{\zeta}_{k,p} - \hat{\mu}_{k-1} \, (\mathsf{P}^{\hat{\pi}_k}_{\hat{\mu}_{k-1}})^M \right\|_2 \leq \frac{1}{P} \sum_{p=1}^{P} \left\| \hat{\zeta}_{k,p} - \hat{\mu}_{k-1} \, (\mathsf{P}^{\hat{\pi}_k}_{\hat{\mu}_{k-1}})^M \right\|_2 \leq 4 \ .$$

Therefore, $\epsilon_k$ is a bounded martingale difference. To show that the increment is bounded with high probability, we use Hoeffding's inequality. Let $\epsilon_k = \sum_{p=1}^{P} \epsilon_{k,p}$, where $\epsilon_{k,p} = \hat{\zeta}_{k,p} - \hat{\mu}_{k-1} \, (\mathsf{P}^{\hat{\pi}_k}_{\hat{\mu}_{k-1}})^M$. Then, for any $t_k > 0$, we have

$$\mathbb{P} \left( \frac{1}{P} \sum_{p=0}^{P} \sum_{s \in \mathcal{S}} \left| \epsilon_{k,p}(s) - \hat{\mu}_{k-1} \, (\mathsf{P}^{\hat{\pi}_k}_{\hat{\mu}_{k-1}})^M(s) \right| \geq \frac{\epsilon}{4} \right) = \mathbb{P} \left( \frac{1}{P} \sum_{p=0}^{P} \left\| \epsilon_{k,p} - \hat{\mu}_{k-1} \, \left(\mathsf{P}^{\hat{\pi}_k}_{\hat{\mu}_{k-1}}\right)^M \right\|_1 \geq \frac{\epsilon}{4} \right)$$

$$\leq 2 \exp \left( -\frac{P\epsilon^2}{64} \right) =: \delta \ .$$

This consideration is a special case of the Generalized Freedman inequality as presented in Harvey et al. (2019). The inequality provides sharp high-probability bounds for the sum of bounded, dependent random variables. In our case, the formulation is simplified due to the presence of a uniform bound on the variables we aim to control.

Therefore, in order to guarantee that

$$\left\| \epsilon_k - \hat{\mu}_{k-1} \, \left(\mathsf{P}^{\hat{\pi}_k}_{\hat{\mu}_{k-1}}\right)^M \right\|_2 \leq \frac{4}{P} \sum_{p=0}^{P} \left\| \epsilon_{k,p} - \hat{\mu}_{k-1} \, \left(\mathsf{P}^{\hat{\pi}_k}_{\hat{\mu}_{k-1}}\right)^M \right\|_1$$

$$\leq \epsilon \ ,$$

we need the number of trajectories $P$ to be at least

$$P \geq \frac{64}{\epsilon^2} \log \frac{2}{\delta} \ .$$

$\square$

## D.4. Convergence of `Sample-Based MF-TRPO`

We now extend the exact analysis of `Exact MF-TRPO` to its sample-based counterpart, establishing global sample complexity bounds. While the exact algorithm benefits from having full knowledge of the transition kernel and reward function, the sample-based version introduces additional approximation errors due to finite sampling. We quantify these errors and derive high-probability guarantees on the convergence of the algorithm. This requires adapting the theoretical tools developed in the exact setting to account for trajectory-based estimations and ensuring that the resulting policy updates remain stable despite stochastic approximations.

**Theorem D.4.** *Suppose that Assumptions 1, 2, 3, and 4 hold. Assume that the following holds*

$$\beta_k < b_1 := \frac{\tau}{6C_{\pi,\mu}^2 C_{\text{Erg,M}}^2 + 2\tau} \ , \qquad \text{for } k \geq 1 \ . \tag{23}$$

*For any $\epsilon > 0$ and $\delta > 0$, if the number of trajectories in each iteration for `Sample-Based MF-TRPO` satisfies*

$$P \geq \frac{64}{\epsilon^2} \log \frac{2}{\delta} \ , \tag{24}$$

*and the number of iteration in each epoch of* `Sample-Based TRPO` *satisfies*

$$I_\ell \geq \frac{|\mathcal{A}|^2 \left(\|\mathbf{r}\|_\infty + \eta^2 \log^2 |\mathcal{A}|\right) \left(|\mathcal{S}| \log 2|\mathcal{A}| + \log \frac{1}{\delta}\right)}{(1-\gamma)^2 \epsilon^2} \tag{25}$$

*and the rollout is performed up to time $T_\ell$ with*

$$T_\ell \geq \frac{1}{1-\gamma} \log \left(\frac{\epsilon}{|\mathcal{A}| \left(\|\mathbf{r}\|_\infty + \eta \log |\mathcal{A}|\right)}\right) . \tag{26}$$

*Then, with probability at least $1 - \delta$, we have that*

$$\|\hat{\mu}_K - \mu_\star\|_2^2 \leq \exp\left(-\frac{\tau}{2} \sum_{j=1}^{k} \beta_j\right) \|\mu_0 - \mu_\star\|_2^2 + \frac{2C_{\mathrm{MF},1}}{\tau} \frac{\log(L)}{L} + \frac{2C_{\mathrm{MF},2}}{\tau} \epsilon . \tag{27}$$

*with*

$$\tau := 1 - C_{\mathrm{op,MFG}}$$

$$C_{\mathrm{MF},1} := C_{\mathrm{Erg,M}} \cdot C_{\mathrm{TRPO},1} \cdot \frac{2 + b_1}{\tau} ,$$

$$C_{\mathrm{MF},2} := \frac{2 + b_1}{\tau} \left(C_{\mathrm{Erg,M}} \, C_{\mathrm{TRPO},2} + 1\right) ,$$

$$C_{\mathrm{TRPO},1} := \frac{2C'_{\mathrm{TRPO},1}}{\eta(1-\gamma)} \cdot \frac{\left(\|\mathbf{r}\|_\infty^2 + \eta^2 \log^2 |\mathcal{A}|\right) |\mathcal{A}|^2 \log L}{\eta(1-\gamma)^3 L} ,$$

$$C_{\mathrm{TRPO},2} := \frac{2C'_{\mathrm{TRPO},1}}{\eta(1-\gamma)} \cdot \frac{1}{(1-\gamma)^2} \left\|\frac{\overline{\mathbf{d}}_{\mu,\mu}^{\pi_\mu}}{\nu}\right\|_\infty ,$$

*with $C'_{\mathrm{TRPO},1}$ the constant coming from Theorem D.2.*

*Proof.* We focus on the convergence of the sequence $\hat{\mu}_k \to \mu_\star$, with $\mu_\star$ as in (8). Denote $\hat{\pi}_k$ (resp. $\hat{\zeta}_k$) the output of `Sample-Based TRPO($\hat{\mu}_k$)` at each step (resp. the estimator used in the update of $\hat{\mu}_k$ in `Sample-Based MF-TRPO`).

We then have that

$$
\begin{aligned}
\|\hat{\mu}_k - \mu_\star\|_2^2 &= \left\| \hat{\mu}_{k-1} - \mu_\star + \beta_k \left( \hat{\zeta}_k - \hat{\mu}_{k-1} \right) \right\|_2^2 \\
&= \left\| \hat{\mu}_{k-1} - \mu_\star + \beta_k \left( \hat{\zeta}_k - \hat{\mu}_{k-1} \left( \mathsf{P}_{\hat{\mu}_{k-1}}^{\hat{\pi}_k} \right)^M \right) + \beta_k \left( \hat{\mu}_{k-1} \left( \mathsf{P}_{\hat{\mu}_{k-1}}^{\hat{\pi}_k} \right)^M - \hat{\mu}_{k-1} \right) \right\|_2^2 \\
&= \left\| \hat{\mu}_{k-1} - \mu_\star + \beta_k \left( \hat{\zeta}_k - \hat{\mu}_{k-1} \left( \mathsf{P}_{\hat{\mu}_{k-1}}^{\hat{\pi}_k} \right)^M \right) \right. \\
&\qquad\qquad + \beta_k \left( \hat{\mu}_{k-1} \left( \mathsf{P}_{\hat{\mu}_{k-1}}^{\hat{\pi}_k} \right)^M - \hat{\mu}_{k-1} \left( \mathsf{P}_{\hat{\mu}_{k-1}}^{\pi_{\hat{\mu}_{k-1}}} \right)^M \right) \\
&\qquad\qquad + \left. \beta_k \left( \hat{\mu}_{k-1} \left( \mathsf{P}_{\hat{\mu}_{k-1}}^{\pi_{\hat{\mu}_{k-1}}} \right)^M - \hat{\mu}_{k-1} \right) \right\|_2^2 \\
&= \left\| \hat{\mu}_{k-1} - \mu_\star + \beta_k \left( \hat{\zeta}_k - \hat{\mu}_{k-1} \left( \mathsf{P}_{\hat{\mu}_{k-1}}^{\hat{\pi}_k} \right)^M \right) \right. \\
&\qquad\qquad + \beta_k \left( \hat{\mu}_{k-1} \left( \mathsf{P}_{\hat{\mu}_{k-1}}^{\hat{\pi}_k} \right)^M - \hat{\mu}_{k-1} \left( \mathsf{P}_{\hat{\mu}_{k-1}}^{\pi_{\hat{\mu}_{k-1}}} \right)^M \right) \\
&\qquad\qquad + \left. \beta_k \left( \hat{\mu}_{k-1} \left( \mathsf{P}_{\hat{\mu}_{k-1}}^{\pi_{\hat{\mu}_{k-1}}} \right)^M - \hat{\mu}_{k-1} \right) - \beta_k \left( \mu_\star \left( \mathsf{P}_{\mu_\star}^{\pi_{\mu_\star}} \right)^M - \mu_\star \right) \right\|_2^2 \\
&= \left\| (1 - \beta_k)(\hat{\mu}_{k-1} - \mu_\star) + \beta_k \left( \hat{\zeta}_k - \hat{\mu}_{k-1} \left( \mathsf{P}_{\hat{\mu}_{k-1}}^{\hat{\pi}_k} \right)^M \right) \right. \\
&\qquad\qquad + \beta_k \left( \hat{\mu}_{k-1} \left( \mathsf{P}_{\hat{\mu}_{k-1}}^{\hat{\pi}_k} \right)^M - \hat{\mu}_{k-1} \left( \mathsf{P}_{\hat{\mu}_{k-1}}^{\pi_{\hat{\mu}_{k-1}}} \right)^M \right) \\
&\qquad\qquad + \left. \beta_k \left( \hat{\mu}_{k-1} \left( \mathsf{P}_{\hat{\mu}_{k-1}}^{\pi_{\hat{\mu}_{k-1}}} \right)^M - \mu_\star \left( \mathsf{P}_{\mu_\star}^{\pi_{\mu_\star}} \right)^M \right) \right\|_2^2 \\
&= (1 - \beta_k)^2 \|\hat{\mu}_{k-1} - \mu_\star\|_2^2 \\
&\quad + 2(1 - \beta_k)\beta_k \left\langle \hat{\mu}_{k-1} - \mu_\star, \hat{\zeta}_k - \hat{\mu}_{k-1} \left( \mathsf{P}_{\hat{\mu}_{k-1}}^{\hat{\pi}_k} \right)^M \right\rangle \\
&\quad + \beta_k^2 \left\| \hat{\zeta}_k - \hat{\mu}_{k-1} \left( \mathsf{P}_{\hat{\mu}_{k-1}}^{\hat{\pi}_k} \right)^M \right\|_2^2 \\
&\quad + 2\beta_k^2 \left\langle \hat{\zeta}_k - \hat{\mu}_{k-1} \left( \mathsf{P}_{\hat{\mu}_{k-1}}^{\hat{\pi}_k} \right)^M, \hat{\mu}_{k-1} \left( \mathsf{P}_{\hat{\mu}_{k-1}}^{\hat{\pi}_k} \right)^M - \hat{\mu}_{k-1} \left( \mathsf{P}_{\hat{\mu}_{k-1}}^{\pi_{\hat{\mu}_{k-1}}} \right)^M \right\rangle \\
&\quad + 2\beta_k^2 \left\langle \hat{\zeta}_k - \hat{\mu}_{k-1} \left( \mathsf{P}_{\hat{\mu}_{k-1}}^{\hat{\pi}_k} \right)^M, \hat{\mu}_{k-1} \left( \mathsf{P}_{\hat{\mu}_{k-1}}^{\pi_{\hat{\mu}_{k-1}}} \right)^M - \mu_\star \left( \mathsf{P}_{\mu_\star}^{\pi_{\mu_\star}} \right)^M \right\rangle \\
&\quad + 2(1 - \beta_k)\beta_k \left\langle \hat{\mu}_{k-1} - \mu_\star, \hat{\mu}_{k-1} \left( \mathsf{P}_{\hat{\mu}_{k-1}}^{\pi_{\hat{\mu}_{k-1}}} \right)^M - \mu_\star \left( \mathsf{P}_{\mu_\star}^{\pi_{\mu_\star}} \right)^M \right\rangle \\
&\quad + \beta_k^2 \left\| \hat{\mu}_{k-1} \left( \mathsf{P}_{\hat{\mu}_{k-1}}^{\pi_{\hat{\mu}_{k-1}}} \right)^M - \mu_\star \left( \mathsf{P}_{\mu_\star}^{\pi_{\mu_\star}} \right)^M \right\|_2^2 \\
&\quad + 2(1 - \beta_k)\beta_k \left\langle \hat{\mu}_{k-1} - \mu_\star, \hat{\mu}_{k-1} \left( \mathsf{P}_{\hat{\mu}_{k-1}}^{\hat{\pi}_k} \right)^M - \hat{\mu}_{k-1} \left( \mathsf{P}_{\hat{\mu}_{k-1}}^{\pi_{\hat{\mu}_{k-1}}} \right)^M \right\rangle \\
&\quad + \beta_k^2 \left\| \hat{\mu}_{k-1} \left( \mathsf{P}_{\hat{\mu}_{k-1}}^{\hat{\pi}_k} \right)^M - \hat{\mu}_{k-1} \left( \mathsf{P}_{\hat{\mu}_{k-1}}^{\pi_{\hat{\mu}_{k-1}}} \right)^M \right\|_2^2 \\
&\quad + 2\beta_k^2 \left\langle \hat{\mu}_{k-1} \left( \mathsf{P}_{\hat{\mu}_{k-1}}^{\hat{\pi}_k} \right)^M - \hat{\mu}_{k-1} \left( \mathsf{P}_{\hat{\mu}_{k-1}}^{\pi_{\hat{\mu}_{k-1}}} \right)^M, \hat{\mu}_{k-1} \left( \mathsf{P}_{\hat{\mu}_{k-1}}^{\pi_{\hat{\mu}_{k-1}}} \right)^M - \mu_\star \left( \mathsf{P}_{\mu_\star}^{\pi_{\mu_\star}} \right)^M \right\rangle .
\end{aligned}
$$

Applying Assumptions 2 and Corollary E.4 together with Lemma E.1, and following the same lines as in the proof

of Proposition C.3, the previous equality implies that

$$
\begin{aligned}
\|\hat\mu_k - \mu_\star\|_2^2 &\leq \left[(1-\beta_k)\left(1+(2C_{\mathrm{op,MFG}}-1)\beta_k\right) + \beta_k^2 C_{\pi,\mu}^2 C_{\mathrm{Erg,M}}^2\right]\|\hat\mu_{k-1}-\mu_\star\|_2^2 \\
&\quad + 2(1-\beta_k)\beta_k\left\langle \hat\mu_{k-1}-\mu_\star, \hat\zeta_k - \hat\mu_{k-1}\left(\mathsf{P}_{\hat\mu_{k-1}}^{\hat\pi_k}\right)^M \right\rangle \\
&\quad + \beta_k^2\left\|\hat\zeta_k - \hat\mu_{k-1}\left(\mathsf{P}_{\hat\mu_{k-1}}^{\hat\pi_k}\right)^M\right\|_2^2 \\
&\quad + 2\beta_k^2\left\langle \hat\zeta_k - \hat\mu_{k-1}\left(\mathsf{P}_{\hat\mu_{k-1}}^{\hat\pi_k}\right)^M, \hat\mu_{k-1}\left(\mathsf{P}_{\hat\mu_{k-1}}^{\hat\pi_k}\right)^M - \hat\mu_{k-1}\left(\mathsf{P}_{\hat\mu_{k-1}}^{\pi_{\hat\mu_{k-1}}}\right)^M\right\rangle \\
&\quad + 2\beta_k^2\left\langle \hat\zeta_k - \hat\mu_{k-1}\left(\mathsf{P}_{\hat\mu_{k-1}}^{\hat\pi_k}\right)^M, \hat\mu_{k-1}\left(\mathsf{P}_{\hat\mu_{k-1}}^{\pi_{\hat\mu_{k-1}}}\right)^M - \mu_\star\left(\mathsf{P}_{\mu_\star}^{\pi_{\mu_\star}}\right)^M\right\rangle \\
&\quad + 2(1-\beta_k)\beta_k\left\langle \hat\mu_{k-1}-\mu_\star, \hat\mu_{k-1}\left(\mathsf{P}_{\hat\mu_{k-1}}^{\hat\pi_k}\right)^M - \hat\mu_{k-1}\left(\mathsf{P}_{\hat\mu_{k-1}}^{\pi_{\hat\mu_{k-1}}}\right)^M\right\rangle \\
&\quad + \beta_k^2\left\|\hat\mu_{k-1}\left(\mathsf{P}_{\hat\mu_{k-1}}^{\hat\pi_k}\right)^M - \hat\mu_{k-1}\left(\mathsf{P}_{\hat\mu_{k-1}}^{\pi_{\hat\mu_{k-1}}}\right)^M\right\|_2^2 \\
&\quad + 2\beta_k^2\left\langle \hat\mu_{k-1}\left(\mathsf{P}_{\hat\mu_{k-1}}^{\hat\pi_k}\right)^M - \hat\mu_{k-1}\left(\mathsf{P}_{\hat\mu_{k-1}}^{\pi_{\hat\mu_{k-1}}}\right)^M, \hat\mu_{k-1}\left(\mathsf{P}_{\hat\mu_{k-1}}^{\pi_{\hat\mu_{k-1}}}\right)^M - \mu_\star\left(\mathsf{P}_{\mu_\star}^{\pi_{\mu_\star}}\right)^M\right\rangle \\
&=: \left[(1-\beta_k)\left(1+(2C_{\mathrm{op,MFG}}-1)\beta_k\right) + \beta_k^2 C_{\pi,\mu}^2 C_{\mathrm{Erg,M}}^2\right]\|\hat\mu_{k-1}-\mu_\star\|_2^2 \\
&\quad + 2\beta_k(1-\beta_k)\mathbf{E}_1 + \beta_k^2\mathbf{E}_2 + 2\beta_k^2\mathbf{E}_3 + 2\beta_k^2\mathbf{E}_4 \\
&\quad + 2\beta_k(1-\beta_k)\mathbf{T}_1 + \beta_k^2\mathbf{T}_2 + 2\beta_k^2\mathbf{T}_3 \ .
\end{aligned}
$$

We now proceed in studying the terms $\mathbf{E}_1$, $\mathbf{E}_2$, $\mathbf{E}_3$, $\mathbf{E}_4$, $\mathbf{T}_1$, $\mathbf{T}_2$, and $\mathbf{T}_3$. Using Young's inequality, we get that

$$
\begin{aligned}
|\mathbf{E}_1| &\leq \frac{\tau}{4}\|\hat\mu_{k-1}-\mu_\star\|_2^2 + \frac{1}{\tau}\left\|\hat\zeta_k - \hat\mu_{k-1}\left(\mathsf{P}_{\hat\mu_{k-1}}^{\hat\pi_k}\right)^M\right\|_2^2 \leq \frac{\tau}{4}\|\hat\mu_{k-1}-\mu_\star\|_2^2 + \frac{1}{\tau}\mathbf{E}_2 \ , \\
|\mathbf{E}_3| &\leq \frac{1}{2}\left\|\hat\mu_{k-1}\left(\mathsf{P}_{\hat\mu_{k-1}}^{\hat\pi_k}\right)^M - \hat\mu_{k-1}\left(\mathsf{P}_{\hat\mu_{k-1}}^{\pi_{\hat\mu_{k-1}}}\right)^M\right\|_2^2 + \frac{1}{2}\left\|\hat\zeta_k - \hat\mu_{k-1}\left(\mathsf{P}_{\hat\mu_{k-1}}^{\hat\pi_k}\right)^M\right\|_2^2 \leq \frac{1}{2}\mathbf{T}_2 + \frac{1}{2}\mathbf{E}_2 \ , \\
|\mathbf{E}_4| &\leq \frac{1}{2}\left\|\hat\mu_{k-1}\left(\mathsf{P}_{\hat\mu_{k-1}}^{\pi_{\hat\mu_{k-1}}}\right)^M - \mu_\star\left(\mathsf{P}_{\mu_\star}^{\pi_{\mu_\star}}\right)^M\right\|_2^2 + \frac{1}{2}\left\|\hat\zeta_k - \hat\mu_{k-1}\left(\mathsf{P}_{\hat\mu_{k-1}}^{\hat\pi_k}\right)^M\right\|_2^2 \\
&\leq \frac{1}{2}C_{\pi,\mu}^2 C_{\mathrm{Erg,M}}^2\|\hat\mu_{k-1}-\mu_\star\|_2^2 + \frac{1}{2}\mathbf{E}_2 \ ,
\end{aligned}
$$

where we used Lemma E.1 and Corollary E.4 in the last inequality. Using Young's inequality, we get that

$$
\begin{aligned}
|\mathbf{T}_1| &\leq \frac{\tau}{4}\|\hat\mu_{k-1}-\mu_\star\|_2^2 + \frac{1}{\tau}\left\|\hat\mu_{k-1}\left(\mathsf{P}_{\hat\mu_{k-1}}^{\hat\pi_k}\right)^M - \hat\mu_{k-1}\left(\mathsf{P}_{\hat\mu_{k-1}}^{\pi_{\hat\mu_{k-1}}}\right)^M\right\|_2^2 \leq \frac{\tau}{4}\|\hat\mu_{k-1}-\mu_\star\|_2^2 + \frac{1}{\tau}\mathbf{T}_2 \ , \\
|\mathbf{T}_3| &\leq \frac{1}{2}\left\|\hat\mu_{k-1}\left(\mathsf{P}_{\hat\mu_{k-1}}^{\pi_{\hat\mu_{k-1}}}\right)^M - \mu_\star\left(\mathsf{P}_{\mu_\star}^{\pi_{\mu_\star}}\right)^M\right\|_2^2 + \frac{1}{2}\left\|\hat\mu_{k-1}\left(\mathsf{P}_{\hat\mu_{k-1}}^{\hat\pi_k}\right)^M - \hat\mu_{k-1}\left(\mathsf{P}_{\hat\mu_{k-1}}^{\pi_{\hat\mu_{k-1}}}\right)^M\right\|_2^2 \\
&\leq \frac{1}{2}C_{\pi,\mu}^2 C_{\mathrm{Erg,M}}^2\|\hat\mu_{k-1}-\mu_\star\|_2^2 + \frac{1}{2}\mathbf{T}_2 \ ,
\end{aligned}
$$

where we used Lemma E.1 and Corollary E.4 in the last inequality. Since $\tau < 1$ from Assumption 2 and $\beta_k$ satisfies (23), a straightforward computation shows that

$$
(1-\beta_k)\left(1+(1-\tau)\beta_k\right) + 3\beta_k^2 C_{\pi,\mu}^2 C_{\mathrm{Erg,M}}^2 < 1 - \frac{\tau}{2}\beta_k \ .
$$

Since we have that (25) holds, we can apply Theorem D.2, together with Lemma E.1, to get that

$$
\begin{aligned}
\mathbf{T}_2 &\leq C_{\mathrm{Erg,M}} \left( \sum_{s \in \mathcal{S}} \hat{\mu}_{k-1}^2(s) \left\| \hat{\pi}_k(\cdot|s) - \pi_{\hat{\mu}_{k-1}}(\cdot|s) \right\|_{\mathrm{TV}}^2 \right) \\
&\leq C_{\mathrm{Erg,M}} \left( \sum_{s \in \mathcal{S}} \hat{\mu}_{k-1}(s) \left\| \hat{\pi}_k(\cdot|s) - \pi_{\hat{\mu}_{k-1}}(\cdot|s) \right\|_{\mathrm{TV}}^2 \right) \\
&\leq C_{\mathrm{Erg,M}} \left( J^{\mathrm{MFG}}(\pi_{\hat{\mu}_{k-1}}, \hat{\mu}_{k-1}, \hat{\mu}_{k-1}) - J^{\mathrm{MFG}}(\hat{\pi}_k, \hat{\mu}_{k-1}, \hat{\mu}_{k-1}) \right) \\
&\leq C_{\mathrm{Erg,M}} \frac{2}{\eta(1-\gamma)} \left( C_{\mathrm{TRPO,1}} \frac{\log(L)}{L} + C_{\mathrm{TRPO,2}} \, \epsilon \right) .
\end{aligned}
$$

Moreover, since (24) holds, using Proposition D.3, we have that, with probability at least $1 - \delta$,

$$
\mathbf{E}_2 \leq \epsilon.
$$

Therefore, combining the previous inequalities, we have that

$$
\begin{aligned}
&\|\hat{\mu}_k - \mu_\star\|_2^2 \\
&\leq \left(1 - \frac{\tau}{2}\beta_k\right) \|\hat{\mu}_{k-1} - \mu_\star\|_2^2 \\
&\quad + \beta_k \left( \frac{2(1-\beta_k)}{\tau} - 3\beta_k \right) \cdot \left[ C_{\mathrm{Erg,M}} \left( C_{\mathrm{TRPO,1}} \frac{\log(L)}{L} + C_{\mathrm{TRPO,2}} \, \epsilon \right) + \epsilon \right] \\
&\leq \left(1 - \frac{\tau}{2}\beta_k\right) \|\hat{\mu}_{k-1} + \mu_\star\|_2^2 + \beta_k \frac{2 + b_1}{\tau} \left[ C_{\mathrm{Erg,M}} \left( C_{\mathrm{TRPO,1}} \frac{\log(L)}{L} + C_{\mathrm{TRPO,2}} \, \epsilon \right) + \epsilon \right] \\
&\leq \left(1 - \frac{\tau}{2}\beta_k\right) \|\hat{\mu}_{k-1} - \mu_\star\|_2^2 + \beta_k \, C_{\mathrm{MF,1}} \frac{\log(L)}{L} + \beta_k \, C_{\mathrm{MF,2}} \, \epsilon .
\end{aligned}
\tag{28}
$$

Developping the recursion (28), we obtain

$$
\begin{aligned}
\|\hat{\mu}_k - \mu_\star\|_2^2 &\leq \prod_{j=1}^{k} \left(1 - \frac{\tau}{2}\beta_j\right) \|\mu_0 - \mu_\star\|_2^2 \\
&\quad + \left( C_{\mathrm{MF,1}} \frac{\log(L)}{L} + \beta_k \, C_{\mathrm{MF,2}} \, \epsilon \right) \sum_{j=1}^{k} \beta_j \prod_{\ell=j+1}^{k} \left(1 - \frac{\tau}{2}\beta_\ell\right) \\
&\leq \exp\left( -\frac{\tau}{2} \sum_{j=1}^{k} \beta_j \right) \|\mu_0 - \mu_\star\|_2^2 \\
&\quad + \left( C_{\mathrm{MF,1}} \frac{\log(L)}{L} + \beta_k \, C_{\mathrm{MF,2}} \, \epsilon \right) \sum_{j=1}^{k} \beta_j \prod_{\ell=j+1}^{k} \left(1 - \frac{\tau}{2}\beta_\ell\right) .
\end{aligned}
$$

Note that the second term of the r.h.s. of the previous inequality is a telescopic sum, as the central term can be rewritten as

$$
\beta_j \prod_{\ell=j+1}^{k} \left(1 - \frac{\tau}{2}\beta_\ell\right) = \frac{2}{\tau} \left[ \prod_{\ell=j+1}^{k} \left(1 - \frac{\tau}{2}\beta_\ell\right) - \prod_{\ell=j}^{k} \left(1 - \frac{\tau}{2}\beta_\ell\right) \right] .
$$

Therefore, we get (18). Moreover, since $\beta_k$ satisfy (14), this concludes the proof. $\qquad \square$

## D.5. $\varepsilon$-MFNE

We aim to characterize the proximity of an approximate Nash equilibrium, specifically an $\varepsilon$-Nash equilibrium. In this context, we address two key questions:

1. Given a fixed budget $K$ of sampled trajectories, how close the value function to the unique Nash equilibrium?

2. Given a target approximation level $\varepsilon$, how many trajectories $K$ are required to achieve an $\varepsilon$-Nash equilibrium?

These questions are crucial for understanding the sample complexity of learning equilibria in mean-field settings and provide insights into the efficiency of our algorithmic approach.

**Corollary D.5.** *Suppose that Assumptions 1, 2, 3, and 4 hold. Fix $\epsilon, \delta > 0$. Assume that, for any $k \geq 0$, the learning rate $\beta_k$ satisfies (23), and let $P$ be the number of trajectories in each iteration for* `Sample-Based MF-TRPO` *satisfying (24). Let $\hat{\mu}_k$ (resp. $\hat{\pi}_L^{Unif,\hat{\mu}_k}$) the output of* `Sample-Based MF-TRPO` *(resp. of* `Sample-Based TRPO($\hat{\mu}_k$)`*). Then, we have the following bound on the exploitability*

$$\phi(\hat{\pi}_L^{Unif,\hat{\mu}_k}, \hat{\mu}_k) \leq \varepsilon_k ,$$

*with*

$$\hat{\varepsilon}_k := \hat{\delta}_{NE,1,k} + C_\phi \left( \left(1 + C_{\mathrm{Erg},\infty}\left(1 + C_{\pi,\mu}\right)\right) \sqrt{\hat{\delta}_{NE,2,k}} + \frac{2C_{\mathrm{Erg},\infty}\sqrt{|\mathcal{S}|}}{\eta(1-\gamma)} \cdot \sqrt{\hat{\delta}_{NE,1,k}} \right) ,$$

$$\hat{\delta}_{NE,1,k} = C'_{\mathrm{TRPO},1} \left( \frac{\left(\|\mathsf{r}\|_\infty^2 + \eta^2 \log^2 |\mathcal{A}|\right)|\mathcal{A}|^2 \log L}{\eta(1-\gamma)^3(L+1)} + \frac{\epsilon}{(1-\gamma)^2} \sup_{\mu \in \mathcal{P}(\mathcal{S})} \left\| \frac{\overline{\mathsf{d}}_{\mu,\mu}^{\pi_\mu}}{\nu} \right\|_\infty \right) ,$$

$$\hat{\delta}_{NE,2,k} = \exp\left( -\frac{\tau}{2} \sum_{j=1}^k \beta_j \right) \|\mu_0 - \mu_\star\|_2^2 + \frac{2C_{\mathrm{MF},1}}{\tau} \frac{\log(L)}{L} + \frac{2C_{\mathrm{MF},2}}{\tau}\epsilon .$$

*Proof.* From Proposition E.5, we have that the exploitability of a policy $\pi$ and a mean-field parameter $\mu$ can be bound by the gap of optimality of the $\pi$ w.r.t. the value function $J^{\mathrm{MFG}}(\cdot, \mu, \mu)$ and the distance between $\mu$ and the stationary distribution $\lambda_{\pi,\mu}$.

From Theorem D.2, we have that

$$\max_{\pi \in \Pi} J^{\mathrm{MFG}}(\pi, \hat{\mu}_k, \hat{\mu}_k) - J^{\mathrm{MFG}}(\hat{\pi}_L^{Unif,\hat{\mu}_k}, \hat{\mu}_k, \hat{\mu}_k) \leq \hat{\delta}_{\mathrm{NE},1,k} . \tag{29}$$

On the other hand, using the fact that $(\pi_{\mu_\star}, \mu_\star)$ is a MFNE, we have

$$\hat{\mu}_k - \lambda_{\hat{\pi}_L^{Unif,\hat{\mu}_k}, \hat{\mu}_k} = (\hat{\mu}_k - \mu_\star) + \left( \lambda_{\pi_{\mu_\star}, \mu_\star} - \lambda_{\pi_{\hat{\mu}_k}, \hat{\mu}_k} \right) + \left( \lambda_{\pi_{\hat{\mu}_k}, \hat{\mu}_k} - \lambda_{\hat{\pi}_L^{Unif,\hat{\mu}_k}, \hat{\mu}_k} \right) ,$$

for $\ell = 0, \ldots, L$. As in proof of Corollary C.6, we have

$$\left\| \lambda_{\pi_{\mu_\star}, \mu_\star} - \lambda_{\pi_{\hat{\mu}_k}, \hat{\mu}_k} \right\|_2 \leq C_{\mathrm{Erg},\infty}\left(1 + C_{\pi,\mu}\right) \|\mu_\star - \hat{\mu}_k\|_2 .$$

Moreover,

$$\left\| \lambda_{\pi_{\hat{\mu}_k}, \hat{\mu}_k} - \lambda_{\hat{\pi}_L^{Unif,\hat{\mu}_k}, \hat{\mu}_k} \right\|_2 \leq C_{\mathrm{Erg},\infty} \sum_{s \in \mathcal{S}} \hat{\mu}_k(s) \left\| \hat{\pi}_L^{Unif,\hat{\mu}_k}(\cdot|s) - \pi_{\hat{\mu}_k}(\cdot|s) \right\|_{\mathrm{TV}}$$

$$\leq \frac{2C_{\mathrm{Erg},\infty}\sqrt{|\mathcal{S}|}}{\eta(1-\gamma)} \sqrt{J^{\mathrm{MFG}}(\pi_{\hat{\mu}_k}, \hat{\mu}_k, \hat{\mu}_k) - J^{\mathrm{MFG}}(\hat{\pi}_L^{Unif,\hat{\mu}_k}, \hat{\mu}_k, \hat{\mu}_k)}$$

$$\leq \frac{2C_{\mathrm{Erg},\infty}\sqrt{|\mathcal{S}|}}{\eta(1-\gamma)} \cdot \sqrt{\hat{\delta}_{\mathrm{NE},1,k}} .$$

Using the triangle inequality, together with Theorem D.2, we can bound $\left\| \hat{\mu}_k - \lambda_{\hat{\pi}_L^{Unif,\hat{\mu}_k}, \hat{\mu}_k} \right\|_2$ as

$$\left\| \hat{\mu}_k - \lambda_{\hat{\pi}_L^{Unif,\hat{\mu}_k}, \hat{\mu}_k} \right\|_2 \leq \left(1 + C_{\mathrm{Erg},\infty}\left(1 + C_{\pi,\mu}\right)\right) \|\mu_\star - \hat{\mu}_k\|_2 + \frac{2C_{\mathrm{Erg},\infty}\sqrt{|\mathcal{S}|}}{\eta(1-\gamma)} \cdot \sqrt{\hat{\delta}_{\mathrm{NE},1,k}}$$

$$\leq \left(1 + C_{\mathrm{Erg},\infty}\left(1 + C_{\pi,\mu}\right)\right) \sqrt{\hat{\delta}_{\mathrm{NE},2,k}} + \frac{2C_{\mathrm{Erg},\infty}\sqrt{|\mathcal{S}|}}{\eta(1-\gamma)} \cdot \sqrt{\hat{\delta}_{\mathrm{NE},1,k}} .$$

Using the last inequality and (20), together with Proposition E.5, we have that $\phi(\hat{\pi}_L^{\mathtt{Unif},\hat{\mu}_k}, \mu_k) \leq \varepsilon_k$ .

$\square$

*Remark* D.6. It is important to note that our analysis does not directly bound the exploitability of the last iterate but on the **uniform mixture of policies** over the learning process. This distinction arises due to the absence of an exact counterpart to *Howard's theorem* (Howard, 1960) in `Sample-Based TRPO`, as noted in Appendix D.1. Unlike `Exact TRPO`, where policy improvement guarantees can be established step by step, sampling errors introduce additional variability that prevents such guarantees in the sample-based setting.

Despite this limitation, our results demonstrate that the learned policies perform well *on average* and that we approximate the MFNE accordingly. The bounded average exploitability ensures that, over time, the algorithm remains close to an equilibrium, reinforcing the practical effectiveness of the proposed approach in large-scale multi-agent learning.

*Remark* D.7. From the obtained sample complexity result, it follows directly that to achieve an $\varepsilon$-MFNE, the required number of inner policy updates $L$ and outer population updates $K$ must satisfy the following scaling conditions:

$$L \in \widetilde{O}(1/\varepsilon^2) \quad \text{and} \quad K \in \widetilde{O}(\log(1/\varepsilon^2)).$$

In addition to these requirements, we also establish that the number of episodes $P$ and the number of iterations per policy update $I_\ell$ must satisfy

$$P, I_\ell \in \widetilde{O}(1/\varepsilon^4).$$

These additional conditions ensure that the variance introduced by the sampling procedure remains controlled, allowing for a sufficiently accurate estimation of the value function and policy updates. This highlights the tradeoff between computational efficiency and precision in approximating the MFNE, showing that our algorithm achieves a well-balanced complexity while ensuring convergence guarantees.

At each iteration of `Sample-Based MF-TRPO`, the total number of calls to the environment consists of those required by the `Sample-Based TRPO` procedure plus the additional subroutine for updating the mean-field parameter. `Sample-Based TRPO` requires $\widetilde{O}(1/\varepsilon^6)$ environment calls, scaling proportionally to the product $L \times I_\ell$. This aligns with Shani et al. (2020); however, we highlight a distinction stemming from the chosen metrics: the metric they use corresponds to the square root of our exploitability measure, introducing a cubic dependency in terms of $\varepsilon$. Additionally, the total complexity includes a multiplicative factor $K$, whose contribution is negligible in practice due to its logarithmic scaling, preserving overall algorithmic efficiency.

On the other hand, in each iteration of `Sample-Based MF-TRPO`, the update step for the mean-field distribution scales as $\widetilde{O}(P \times I_\ell \times K)$, which means $\widetilde{O}(1/\varepsilon^2)$ calls to the MF-MDP. This complexity arises naturally from the oracle assumption adopted, which involves an initialization step at each iteration potentially requiring up to $K$ steps to accurately initialize the mean-field distribution. While introducing additional complexity, this initialization procedure is crucial for maintaining consistency across iterative population updates, thereby ensuring the stability and convergence accuracy of the algorithm towards the mean-field Nash equilibrium.

Combining these two contributions, we obtain an overall complexity that scales as $\widetilde{O}(1/\varepsilon^6)$, consistent with established convergence rates in the RL literature.

## E. Technical Lemmata

### E.1. Lipschitzness of the Markov reward process iterates

We show in this section that Assumption 1 implies the existence of a Lipschitz constant for the operator $\lambda_{\pi,\mu}$ and $(\mathsf{P}_\mu^\pi)^M$, for any $\mu \in \mathcal{P}(\mathcal{S})$ and $M \geq 0$.

**Lemma E.1.** *Suppose Assumptions 1 and 3 holds. Fix $M \geq 0$ (resp. $M = \infty$). Then, there exists a constant $C_{\mathrm{Erg}} \geq 0$*

*such that*

$$\left\| \xi\left(\mathsf{P}_\mu^\pi\right)^M - \xi\left(\mathsf{P}_{\mu'}^{\pi'}\right)^M \right\|_{\mathrm{TV}} \leq C_{\mathrm{Erg,M}} \left( \sum_{s\in\mathcal{S}} \xi(s) \left\| \pi(\cdot|s) - \pi'(\cdot|s) \right\|_{\mathrm{TV}} + \left\| \mu - \mu' \right\|_2 \right)$$

$$\leq C_{\mathrm{Erg,M}} \left( \sup_{s\in\mathcal{S}} \left\| \pi(\cdot|s) - \pi'(\cdot|s) \right\|_{\mathrm{TV}} + \left\| \mu - \mu' \right\|_2 \right)$$

$$(\text{resp. } \left\| \lambda_{\pi,\mu} - \lambda_{\pi',\mu'} \right\|_{\mathrm{TV}} \leq C_{\mathrm{Erg},\infty} \left( \sum_{s\in\mathcal{S}} \xi(s) \left\| \pi(\cdot|s) - \pi'(\cdot|s) \right\|_{\mathrm{TV}} + \left\| \mu - \mu' \right\|_2 \right)$$

$$\leq C_{\mathrm{Erg},\infty} \left( \sup_{s\in\mathcal{S}} \left\| \pi(\cdot|s) - \pi'(\cdot|s) \right\|_{\mathrm{TV}} \xi(s) + \left\| \mu - \mu' \right\|_2 \right) \; ), \tag{30}$$

*with*

$$C_{\mathrm{Erg,M}} = C_{\mathrm{Erg}} \, L_\mathsf{P} \, \frac{1 - \rho^M}{1 - \rho} \;, \qquad (\text{resp. } C_{\mathrm{Erg},\infty} = \frac{C_{\mathrm{Erg}} \, L_\mathsf{P}}{1 - \rho} \; ).$$

*for $\pi, \pi' \in \Pi$ and $\mu, \mu' \in \mathcal{P}(\mathcal{S})$.*

*Proof.* This proof is adapted from Fort et al. (Lemma 4.2, 2011) on parametrized Markov chains.

**Step 1.** Consider first $M < \infty$. By employing a telescoping sum, we obtain

$$\left(\mathsf{P}_\mu^\pi\right)^M - \left(\mathsf{P}_{\mu'}^{\pi'}\right)^M = \sum_{m=0}^{M-1} \left(\mathsf{P}_\mu^\pi\right)^{M-m-1} \left(\mathsf{P}_\mu^\pi - \mathsf{P}_{\mu'}^{\pi'}\right) \left(\mathsf{P}_{\mu'}^{\pi'}\right)^m \;.$$

Consider a function $\psi \colon \mathcal{S} \to \mathbb{R}_+$ with $\|\psi\|_\infty \leq 1$. Therefore, since $\mathsf{P}_\mu^\pi - \mathsf{P}_{\mu'}^{\pi'}$ is a difference of probabilities, we have that

$$\sum_{s\in\mathcal{S}} \left[ \xi\left(\mathsf{P}_\mu^\pi\right)^M \psi \right](s) - \left[ \xi\left(\mathsf{P}_{\mu'}^{\pi'}\right)^M \psi \right](s)$$

$$= \sum_{s\in\mathcal{S}} \sum_{m=0}^{M-1} \sum_{s'\in\mathcal{S}} \xi\left(\mathsf{P}_\mu^\pi\right)^{M-m-1} \left(\mathsf{P}_\mu^\pi - \mathsf{P}_{\mu'}^{\pi'}\right) \left( \left(\mathsf{P}_{\mu'}^{\pi'}\right)^m (s',s)\psi(s) - \lambda_{\pi',\mu'}\psi(s) \right) \;. \tag{31}$$

This is due to the fact that whenever we evaluate the previous difference of probabilities matrices on $\psi$, the term $\sum_{s\in\mathcal{S}} \lambda_{\pi',\mu'}\psi(s)$ is perceived a constant by the transition kernels $\mathsf{P}_\mu^\pi$ and $\mathsf{P}_{\mu'}^{\pi'}$, summing this part to zero.

Define $\phi_m \colon \mathcal{S} \to \mathbb{R}_+$ as

$$\phi_m(s) = \sum_{s'\in\mathcal{S}} \left(\mathsf{P}_\mu^\pi - \mathsf{P}_{\mu'}^{\pi'}\right)(s,s') \sum_{s''\in\mathcal{S}} \left( \left(\mathsf{P}_{\mu'}^{\pi'}\right)^m (s',s'')\psi(s'') - \lambda_{\pi',\mu'}\psi(s'') \right) \;.$$

From Assumption 3, we have that

$$\left| \left(\mathsf{P}_{\mu'}^{\pi'}\right)^m (s',s'')\psi(s'') - \lambda_{\pi',\mu'}\psi(s'') \right| \leq C_{\mathrm{Erg}}\rho^m \|\psi\|_\infty \;. \tag{32}$$

Therefore, we have

$$|\phi_m(s)| \leq \sup_{s'\in\mathcal{S}} \left( \mathsf{P}_\mu^\pi(s',s) - \mathsf{P}_{\mu'}^{\pi'}(s',s) \right) C_{\mathrm{Erg}}\rho^m \|\psi\|_\infty \;,$$

and, applying Assumption 1, we get

$$|\phi(s)| \leq L_\mathsf{P} \left( \|\pi(\cdot|s) - \pi'(\cdot|s)\|_{\mathrm{TV}} + \|\mu - \mu'\|_2 \right) \times C_{\mathrm{Erg}}\rho^m \|\psi\|_\infty \;,$$

for any $s' \in \mathcal{S}$. Combining this with (31), we get, from the characterization of the total variation norm of the integral with respect to the positive functions bounded in sup-norm by 1, that

$$
\begin{aligned}
\left\| \xi \left( \mathsf{P}_\mu^\pi \right)^M - \xi \left( \mathsf{P}_{\mu'}^{\pi'} \right)^M \right\|_{\mathrm{TV}} &\leq L_\mathsf{P}\, C_{\mathrm{Erg}} \sum_{m=0}^{M-1} \rho^m \sum_{s \in \mathcal{S}} \xi \left( \mathsf{P}_\mu^\pi \right)^{M-m-1} \left( \|\pi(\cdot|s) - \pi'(\cdot|s)\|_{\mathrm{TV}} + \|\mu - \mu'\|_2 \right) \\
&\leq L_\mathsf{P}\, C_{\mathrm{Erg}} \sum_{m=0}^{M-1} \rho^m \sum_{s \in \mathcal{S}} \xi(s) \left( \|\pi(\cdot|s) - \pi'(\cdot|s)\|_{\mathrm{TV}} + \|\mu - \mu'\|_2 \right) \\
&\leq L_\mathsf{P}\, C_{\mathrm{Erg}} \frac{1 - \rho^M}{1 - \rho} \left( \sum_{s \in \mathcal{S}} \xi(s) \|\pi(\cdot|s) - \pi'(\cdot|s)\|_{\mathrm{TV}} + \|\mu - \mu'\|_2 \right) \\
&\leq L_\mathsf{P}\, C_{\mathrm{Erg}} \frac{1 - \rho^M}{1 - \rho} \left( \sup_{s \in \mathcal{S}} \|\pi(\cdot|s) - \pi'(\cdot|s)\|_{\mathrm{TV}} + \|\mu - \mu'\|_2 \right) ,
\end{aligned}
$$

where we have used the fact that $\mathsf{P}_\mu^\pi$ a stochastic matrix, thus its biggest eigenvalue is 1 and $\xi$ is a vector of just positive components.

**Step 2.** Consider now the ergodic distributions. Fix $M \geq 1$. From triangle inequality, we have

$$
\|\lambda_{\pi,\mu} - \lambda_{\pi',\mu'}\|_{\mathrm{TV}} \leq \left\| \lambda_{\pi,\mu} - \left( \mathsf{P}_\mu^\pi \right)^M \right\|_{\mathrm{TV}} + \left\| \left( \mathsf{P}_\mu^\pi \right)^M - \left( \mathsf{P}_{\mu'}^{\pi'} \right)^M \right\|_{\mathrm{TV}} + \left\| \left( \mathsf{P}_{\mu'}^{\pi'} \right)^M - \lambda_{\pi',\mu'} \right\|_{\mathrm{TV}} .
$$

From Assumption 3 together with Step 1, we obtain

$$
\begin{aligned}
\|\lambda_{\pi,\mu} - \lambda_{\pi',\mu'}\|_{\mathrm{TV}} &\leq C_{\mathrm{Erg}} \rho^M + L_\mathsf{P}\, C_{\mathrm{Erg}} \frac{1 - \rho^M}{1 - \rho} \left( \sum_{s \in \mathcal{S}} \xi(s) \|\pi(\cdot|s) - \pi'(\cdot|s)\|_{\mathrm{TV}} + \|\mu - \mu'\|_2 \right) + C_{\mathrm{Erg}} \rho^M \\
&\leq C_{\mathrm{Erg}} \rho^M + \frac{L_\mathsf{P}\, C_{\mathrm{Erg}}}{1 - \rho} \left( \sum_{s \in \mathcal{S}} \xi(s) \|\pi(\cdot|s) - \pi'(\cdot|s)\|_{\mathrm{TV}} + \|\mu - \mu'\|_2 \right) + C_{\mathrm{Erg}} \rho^M \\
&\leq C_{\mathrm{Erg}} \rho^M + \frac{L_\mathsf{P}\, C_{\mathrm{Erg}}}{1 - \rho} \left( \sup_{s \in \mathcal{S}} \|\pi(\cdot|s) - \pi'(\cdot|s)\|_{\mathrm{TV}} + \|\mu - \mu'\|_2 \right) + C_{\mathrm{Erg}} \rho^M .
\end{aligned}
$$

As this is true for any $M \geq 1$, taking $M$ to infinity, from Fatou's lemma we get (30). $\qquad\square$

### E.2. From bound on Value function to bounds on Policy

In this section, we demonstrate how a bound on the value function naturally leads to a corresponding bound on the policies. In the seminal work by Shani et al. (2020), an $\widetilde{O}(1/N)$ bound was established for the cost functions. This result can be extended to derive a bound on the distance between policies by leveraging the properties of regularization. The connection between the value function and policies highlights the role of regularization in maintaining both theoretical guarantees and practical performance stability.

Indeed, from the entropic regularization, the optimization problem (3) with respect to the profile $\mu$ admits a unique solution $\pi_\mu$. These considerations form the foundation of the following proposition.

**Proposition E.2.** *We have that*

$$
\|\pi(\cdot|s_0) - \pi_\mu(\cdot|s_0)\|_{\mathrm{TV}}^2 \leq \frac{2}{\eta(1 - \gamma)} \left( J^{\mathrm{MFG}}(\pi_\mu, \mu, s_0) - J^{\mathrm{MFG}}(\pi, \mu, s_0) \right) , \tag{33}
$$

*for any $s_0 \in \mathcal{S}$.*

*Proof.* Denote $\Omega$ the entropic regularization term in the reward function (2) as a function of the occupation measure, *i.e.*,

$$
\Omega \left( \mathsf{d}_{\xi,\mu}^\pi \right) := \sum_{a \in \mathcal{A}, s \in \mathcal{S}} \mathsf{d}_{\xi,\mu}^\pi(s, a) \log(\pi(a|s)) ,
$$

with $\mathsf{d}_{\xi,\mu}^{\pi}$ as in 5. Therefore, we can express $J^{\mathrm{MFG}}$ as

$$J^{\mathrm{MFG}}(\pi,\mu,\xi) = \sum_{a\in\mathcal{A},s\in\mathcal{S}} \mathsf{r}(s,a,\mu)\mathsf{d}_{\xi,\mu}^{\pi}(a,s) + \eta\Omega\left(\mathsf{d}_{\xi,\mu}^{\pi}\right) \ . \tag{34}$$

Taking the disentegration on the spatial component, we have the following relationship between the occupation measure and its marginal

$$\begin{aligned} \mathsf{d}_{\xi,\mu}^{\pi}(s,a) &= \sum_{t=0}^{\infty} \gamma^t \pi(a|s)\mathsf{P}_{\pi}\Big(s_t = s\Big|s_0 \sim \xi,\ s_{t+1}\sim\mathsf{P}\left(\cdot|s_t,a,\mu\right)\Big) \\ &= \pi(a|s)\,\overline{\mathsf{d}}_{\mu,\xi}^{\pi}(s)\ , \end{aligned} \tag{35}$$

with $\overline{\mathsf{d}}_{\mu,\xi}^{\pi}$ as in 6. This also implies that

$$\Omega\left(\mathsf{d}_{\xi,\mu}^{\pi}\right) = \sum_{a\in\mathcal{A},s\in\mathcal{S}} \mathsf{d}_{\xi,\mu}^{\pi}(s,a)\log(\pi(a|s)) = \sum_{a\in\mathcal{A},s\in\mathcal{S}} \pi(a|s)\log(\pi(a|s))\,\overline{\mathsf{d}}_{\mu,\xi}^{\pi}(s)\ . \tag{36}$$

From (34), as the optimal value does not depend on the initial condition, we have that the optimal policy $\pi_{\mu}$ of the previous optimization problem satisfies

$$\nabla_{\pi} J^{\mathrm{MFG}}(\pi_{\mu},\mu,s_0) = 0\ ,$$

for any $s_0 \in \mathcal{S}$. Combining this with (35), we obtain that the previous condition equivalent to

$$\mathsf{r}(s,a,\mu) = -\eta\nabla\Omega\left(\mathsf{d}_{s_0,\mu}^{\pi_{\mu}}\right)(s,a)\ ,$$

for any $a\in\mathcal{A}$, $s_0\in\mathcal{S}$. We recall that the Bregman divergence $\mathbf{D}_{\Omega}$ with respect to the regularization $\Omega$ is defined as follows

$$\mathbf{D}_{\Omega}(\nu\|\nu') = \Omega(\nu) - \Omega(\nu) - \nabla\Omega(\nu')^{\top}(\nu - \nu')\ , \qquad \text{for } \nu,\nu' \in \mathcal{A}\times\mathcal{S}\ .$$

Therefore,

$$\begin{aligned} &J^{\mathrm{MFG}}(\pi,\mu,s_0) - J^{\mathrm{MFG}}(\pi_{\mu},\mu,s_0) \\ &= \sum_{(s,a)\in\mathcal{S}\times\mathcal{A}} \mathsf{r}(s,a,\mu)\left(\mathsf{d}_{s_0,\mu}^{\pi} - \mathsf{d}_{s_0,\mu}^{\pi_{\mu}}\right)(s,a) + \eta\Omega\left(\mathsf{d}_{s_0,\mu}^{\pi}\right) - \eta\Omega\left(\mathsf{d}_{s_0,\mu}^{\pi_{\mu}}\right) \\ &= -\eta\sum_{(s,a)\in\mathcal{S}\times\mathcal{A}} \nabla\Omega\left(\mathsf{d}_{s_0,\mu}^{\pi_{\mu}}\right)(s,a)\left(\mathsf{d}_{s_0,\mu}^{\pi} - \mathsf{d}_{s_0,\mu}^{\pi_{\mu}}\right)(s,a) + \eta\Omega\left(\mathsf{d}_{s_0,\mu}^{\pi}\right) - \eta\Omega\left(\mathsf{d}_{s_0,\mu}^{\pi_{\mu}}\right) \\ &= \eta\cdot\mathbf{D}_{\Omega}\left(\mathsf{d}_{s_0,\mu}^{\pi}\big\|\mathsf{d}_{s_0,\mu}^{\pi_{\mu}}\right)\ . \end{aligned} \tag{37}$$

However, we have that with the Bregman divergence corresponding to the entropy regularization $\Omega$ has the following expression (see, *e.g.*, Neu et al., 2017)

$$\begin{aligned} \mathbf{D}_{\Omega}\left(\mathsf{d}_{s_0,\mu}^{\pi}\big\|\mathsf{d}_{s_0,\mu}^{\pi_{\mu}}\right) &= \sum_{(s,a)\in\mathcal{S}\times\mathcal{A}} \mathsf{d}_{s_0,\mu}^{\pi}(s,a)\log\left(\frac{\pi(a|s)}{\pi_{\mu}(a|s)}\right) \\ &= \sum_{s\in\mathcal{S}}\overline{\mathsf{d}}_{\mu,s_0}^{\pi}(s)\sum_{a\in\mathcal{A}}\pi(a|s)\log\left(\frac{\pi(a|s)}{\pi_{\mu}(a|s)}\right) \\ &= \sum_{s\in\mathcal{S}}\overline{\mathsf{d}}_{\mu,s_0}^{\pi}(s)\mathrm{KL}\left(\pi(a|s)\big\|\pi_{\mu}(a|s)\right)\ . \end{aligned}$$

Moreover, from the definition of $\overline{\mathsf{d}}$, extracting the first term of the series, we obtain

$$\begin{aligned} \overline{\mathsf{d}}_{\mu,s_0}^{\pi}(s) &= (1-\gamma)\sum_{t=0}^{\infty}\gamma^t\mathsf{P}_{\mu}^{\pi}(s_t = s) \\ &= (1-\gamma)\delta_{s_0}(s) + (1-\gamma)\sum_{t=1}^{\infty}\gamma^t\mathsf{P}_{\mu}^{\pi}(s_t = s) \\ &= (1-\gamma)\delta_{s_0}(s) + (1-\gamma)\gamma\sum_{t=0}^{\infty}\gamma^t\mathsf{P}_{\mu}^{\pi}(s_{t+1} = s)\ . \end{aligned}$$

Using the decomposition of $\mathsf{P}_\mu^\pi(s_{t+1} = s)$ as

$$\mathsf{P}_\mu^\pi(s_{t+1} = s) = \sum_{(s',a') \in \mathcal{S} \times \mathcal{A}} \mathsf{P}(s_{t+1} = s | s_t = s', a_t = a', \mu) \mathsf{P}_\mu^\pi(s_t = s', a_t = a')$$

$$= \sum_{(s',a') \in \mathcal{S} \times \mathcal{A}} \mathsf{P}(s | s', a', \mu) \mathsf{P}_\mu^\pi(s_t = s', a_t = a') ,$$

we get

$$\overline{\mathsf{d}}_{\mu,s_0}^\pi(s) = (1 - \gamma)\delta_{s_0}(s) + \gamma \sum_{(s',a') \in \mathcal{S} \times \mathcal{A}} \mathsf{P}(s | s', a', \mu)(1 - \gamma) \sum_{t=0}^\infty \gamma^t \mathsf{P}_\mu^\pi(s_t = s', a_t = a')$$

$$= (1 - \gamma)\delta_{s_0}(s) + \gamma \sum_{(s',a') \in \mathcal{S} \times \mathcal{A}} \mathsf{P}(s | s', a', \mu) \mathsf{d}_{s_0,\mu}^\pi(s', a') .$$

Therefore, for a function $\psi : \mathcal{S} \to [0, \infty)$, we have

$$\sum_{s \in \mathcal{S}} \psi(s) \overline{\mathsf{d}}_{\mu,s_0}^\pi(s) = (1 - \gamma)\psi(s_0) + \gamma \sum_{(s',a') \in \mathcal{S} \times \mathcal{A}} \sum_{s \in \mathcal{S}} \psi(s) \mathsf{P}(s | s', a', \mu) \mathsf{d}_{s_0,\mu}^\pi(s', a')$$

$$\geq (1 - \gamma)\psi(s_0) ,$$

since $\psi$ is a positive function and $\mathsf{d}_{s_0,\mu}^\pi$ is a positive measure. Applying this to the positive function $s \mapsto \mathrm{KL}\big(\pi(a|s)\big\|\pi_\mu(a|s)\big)$, together with Pinsker's inequality (see, *e.g.*, Cover, 1999), we have

$$\mathbf{D}_\Omega\left(\mathsf{d}_{s_0,\mu}^\pi\big\|\mathsf{d}_{s_0,\mu}^{\pi_\mu}\right) \geq (1 - \gamma)\mathrm{KL}\big(\pi(a|s_0)\big\|\pi_\mu(a|s_0)\big)$$

$$\geq \frac{1 - \gamma}{2} \|\pi(a|s_0) - \pi_\mu(a|s_0)\|_{\mathrm{TV}}^2 .$$

Combining this with (37), we get (33). □

**Proposition E.3.** *Suppose Assumption 1 holds. Then, we have for any two $\mu, \mu'$ that*

$$\left|J^{\mathrm{MFG}}(\pi_\mu, \mu, s_0) - J^{\mathrm{MFG}}(\pi_{\mu'}, \mu', s_0)\right| \leq \frac{L_\mathsf{r} + \frac{\gamma}{1-\gamma}L_\mathsf{P}\left(\|\mathsf{r}\|_\infty + \eta \log|\mathcal{A}|\right)}{1 - \gamma} \cdot \|\mu - \mu'\|_2 , \tag{38}$$

*and*

$$\left|J^{\mathrm{MFG}}(\pi, \mu, s_0) - J^{\mathrm{MFG}}(\pi, \mu', s_0)\right| \leq \frac{L_\mathsf{r} + \frac{\gamma}{1-\gamma}L_\mathsf{P}\left(\|\mathsf{r}\|_\infty + \eta \log|\mathcal{A}|\right)}{1 - \gamma} \cdot \|\mu - \mu'\|_2 , \tag{39}$$

*for any $s_0 \in \mathcal{S}$ and any $\pi \in \Pi$.*

*Proof.* **Step 1.** Let us state the optimal Bellman equations

$$Q_\mu^{\pi_\mu}(s, a) = \mathsf{r}(s, a, \mu) + \gamma \sum_{s' \in \mathcal{S}} \mathsf{P}(s'|s, a, \mu) \cdot J^{\mathrm{MFG}}(\pi_\mu, \mu, s') ,$$

$$J^{\mathrm{MFG}}(\pi_\mu, \mu, s) = \eta \log\left(\sum_{a \in \mathcal{A}} \exp\left\{\frac{1}{\eta} Q_\mu^{\pi_\mu}(s, a)\right\}\right) .$$

We notice that a function $x \mapsto \eta \cdot \log\left(\sum_{i=1}^d \exp\{\frac{1}{\eta}x_i\}\right)$ is 1-Lipschitz in $\ell_\infty$-norm since the $\ell_1$-norm of the gradient of this function always lies on a probability simplex (see, *e.g.*, Geist et al., 2019). Thus, we have

$$J^{\mathrm{MFG}}(\pi_\mu, \mu, s_0) - J^{\mathrm{MFG}}(\pi_{\mu'}, \mu', s_0) \leq \max_{a \in \mathcal{A}} \left|Q_\mu^{\pi_\mu}(s_0, \cdot) - Q_{\mu'}^{\pi_{\mu'}}(s_0, \cdot)\right| .$$

Then, we study the Lipschitzness of optimal $Q$-values for arbitrary action $a_0 \in \mathcal{A}$

$$
\begin{aligned}
\left| Q_\mu^{\pi_\mu}(s_0, a_0) - Q_{\mu'}^{\pi_{\mu'}}(s_0, a_0) \right| &\leq \left| \mathsf{r}(s_0, a_0, \mu) - \mathsf{r}(s_0, a_0, \mu') \right| \\
&+ \gamma \left| \sum_{s' \in \mathcal{S}} \left[ \mathsf{P}(s'|s_0, a_0, \mu) - \mathsf{P}(s'|s_0, a_0, \mu') \right] \cdot J^{\mathrm{MFG}}(\pi_\mu, \mu, s') \right| \\
&+ \gamma \left| \sum_{s' \in \mathcal{S}} \mathsf{P}(s'|s_0, a_0, \mu') \cdot \left[ J^{\mathrm{MFG}}(\pi_\mu, \mu, s') - J^{\mathrm{MFG}}(\pi_{\mu'}, \mu', s') \right] \right| .
\end{aligned}
$$

By Assumption 1, we have

$$
\left| \mathsf{r}(s_0, a_0, \mu) - \mathsf{r}(s_0, a_0, \mu') \right| \leq L_\mathsf{r} \left\| \mu - \mu' \right\|_2 , \quad \left\| \mathsf{P}(\cdot|s_0, a_0, \mu) - \mathsf{P}(\cdot|s_0, a_0, \mu') \right\|_{\mathrm{TV}} \leq L_\mathsf{P} \left\| \mu - \mu' \right\|_2 .
$$

thus

$$
\begin{aligned}
\left| Q_\mu^{\pi_\mu}(s_0, a_0) - Q_{\mu'}^{\pi_{\mu'}}(s_0, a_0) \right| &\leq \left( L_\mathsf{r} + \gamma L_\mathsf{P} \left\| J^{\mathrm{MFG}}(\pi_\mu, \mu, \cdot) \right\|_\infty \right) \cdot \left\| \mu - \mu' \right\|_2 \\
&+ \gamma \left\| J^{\mathrm{MFG}}(\pi_\mu, \mu, \cdot) - J^{\mathrm{MFG}}(\pi_{\mu'}, \mu', \cdot) \right\|_\infty .
\end{aligned}
$$

Overall, we have a recursive bound on difference between value functions

$$
\begin{aligned}
\left\| J^{\mathrm{MFG}}(\pi_\mu, \mu, \cdot) - J^{\mathrm{MFG}}(\pi_{\mu'}, \mu', \cdot) \right\|_\infty &\leq \left( L_\mathsf{r} + \gamma L_\mathsf{P} \left\| J^{\mathrm{MFG}}(\pi_\mu, \mu, \cdot) \right\|_\infty \right) \cdot \left\| \mu - \mu' \right\|_2 \\
&+ \gamma \left\| J^{\mathrm{MFG}}(\pi_\mu, \mu, \cdot) - J^{\mathrm{MFG}}(\pi_{\mu'}, \mu', \cdot) \right\|_\infty ,
\end{aligned}
$$

therefore

$$
\left\| J^{\mathrm{MFG}}(\pi_\mu, \mu, \cdot) - J^{\mathrm{MFG}}(\pi_{\mu'}, \mu', \cdot) \right\|_\infty \leq \frac{L_\mathsf{r} + \gamma L_\mathsf{P} \left\| J^{\mathrm{MFG}}(\pi_\mu, \mu, \cdot) \right\|_\infty}{1 - \gamma} \left\| \mu - \mu' \right\|_2 .
$$

By a bound $\left\| J^{\mathrm{MFG}}(\pi_\mu, \mu, \cdot) \right\|_\infty \leq (\|\mathsf{r}\|_\infty + \eta \log |\mathcal{A}|)/(1 - \gamma)$, we conclude the statement (38).

**Step 2.** Applying directly the Bellman equation, we have

$$
\begin{aligned}
&\left| J^{\mathrm{MFG}}(\pi, \mu, s_0) - J^{\mathrm{MFG}}(\pi, \mu', s_0) \right| \\
&\leq \sum_{a_0 \in \mathcal{A}} \pi(a_0|s_0) \left| \mathsf{r}(s_0, a_0, \mu) - \mathsf{r}(s_0, a_0, \mu') \right| \\
&+ \gamma \sum_{a_0 \in \mathcal{A}} \pi(a_0|s_0) \left| \sum_{s' \in \mathcal{S}} \cdot \left[ \mathsf{P}(s'|s_0, a_0, \mu) - \mathsf{P}(s'|s_0, a_0, \mu') \right] \cdot J^{\mathrm{MFG}}(\pi_\mu, \mu, s') \right| \\
&+ \gamma \sum_{a_0 \in \mathcal{A}} \pi(a_0|s_0) \left| \sum_{s' \in \mathcal{S}} \mathsf{P}(s'|s_0, a_0, \mu') \cdot \left[ J^{\mathrm{MFG}}(\pi_\mu, \mu, s') - J^{\mathrm{MFG}}(\pi_{\mu'}, \mu', s') \right] \right| .
\end{aligned}
$$

Following the same lines as in the Step 1, we can then obtain (39).

$\square$

**Corollary E.4.** *Suppose Assumption 1 holds. Then, we have for any two $\mu, \mu'$ that*

$$
\sup_{s \in \mathcal{S}} \left\| \pi_\mu(\cdot|s) - \pi_{\mu'}(\cdot|s) \right\|_{\mathrm{TV}}^2 \leq C_{\pi,\mu} \left\| \mu - \mu' \right\|_2 , \tag{40}
$$

*with*

$$
C_{\pi,\mu} := \frac{4}{\eta(1 - \gamma)} \cdot \frac{L_\mathsf{r} + \frac{\gamma}{1-\gamma} L_\mathsf{P} \left( \|\mathsf{r}\|_\infty + \eta \log |\mathcal{A}| \right)}{1 - \gamma} .
$$

*Proof.* From E.2, we have that

$$\|\pi(\cdot|s_0) - \pi_\mu(\cdot|s_0)\|_{\mathrm{TV}}^2$$
$$\leq \frac{2}{\eta(1-\gamma)} \left( J^{\mathrm{MFG}}(\pi_\mu, \mu, s_0) - J^{\mathrm{MFG}}(\pi_{\mu'}, \mu, s_0) \right)$$
$$\leq \frac{2}{\eta(1-\gamma)} \left( J^{\mathrm{MFG}}(\pi_\mu, \mu, s_0) - J^{\mathrm{MFG}}(\pi_{\mu'}, \mu', s_0) + J^{\mathrm{MFG}}(\pi_{\mu'}, \mu', s_0) - J^{\mathrm{MFG}}(\pi_{\mu'}, \mu, s_0) \right) .$$

Then, applying twice Proposition E.3, we obtain (40). $\qquad\square$

### E.3. Bound on the Exploitability

To analyze the exploitability $\phi$ of a given policy $\pi$ and a given mean-field parameter $\mu$, we decompose it into two key contributions. The first term captures the suboptimality of the best response against the mean-field distribution, quantifying how much an agent can improve its reward by deviating optimally. The second term accounts for the discrepancy between the current population distribution and the stationary distribution of the Markov reward process induced by $(\pi, \mu)$. This decomposition allows us to explicitly bound the exploitability by controlling both the policy's optimality and the convergence of the population dynamics to equilibrium.

**Proposition E.5.** *Fix a policy $\pi \in \Pi$ and two mean-field parameter $\mu \in \mathcal{P}(\mathcal{S})$. Then, we have that the exploitability $\phi$ as defined in (7) is bounded by*

$$\phi(\pi, \mu) \leq \left( \max_{\pi'} J^{\mathrm{MFG}}(\pi', \mu, \mu) - J^{\mathrm{MFG}}(\pi, \mu, \mu) \right) + C_\phi \|\lambda_{\pi,\mu} - \mu\|_2 ,$$

*with*

$$C_\phi := 2 \frac{L_{\mathsf{r}} + \frac{\gamma}{1-\gamma} L_{\mathsf{P}} \left( \|\mathsf{r}\|_\infty + \eta \log |\mathcal{A}| \right)}{1-\gamma} + 2\sqrt{|\mathcal{S}|} \cdot \frac{\|\mathsf{r}\|_\infty + \eta \log(|\mathcal{A}|)}{1-\gamma} . \tag{41}$$

*Proof.* Fix a policy $\pi \in \Pi$ and two mean-field parameter $\mu, \mu' \in \mathcal{P}(\mathcal{S})$. Then, we have that

$$J^{\mathrm{MFG}}(\pi, \mu, \mu) = \left( J^{\mathrm{MFG}}(\pi, \mu, \mu) - J^{\mathrm{MFG}}(\pi, \mu', \mu) \right) + \left( J^{\mathrm{MFG}}(\pi, \mu', \mu) - J^{\mathrm{MFG}}(\pi, \mu', \mu') \right) + J^{\mathrm{MFG}}(\pi, \mu', \mu') .$$

On the one hand, from Proposition E.3, we have that

$$\left| J^{\mathrm{MFG}}(\pi, \mu, \mu) - J^{\mathrm{MFG}}(\pi, \mu', \mu) \right| \leq \frac{L_{\mathsf{r}} + \frac{\gamma}{1-\gamma} L_{\mathsf{P}} \left( \|\mathsf{r}\|_\infty + \eta \log |\mathcal{A}| \right)}{1-\gamma} \cdot \|\mu - \mu'\|_2 .$$

On the other hand, we have that

$$J^{\mathrm{MFG}}(\pi, \mu', \mu) = \sum_{s \in \mathcal{S}} J^{\mathrm{MFG}}(\pi, \mu', s_0) \mu(s_0) ,$$

and a similar decomposition applies for $J^{\mathrm{MFG}}(\pi, \mu', \mu')$. This means that

$$\left| J^{\mathrm{MFG}}(\pi, \mu', \mu) - J^{\mathrm{MFG}}(\pi, \mu', \mu') \right| \leq \sum_{s \in \mathcal{S}} \left| J^{\mathrm{MFG}}(\pi, \mu', s_0) \right| |\mu'(s) - \mu(s)|$$
$$\leq \frac{\|\mathsf{r}\|_\infty + \eta \log(|\mathcal{A}|)}{1-\gamma} \sum_{s \in \mathcal{S}} |\mu'(s) - \mu(s)|$$
$$\leq \sqrt{|\mathcal{S}|} \cdot \frac{\|\mathsf{r}\|_\infty + \eta \log(|\mathcal{A}|)}{1-\gamma} \|\mu' - \mu\|_2 ,$$

where we have applied Cauchy-Schwarz inequality in the last bound. Therefore, we can bound the exploitability $\phi$ as defined in (7) as

$$\phi(\pi, \mu) = \max_{\pi' \in \Pi} J(\pi', \lambda_{\pi,\mu}, \lambda_{\pi,\mu}) - J(\pi, \lambda_{\pi,\mu}, \lambda_{\pi,\mu})$$
$$\leq \max_{\pi' \in \Pi} J(\pi', \mu, \mu) - J(\pi, \mu, \mu) + C_\phi \|\lambda_{\pi,\mu} - \mu\|_2$$

$\qquad\square$

### E.4. Discussion on the monotonicity of the optimal Markov Kernel

Define the operator $\mathcal{P}_M$ as $\mathcal{P}_M(\mu) := \mu \left( \mathsf{P}_\mu^{\pi_\mu} \right)^M$. In this section, we outline sufficient conditions under which this operator, responsible for updating the population distribution in the MFG framework, exhibits monotonicity. Monotonicity of $\mathcal{P}_M$ is a crucial property that ensures stability and convergence of the iterative updates toward the Nash equilibrium.

This operator represents a generalization of the standard contractivity condition, which is traditionally formulated with $M = 1$. This generalization is motivated by the fact that, as we aim at studying the regularized ergodic MFG problem (2)-(3), the condition can hold for some $M > 1$.

This contractivity condition reflects the combined effect of the Lipschitz continuity of the regularized best response $\pi_\mu$ and the ergodicity of the Markov reward process $\mathsf{P}_\mu^{\pi_\mu}$, ensuring the stability and convergence of the mean-field population updates in the ergodic setting.

**Lemma E.6** (Strong monotonicity of $\mathcal{P}_M$). *Suppose that Assumptions 1 and 3 hold. We have for all distribution measures $\mu$ and $\mu'$*

$$\langle \mathcal{P}_M(\mu') - \mathcal{P}_M(\mu), \mu' - \mu \rangle \leq C_{\text{op,MFG}} \left\| \mu' - \mu \right\|_2^2 \ ,$$

*with*

$$C_{\text{op,MFG}} = C_{\text{Erg}} L_{\mathsf{P}} \frac{1 - \rho^M}{1 - \rho} \left( 1 + C_{\pi,\mu} \right) + 2 \left| \mathcal{S} \right| C_{\text{Erg}} \rho^M \ .$$

*Proof.* Consider the following decomposition

$$\langle \mu' - \mu, \mathcal{P}_M(\mu') - \mathcal{P}_M(\mu) \rangle = \left\langle \mu' - \mu, \mu' \left[ \left( \mathsf{P}_{\mu'}^{\pi_{\mu'}} \right)^M - \left( \mathsf{P}_\mu^{\pi_\mu} \right)^M \right] \right\rangle + \left\langle \mu' - \mu, (\mu' - \mu) \left( \mathsf{P}_\mu^{\pi_\mu} \right)^M \right\rangle \ .$$

On one hand, applying Lemma E.1 and Cauchy-Schwartz inequality, we obtain

$$\left\langle \mu' - \mu, \mu' \left[ \left( \mathsf{P}_{\mu'}^{\pi_{\mu'}} \right)^M - \left( \mathsf{P}_\mu^{\pi_\mu} \right)^M \right] \right\rangle$$

$$\leq \left\| \mu' - \mu \right\|_2 \cdot \left\| \left( \mathsf{P}_{\mu'}^{\pi_{\mu'}} \right)^M - \left( \mathsf{P}_\mu^{\pi_\mu} \right)^M \right\|_{\text{TV}}$$

$$\leq \left\| \mu' - \mu \right\|_2 \cdot C_{\text{Erg}} L_{\mathsf{P}} \frac{1 - \rho^M}{1 - \rho} \left( \sup_{s \in \mathcal{S}} \left\| \pi_\mu(\cdot|s) - \pi_{\mu'}(\cdot|s) \right\|_{\text{TV}} + \left\| \mu - \mu' \right\|_2 \right)$$

$$\leq C_{\text{Erg}} L_{\mathsf{P}} \frac{1 - \rho^M}{1 - \rho} \left( 1 + C_{\pi,\mu} \right) \left\| \mu - \mu' \right\|_2^2 \ ,$$

where in the last inequality we have applied Corollary (E.4).

On the other hand, from Assumption 3, we get that

$$\left\langle \mu' - \mu, (\mu' - \mu) \left( \mathsf{P}_\mu^{\pi_\mu} \right)^M \right\rangle \leq \left\| \mu' - \mu \right\|_1 \left\| (\mu' - \mu) \left( \mathsf{P}_\mu^{\pi_\mu} \right)^M \right\|_{\text{TV}}$$

$$\leq \left\| \mu' - \mu \right\|_1 \left( \left\| \mu \left( \mathsf{P}_\mu^{\pi_\mu} \right)^M - \lambda_{\pi_\mu,\mu} \right\|_{\text{TV}} + \left\| \lambda_{\pi_\mu,\mu} - \mu' \left( \mathsf{P}_\mu^{\pi_\mu} \right)^M \right\|_{\text{TV}} \right)$$

$$\leq \left\| \mu' - \mu \right\|_1 \cdot 2 C_{\text{Erg}} \rho^M$$

$$\leq 2 \left| \mathcal{S} \right| C_{\text{Erg}} \rho^M \left\| \mu' - \mu \right\|_2^2 \ .$$

$\square$

The condition $C_{\text{op,MFG}} < 1$ is satisfied when the Lipschitz constant $L_{\mathsf{P}}$ associated with the transition kernel is sufficiently small, and the exponent $M$ is large enough.

Intuitively, a smaller $L_P$ indicates that the transition dynamics of the MDP are less sensitive to changes in the population distribution, reducing the potential for instability. A regularity condition on $L_P$ is a standard assumption in the literature to ensure the uniqueness of the MFNE. Similar assumptions have been employed in various works, including Becherer & Hesse (2024), Espinosa & Touzi (2015), Lacker & Zariphopoulou (2019), and Tangpi & Zhou (2024), among many others. These studies leverage regularity constraints to prevent degeneracies in equilibrium selection and to guarantee well-posedness in the associated fixed-point problems.

Meanwhile, a larger $M$ amplifies the effect of the contraction over multiple iterations of the operator, ensuring convergence even in cases where individual updates are not strongly contractive. This interplay between $L_P$ and $M$ highlights the importance of balancing the model's inherent dynamics with the structural assumptions to guarantee monotonicity and stability in the population updates.

## F. Additional Experiments

We present results for the `Exact MF-TRPO` algorithm on two extensions of the Crowd Modeling game and we benchmark our results against Ficticious Play (FP) (Perrin et al., 2020) and Online Mirror Descent (OMD) (Pérolat et al., 2022). Our findings demonstrate that the exact algorithm matches the performance of state-of-the-art methods, highlighting its effectiveness in these settings. In the following, we provide a detailed overview of the games employed.

**Grid-based Crowd Modeling Game.** This environment, inspired by the Four Rooms example from Geist et al. (2022), is based on a two-dimensional grid with obstacles. Each agent's state is defined by her position on the grid, and she can choose from five possible actions: moving left, right, up, down, or staying in place. The reward function is designed to discourage overcrowding by penalizing agents based on the population density at their next position. Specifically, agents receive a negative reward proportional to the logarithm of the density at their destination, encouraging a more even distribution across the state space. Additionally, a small bonus is given for staying in place, while moving in any direction results in a penalty. Formally, the reward function is defined as

$$\mathsf{r}\left(s, a, \mu\right) = -\kappa \log(\mu(s)) + \Gamma(a) \ ,$$

where $\Gamma(a) = 0.2 \cdot \mathbb{1}_{\{a=\text{Stay}\}} - 0.2 \cdot \mathbb{1}_{\{a \neq \text{Stay}\}}$, with $\mathbb{1}$ being the indicator function and $\kappa$ being a crowd-aversion parameter.

In this environment, the transition matrix does not depend on the mean-field distribution $\mu$; however, some stochasticity is introduced through a slipperiness parameter: when an agent selects an action, she is most likely to follow it, but there remains a smaller probability of performing a different valid move. In particular, for each action, a total slipperiness probability of $0.1$ is evenly distributed among the alternative actions. Furthermore, this game can be extended by introducing a designated point of interest, denoted as $s_{\text{target}}$, which guides the behavior of the players. The modified reward function is defined as

$$\tilde{\mathsf{r}}(s, a, \mu) = \mathsf{r}(s, a, \mu) + \max\left(0.3 - 0.1 \cdot d(s, s_{\text{target}}), 0\right) \ ,$$

where $\mathsf{r}(s, a, \mu)$ denotes the previously defined reward function, and $d(s, s_{\text{target}})$ is the distance between state $s$ and the target state $s_{\text{target}}$, computed as the $\ell_1$ norm of their coordinate difference.

**Two-Islands-Graph Crowd Modeling.** The Two Islands Crowd Modeling Game replaces the grid structure with two interconnected graphs, referred to as *islands*, connected by a single narrow bridge. The main challenge in this setting arises

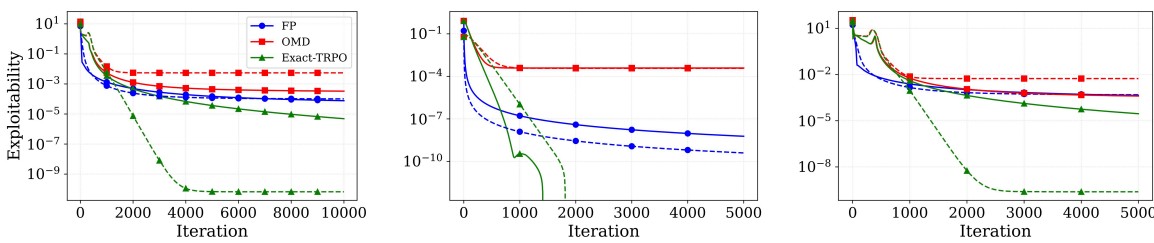

Figure 3. The reading order is (from left to right): Four Rooms Crowd Modeling, Two-Islands-Graph Crowd Modeling, and Four Rooms Crowd Modeling with a point of interest. Solid lines denote $\eta = 0.05$, whereas dashed lines indicate $\eta = 0.3$.

*Table 1.* Parameter settings for the algorithms.

| Algorithm/Parameter | $\kappa$ | $\gamma$ | $\eta$ | $L$ | $\beta$ | $I_\ell$ | $P$ | $M$ |
|---|---|---|---|---|---|---|---|---|
| Exact MF-TRPO | $\{0.2, 0.4\}$ | 0.9 | $\{0.05, 0.3\}$ | 10 | 0.01 | *N/A* | *N/A* | *N/A* |
| Sample-Based MF-TRPO | 0.2 | 0.9 | $\{0.05, 0.3\}$ | 100 | 0.1 | $3 \cdot 10^5$ | $3 \cdot 10^5$ | 100 |

from the limited connectivity between the two sub-populations. The transition matrix is generated randomly, assigning to each node a probability distribution over its neighboring nodes, including itself. The reward function penalizes the logarithm of the mean-field distribution while encouraging movement toward the second island $i_2$,

$$\mathsf{r}(s, \mu) = -\kappa \log(\mu(s))(2 \cdot \mathbb{1}_{s \in i_2} + \mathbb{1}_{s \in i_1}).$$

The `Exact MF-TRPO` algorithm is evaluated on the two proposed variants of the Grid-Based environment and on the Two-Islands-Graph Crowd Modeling game. The former is modeled as an $11 \times 11$ grid with walls delineating four symmetric and interconnected rooms, as in Geist et al. (2022), with all the players starting clustered in the top-left corner. For the latter, we consider a state space of size $|\mathcal{S}| = 14$ and an action space of size $|\mathcal{A}| = 2$, with a branching factor of 2, that is, each state is connected to exactly two neighbors. Here, initially, all players are positioned at location 2 on the first island $i_1$ (see Figure 5).

### F.1. Experimental setting

Results are presented for two different values of the regularization parameter: $\eta = 0.05$ and $\eta = 0.3$ and, throughout all experiments, the discount factor is set to $\gamma = 0.9$. A key feature of both the exact and sample-based methods is the use of a *warm start* for the policy in the RL component. Rather than resetting the policy to a uniform distribution over actions at each iteration, it is initialized with the policy learned from the previous iteration. Moreover, the step size used for updating the distribution remains constant throughout the learning phase, *i.e.*, $\beta_k = \beta$. The key parameters for the two algorithms are summarized in Table 1.

### F.2. Results

The plots presented show the exploitability, defined in Equation (7), to evaluate the effectiveness of our approach, along with the evolution of the mean-field distribution over time. Compared to FP and OMD, `Exact MF-TRPO` performs competitively across all evaluated environments, demonstrating superior long-term performance. As training progresses, the model continually improves its policy and ultimately outperforms the other algorithms (see Figure 3). Moreover, players in grid-based games tend to move toward less crowded areas, gradually achieving a more uniform distribution (see Figure 4). Moreover, when a point of interest is introduced, the players manage to cluster around it (see Figure 6). Finally, as shown in Figure 5, the players progressively concentrate on the second island, attracted by the higher reward present in that region.

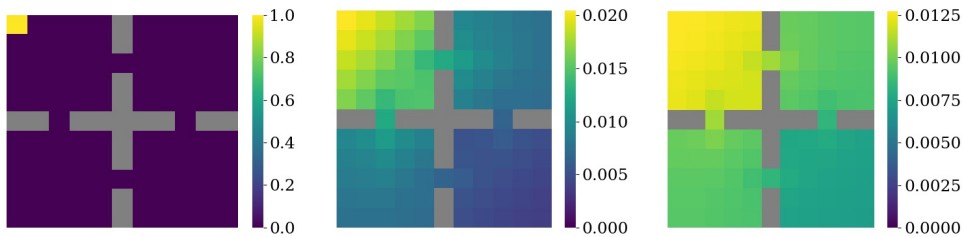

*Figure 4.* Evolution of the mean field distribution for $\eta = 0.05$ in the Four Rooms Crowd Modeling game. From left to right: step 0, step 1000 and step 5000.

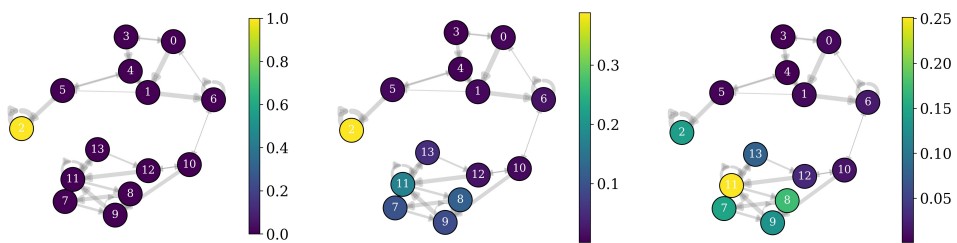

Figure 5. Evolution of the mean field distribution for $\eta = 0.05$ in the Two-Islands Graph Crowd Modeling game. From left to right: step 0, step 2000 and step 5000.

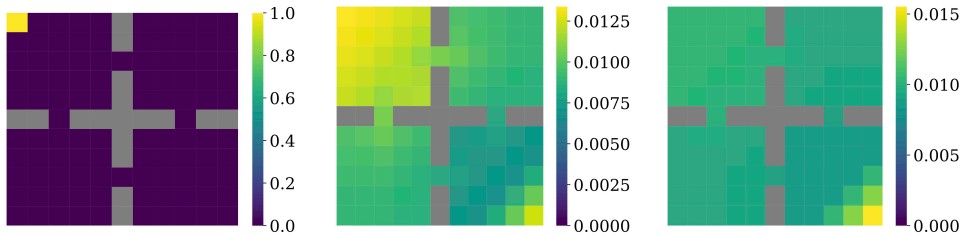

Figure 6. Evolution of the mean field distribution for $\eta = 0.05$ in the Four Rooms Graph Crowd Modeling game with the bottom-right corner being a point of interest. From left to right: step 0, step 1000 and step 5000.

