# OpenReview forum: "Finite-Sample Convergence Bounds for Trust Region Policy Optimization in Mean Field Games"
_ICML.cc/2025/Conference — ICML 2025 poster_

### Official Review · Reviewer_uHrJ · 2025-02-19

**Overall Recommendation:** 2

**Summary:**

This paper extends the known concept of trust region policy optimization (TRPO) from the general reinforcement learning literature to the case of ergodic and entropy regularized mean field games. Besides the exact TRPO, the authors also state a stochastic, sample based version of TRPO and present theoretical results for both the exact and stochastic version.

**Claims And Evidence:**

The claims made by this paper are not very clear. The authors often make rather vague claims such that their learning approach is "a practical and scalable approach for solving mean-field games in large, complex, and data-driven settings" (page 8, lines 409 to 412).  They do not provide any empirical evaluation to back up this claim and the theoretical analysis also appears to be only partially convincing at best.

**Essential References Not Discussed:**

Related works are discussed.

**Experimental Designs Or Analyses:**

There are no experiments.

**Methods And Evaluation Criteria:**

There is no empirical evaluation.

**Other Comments Or Suggestions:**

Comments:
1. The authors should discuss in more detail what a trust region is intuitively and how it entails a better algorithmic performance in mean field games.
2. The authors should explain the intuition behind an occupation measure (page 2).
3. Each of the two Lipschitz constants in Assumption 1 has two different notations. The notations should be unified.
4. Typo page 5, line 243: "showing that show that"
5. What does the notation $\hat{\pi}$ mean in Proposition 5.1 and where is it introduced in the paper?
6. What is the meaning behind the word "informal" in brackets next to Propositions 5.2 and 6.2? Does that mean the statements are not mathematically precise?

**Other Strengths And Weaknesses:**

Strengths: the paper discusses related works and the learning approach might advance the current literature.

Weaknesses:
- no empirical evaluation
- strong and sometimes unclear assumptions
- no precise statement on how the proposed approach is better than existing ones
- the theoretical results do not outline crucial advantages of the proposed learning algorithms

**Questions For Authors:**

The main questions are closely related to the main weaknesses:

1. Why is there no empirical evaluation?
2. What is the practical meaning of Assumptions 3 and 4 and how are they justified?
3. What exactly sets the proposed approach apart from existing learning algorithms?
4. In which way do the theoretical results support the advantages of the MF-TRPO framework compared to existing algorithms?

**Relation To Broader Scientific Literature:**

Although the paper extensively discusses prior works, it remains unclear what the notable contributions of this work are. The authors outline the advantages of entropy regularization for MFGs in their approach multiple times, but these advantages are already well-established in the existing literature. Overall, the authors do not clearly state what sets their work apart from the existing literature. They should precisely pinpoint where the current SOTA learning algorithms fail and why their method provides an improvement in these situations. Additionally, any numerical evaluation and empirical comparison with benchmark methods would be beneficial.

**Theoretical Claims:**

I am not convinced by the theoretical results. First of all, it is not discussed what Assumptions 3 and 4 actually mean in a mean field setting and whether these assumptions are used in any existing mean field game works. For example, how should one check that the two conditions of Assumption 3 are actually fulfilled in an arbitrary given system?

Second, I do not understand the "key insights" of Proposition 5.2 listed in the paper. The mean field distance is bounded by a sum of two terms where only one summand has an exponential character, but the authors claim that there is "exponential convergence". What about the second summand then? Furthermore, I am unsure what the insight "Well-defined Learning rate" means: are there also (many) other models with ill-defined learning rates? If so, the authors should outline their contribution.

---

> ### Author Rebuttal · Authors · 2025-03-31
>
> We thank the reviewer for the thoughtful feedback, which helped us refine the clarity and contextualization of our work. We have revised the manuscript accordingly, including a more detailed comparison with related literature, especially in the ergodic and sample-based MFG setting, to better highlight our contributions.
>
> **Claims:**
>
> By “well-defined learning rate,” we meant rates satisfying the conditions in Propositions B.5 and C.4, which ensure algorithmic convergence. We clarified this phrasing in the revised manuscript. On “exponential convergence,” we now specify that it refers to the convergence of the mean-field parameter (see Proposition B.3), while the second term captures the finite-sample bias in the best-response computation. The revised text reflects this distinction.
>
> Our approach stands out by operating in the $\nu$-restarted RL paradigm, which differs significantly from the generative model assumptions used in prior works such as Angiuli et al. (2021, 2023), Guo et al. (2019), Anahtarci et al. (2023b), Zaman et al. (2023), and Mao et al. (2022). We consider a different paradigm for the ergodic setting. Pérolat et al. (2022) provides no finite-sample guarantees, and Yardim et al. (2023) relies on a strong persistence of excitation assumption, requiring near-uniform policies—something we avoid. By using TRPO instead of OMD, we extend the theoretical analysis to a more stable algorithm. This makes our contribution both practically and theoretically more general than existing approaches.
>
> **Comments:**
>
> 1. **TRPO.** We now provide an expanded discussion explaining that TRPO stabilizes policy updates by constraining their deviation. This is especially valuable in MFGs, where small changes can cause instability due to the interaction through the population parameter. TRPO’s trust region mitigates such effects and ensures more reliable convergence to Nash equilibria.
> 2. **Occupation Measure.** We added a clarification emphasizing its role in the ergodic setting, where it represents the long-run empirical distribution of state-action pairs under a policy. As such, it encodes the joint interaction between policy and mean-field behavior, making it central to our analysis.
> 5. **Notation $\hat{\pi}$.** This was a typo in Proposition 5.1, now corrected for consistency with earlier definitions.
> 6. **``Informal” Propositions.** The label was used to avoid overloading the main text with technical assumptions on learning rates, which are detailed rigorously in the appendix. We’ve added a comment to clarify that the propositions are mathematically precise.
>
> **Questions:**
>
> 1. **Empirical Evaluation.** While our paper was originally theoretical, following the reviewer’s suggestion, we now include a numerical evaluation to demonstrate the practical applicability of our method, see https://anonymous.4open.science/r/TRPO-MFG-0468/rebuttal/experiments.md.
>
> 2. **Assumption 4**, standard in $\nu$-restarted RL (e.g., Kakade & Langford, 2002; Shani et al., 2019), ensures sufficient exploration from the initial distribution. It applies only to optimal mean-field policies, making it milder than uniform assumptions in prior work. **Assumption 3** supports convergence guarantees by ensuring ergodic distributions can be learned, as initial conditions fade and occupancy measures concentrate around their expected values.
>
> Also, we emphasize that these assumptions hold not for any possible policy (that is, uniform concentrability and uniform mixing) but only for the class of optimal mean-field policies. As a result, in our opinion, our assumptions are weak and follow the advances for the single-agent setting. We clarified their roles and practical relevance in the revised manuscript.
>
> 3. **What Sets Our Approach Apart:**
> - We work under a realistic, $\nu$-restarted ergodic MFG paradigm, aligning with practical frameworks like `gymnasium`. This differs from many works that either assume generative models or episodic settings with i.i.d. data.
> - We provide explicit, finite-sample convergence guarantees—still uncommon in the MF-RL literature. We add in the revised version the dependence on key parameters such as the horizon length, population size, and the complexity of the environment.
> - Our assumptions are mild, avoid hard constraints in the softmax temperature for the policies, and are tailored to the practical $\nu$-restarted framework.
> These features make our work distinct, both in theory and potential application, within the current MF-RL landscape.

---

### Official Review · Reviewer_qGy3 · 2025-03-09

**Overall Recommendation:** 3

**Summary:**

The authors formulate algorithms for learning equilibria in MFGs. They provide high-probability guarantees and
finite sample complexity in the ergodic setting, with relaxed assumptions.

## update after rebuttal
I thank the authors for their response and additional experiments. The additional experiments and clarifications sound good to me.

**Claims And Evidence:**

Theoretical evidence has been provided for the developed algorithms. No experiments were performed.

**Essential References Not Discussed:**

N/A

**Experimental Designs Or Analyses:**

N/A

**Methods And Evaluation Criteria:**

The proposed methods make sense, since solving MFGs for NE is a difficult problem, and new sample-based algorithms and analyses are appropriate to move the applicability of MFGs further towards real systems.

**Other Comments Or Suggestions:**

Typo l. 345 "qunatitatives"

**Other Strengths And Weaknesses:**

- A strength is the ergodic setting and sample-based guarantees, as they make contributions more directly applicable in real systems, though assumptions such as $\nu$-restarting and bounds still remain restrictive or hard to verify.
- The writing is very clear, and results are rigorous.
- For me it is a bit unclear how to set all the parameters in practice to run the algorithm, e.g. M.
- I understand the focus is on the theoretical contribution of the paper. Since the contribution is a new algorithm however, and given that this is not the first work on sample based algorithms (Yardim et al.), some experiments would be good to show the usefulness.

**Questions For Authors:**

See above. And in particular, Assumption 2 is discussed to be a relaxed version of existing assumptions, but in fact it seems from Appx. D.3 that the bound C_{op,MFG} will require both Lipschitz conditions as well as conditions on the uniform mixing, the latter of which appears to not always be necessary in existing algorithms, see e.g. (Anahtarci, 2023)?

**Relation To Broader Scientific Literature:**

The explanation of prior work looks good to me, as does the additional exposition in the Appendix.

**Theoretical Claims:**

The proofs look fine and plausible to me, I briefly checked the exact versions of the proofs for results up to page 6.

---

> ### Author Rebuttal · Authors · 2025-03-31
>
> We thank the reviewer for their positive feedback and insightful questions. We address them in order below.
>
> > **A strength is the ergodic setting and sample-based guarantees, as they make contributions more directly applicable in real systems, though assumptions such as $\nu$-restarting and bounds still remain restrictive or hard to verify.**
>
> We agree that the $\nu$-restarting assumption may appear restrictive at first glance; however, it closely mirrors the way environments are typically implemented in practice—for example, in widely-used libraries such as `gymnasium`, where each episode begins by sampling an initial state from a fixed restart distribution. This realism was a key motivation behind adopting this assumption in our mean-field setting, as it allows us to formulate sample-based guarantees applicable to a richer class of examples.
>
> > **For me it is a bit unclear how to set all the parameters in practice to run the algorithm, e.g. M.**
>
> We thank the reviewer for this valuable remark. In the revised version of the manuscript, we have included a more detailed discussion on the role and impact of each parameter—particularly $M$—within the convergence analysis. This was done to better guide the practical implementation of the algorithm. Specifically, we describe how these parameters influence the convergence rate and provide explicit scaling relations that help estimate their appropriate magnitudes. Our analysis begins with the TRPO procedure as a warm start, which offers a structured way to set these parameters based on theoretical guarantees. We hope this clarification helps bridge the gap between theory and practice.
>
> > **I understand the focus is on the theoretical contribution of the paper. Since the contribution is a new algorithm however, and given that this is not the first work on sample based algorithms (Yardim et al.), some experiments would be good to show the usefulness.**
>
> We thank the reviewer for this constructive comment. While our original intention was to focus on the theoretical aspects of the proposed algorithm, we fully acknowledge the importance of empirical validation. Taking this suggestion into account, we have included numerical experiments in the revised manuscript to illustrate the practical applicability and performance of our method, see https://anonymous.4open.science/r/TRPO-MFG-0468/rebuttal/experiments.md.
>
> > **Assumption 2 is discussed to be a relaxed version of existing assumptions, but in fact it seems from Appx. D.3 that the bound C_{op,MFG} will require both Lipschitz conditions as well as conditions on the uniform mixing, the latter of which appears to not always be necessary in existing algorithms, see e.g. (Anahtarci, 2023)?**
>
> We thank the reviewer for this observation. We would like to clarify that while Assumption 2 is framed as a relaxation of classical assumptions, it is indeed necessary for deriving a meaningful and non-asymptotic bound in our ergodic setting. Regarding the comparison with Anahtarci (2023), we point out that Remark 3 in the cited paper explicitly acknowledges the need for a $\beta$-mixing assumption when moving away from the i.i.d. sampling regime, which aligns with our framework. Therefore, our assumptions are directly comparable in scope and purpose, as both approaches require a form of temporal dependence control to ensure valid generalization and convergence guarantees in sample-based reinforcement learning.

---

### Official Review · Reviewer_Dz92 · 2025-03-14

**Overall Recommendation:** 3

**Summary:**

This paper investigates the finite-sample complexity of a TRPO-inspired method for Mean Field Games (MFGs). It is a game with anonymous agents whose number goes to infinity. A mean-field game is formulated through a mean-field MDP that consists of a population distributions over the state spaces and the reward and transition function are dependent on the population density.

A mean-field Nash equilibrium is a population distribution over state that is invariant to transition and also, the policy achieves the maximum value of the MDP for the particular population distribution.

The authors use the TRPO policy update along an update between consecutive population distributions. They begin by analyzing their algorithm for the exact case and show exponential convergence to the optimal population distribution.

Then, the authors demonstrate how their method converges when samples are used to approximate the policy gradients.

**Claims And Evidence:**

The claims are supported by formal mathematical proofs.

**Essential References Not Discussed:**

I do not believe that some key references are missing.

**Experimental Designs Or Analyses:**

N/A

**Methods And Evaluation Criteria:**

No experiments were provided. The theoretical results used mathematical reasoning.

**Other Comments Or Suggestions:**

What does Lipschitz w.r.t. actions mean? Or, where do you define what $|a-a'|$ means?
912 deterinistic -> deterministic
1660 Taking the disentegration between on the spatial component

**Other Strengths And Weaknesses:**

Strengths:
* the design of an algorithm based on TRPO, one of the most widely used in practice algorithms, is remarkable
* the finite-sample complexity result on top of the provable convergence of the latter algorithm is also good.

Weaknesses:
* no experimental evaluation of the algorithm

**Questions For Authors:**

* How does your algorithm compare to other policy optimization techniques?
* Would you expect PPO to work as well?
* Would you be able to incorporate a critic in MF-TRPO?

**Relation To Broader Scientific Literature:**

The paper is related to the broader literature of mean-field games and also policy gradient methods.

**Theoretical Claims:**

I checked extensively the proofs for the exact TRPO and in short the proofs for the stochastic approximation of TRPO.

---

> ### Author Rebuttal · Authors · 2025-03-31
>
> We would like to thank the reviewer for their precise and thoughtful comments, which have greatly contributed to the refinement of our work. While our initial submission was positioned as primarily theoretical, we appreciate the suggestion and have decided to include **numerical experiments** to better illustrate the applicability and practical relevance of our methodology; see https://anonymous.4open.science/r/TRPO-MFG-0468/rebuttal/experiments.md.
>
> > **What does Lipschitz w.r.t. actions mean? Or, where do you define what |a−a′|  means?**
>
> We agree with the reviewer’s comment and have removed the Lipschitz condition with respect to the action variable to avoid any confusion. This assumption was not essential for the main results and its removal clarifies the presentation.
>
> > **How does your algorithm compare to other policy optimization techniques?**
>
> We thank the reviewer for this insightful question. In our work, we adopt a TRPO approach within a $\nu$-restarted episodic setting. This framework is notably different from the generative setting commonly used in conjunction with $Q$-learning techniques, which have been previously explored in the literature.
>
> Our choice of TRPO is motivated by the inherent instability associated with learning a MFNE. We believe that TRPO-based methods, by enforcing stable policy updates through trust region constraints, are better equipped to handle these instabilities. This design choice allows us to align our algorithmic contributions with a theoretically grounded optimization framework while addressing the specific challenges posed by equilibrium learning in large populations.
>
> > **Would you expect PPO to work as well?**
>
> While in practice PPO is known to perform well and could likely be applied in our setting with similar empirical results, the theoretical understanding of PPO remains limited—particularly in the context of convergence guarantees and stability in equilibrium learning. For this reason, we chose TRPO to ground our analysis on a well-understood and principled algorithm, while acknowledging that extending the analysis to PPO remains an interesting direction for future work.
>
> > **Would you be able to incorporate a critic in MF-TRPO?**
>
> In the current theoretical version of MF-TRPO, the critic is implemented via Monte Carlo estimates, which allows for a clean analysis and convergence guarantees. However, in practical implementations, it is indeed possible to replace the Monte Carlo critic with a parametric critic, such as a neural network. This would introduce additional approximation errors and dependencies on function approximation dynamics, which would significantly complicate the theoretical analysis. Extending the theory to account for these effects remains an interesting and challenging direction for future work.
>
> Additionally, we would like to thank the reviewer for their careful reading and for pointing out a number of misprints and minor issues. We have gladly incorporated all the suggested corrections into the revised manuscript.

---

### Official Review · Reviewer_SqLi · 2025-03-16

**Overall Recommendation:** 4

**Summary:**

The paper studies the problem of computing Nash equilibria for discounted tabular Mean Field Games, and proposes a trust region policy optimization framework to learn stable and optimal policies. It provides a theoretical analysis of the framework, showing theoretical guarantees of its convergence.

The paper considers the mean-field entropy-regularized objective, which is well-studied in the literature, and uses a TRPO framework to retain the standard $\tilde{O}(1/L)$ error bounds in the trajectory estimates, but without assumptions of uniformly bounded away trajectories or continuous reward functions.

## update after rebuttal

The authors have addressed all my concerns involving the Lipschitz continuity assumption: I am particularly glad that they can show the validity of the theorems under the (even) weaker assumption of the Lipschitz constant bound of $|r|_\infty$. Moreover, I am happy that the authors have addressed my concerns about assumptions 3 and 4, and I am excited that there is a way for them to incorporate some notion of mean-field sampling, which demonstrates a lot of promise for faster runtimes under the (continued) assumption of homogeneity of the agents.

**Claims And Evidence:**

- The practical significance of the Lipschitz continuity of the reward functions (with respect to $\mu$ and $a$) seems a bit strange in this tabular setting. Lipschitz continuity seems like a more reasonable assumption if the input to the reward function is a point on a manifold, rather than discretized values. In this setting, I would think that the (admittedly, more naïve) $\|r(s, a, \mu) - r(s, a', \mu')\|\_\infty \leq \|r\|_\infty$ is the most reasonable bound, despite the further explanation in the appendix,
- The exact MF-TRPO algorithm requires oracle access to $P$, which is not an assumption stated in the body of the paper,
- In proposition B.3 in the appendix, I wonder if the proposition is missing a further condition on the step size that $\sum_{k=0}^{\infty}\beta_k^2 <\infty$,
- However, besides the points above, I find that the idea of driving high-probability estimates for the estimation error using martingale analysis is appropriately rigorously presented, and the theoretical claims appear to be sufficiently rigorous. For instance, theorem B.1 provides a $\tilde{O}(1/L)$ bound which is exactly what the paper claims.

**Essential References Not Discussed:**

For an audience unfamiliar with mean-field methods, it would be nice if the famous mean-field Q-learning paper by Yang et. al. (2018), which introduces mean-field MARL, were to be discussed as well.

**Experimental Designs Or Analyses:**

There are no experiments.

**Methods And Evaluation Criteria:**

The paper has no experiments.

**Other Comments Or Suggestions:**

- would be good to specify _before equation 2_ that this is the entropy regularized objective function,
- Line 1645 "tem" should be "term",
- How does assumption 3 not follow from ergodicity of the Markov chain?
- Line 243 "showing that" is followed by "show that" which should be removed,
- Line 1296: should be "Jensen's inequality", and it would be good to clarify that this is a Doob martingale

**Other Strengths And Weaknesses:**

- This paper claims to remove some restrictive technical assumptions from previous work. I haven't verified the line-by-line methodology in detail, but it appears to be correct,
- The paper could motivate the additional assumptions a bit further, for instance it relies on a strong Lipschitz condition on the reward function,

**Questions For Authors:**

1) In this mean-field TRPO formulation, can you derive a better sample efficiency through mean-field sampling [1]?
2) Can you extend this formulation to the linear MDP setting beyond the current tabular approach? For instance, the famous mean-field paper by Yang et. al. (2018) provides a Q-learning process in the mean-field setting that works under linear function approximation...

References:
1. Mean-Field Sampling for Cooperative Multi-Agent Reinforcement Learning (Anand et. al. 2025)

**Relation To Broader Scientific Literature:**

This paper contributes to the broader work on TRPO in various mean-field RL settings.

**Theoretical Claims:**

I did not look very carefully at the line-by-line theoretical claims, but the general structure seems reasonable.

---

> ### Author Rebuttal · Authors · 2025-03-31
>
> We would like to thank Reviewer SqLi for the valuable feedback.  Below, we address the issues raised in the review.
>
> > **The practical significance of the Lipschitz continuity of the reward functions (with respect to μ  and a) seems a bit strange in this tabular setting [...] I would think that the (admittedly, more naïve) $ | r ( s, a, \mu ) - r( s, a' ,\mu') | \leq |r|_{ \infty } $**
>
> We respectfully disagree with the Reviewer and would like to point out that the Lipschitz continuity of the reward function with respect to a mean-field measure is critical to establishing convergence rates since, under the proposed condition, any small deviation of the mean-field measure can significantly change the reward function. However, we will remove the continuity condition with respect to actions to simplify the notation.
>
> > **The exact MF-TRPO algorithm requires oracle access to P, which is not an assumption stated in the body of the paper.**
>
> We will explicitly state the assumption for the exact MF-TRPO algorithm in the revised version of the manuscript. Additionally, we would like to point out that the sample-based version of MF-TRPO was designed explicitly to avoid this assumption.
>
> > **In proposition B.3 in the appendix, I wonder if the proposition is missing a further condition on the step size that $\sum_{k=0}^\infty \beta_k^2 < +\infty$.**
>
> No, we do not need this assumption since we are considering the finite-horizon guarantees. The only assumption on $\beta_k$ that we need is stated in (23) (see Theorem C.4 in Appendix) and it requires $\beta_k$ to be only smaller than a constant. Notably, any decreasing step-size condition will eventually start satisfying this bound, but may harm the final convergence bound.
>
> > **The paper has no experiments.**
>
> We considered the paper a theoretical one, but we decided to include numerical studies to stress the applicability of our methods; see https://anonymous.4open.science/r/TRPO-MFG-0468/rebuttal/experiments.md.
>
> > **The paper could motivate the additional assumptions a bit further, for instance it relies on a strong Lipschitz condition on the reward function.**
>
> The assumption of Lipschitz continuity of the reward function in mean-field distribution is standard in the literature, see, e.g. Angiuli et al. (2021, 2023); Guo et al. (2019); Anahtarci et al. (2023a,b); Zaman et al (2023); Yardim et al. (2023). Additionally, we would like to emphasize that in a large part of practical situations, this condition is also satisfied.
>
> > **In this mean-field TRPO formulation, can you derive a better sample efficiency through mean-field sampling [1]?**
>
> We thank the reviewer for this insightful suggestion. The approach proposed in [1] focuses on **mean-field control**, where agents act cooperatively to optimize a shared social welfare objective. In contrast, our setting is that of **mean-field games (MFGs)**, where agents are non-cooperative and aim to selfishly optimize their own value functions in interaction with the aggregate behavior of the population.
> This fundamental distinction complicates a direct application of the approach in [1] to our setting. Specifically, their framework hinges on computing a centralized $Q$-function that captures the collective objective, whereas our formulation requires solving for a fixed point of decentralized best-responses among agents.
> Nonetheless, we agree that exploring the **mean-field control counterpart** of our approach could be a fruitful direction. Adapting our methodology to cooperative settings might yield improved sample efficiency and unify learning procedures for both paradigms. We plan to investigate this avenue in future work.
>
> > **Can you extend this formulation to the linear MDP setting beyond the current tabular approach? For instance, the famous mean-field paper by Yang et. al. (2018) provides a Q-learning process in the mean-field setting that works under linear function approximation.**
>
> We think that such an extension is possible but not straightforward. For example, one needs to guarantee the mean-field distribution's learnability, which might require a new type of assumptions for linear MDPs. However, we find it an exciting direction for further work.
>
>
>
> Additionally, we would like to thank a reviewer for a list of misprints and other small comments. We have happily introduced the required changes in the revised manuscript.

---

> > ### Comment · Reviewer_SqLi · 2025-04-07
> >
> > I thank the authors for their detailed response and apologize for the delay in mine.
> >
> > I would like to discuss this a bit further.
> >
> > - For the practical significance of the paper, I believe that many reward functions _do not_ satisfy the tighter Lipschitz assumption, but _do_ satisfy the looser one, which has Lipschitz constant $|r|_\infty$. This is exacerbated when assuming this continuity WRT to the actions. Can the authors comment on this?
> > - I'm very pleased to see the inclusion of experiments to ground the paper's results in real-world examples.
> > - It would be great if the authors could add a section (or discussion, even in the appendix) on extending the work in this setting to the mean-field (and mean-field sampling) settings. The authors say that mean-field/mean-field sampling may not readily apply as they only hold for cooperative games, but I think it should be possible to readily extend those results to the mean-field sampling setting as well.
> > - After looking at the discussion with reviewer uHrJ, I also think it would be great if the authors could discuss an efficient algorithm to verify assumptions 3 and 4 in the problem setting.
> >
> > I thank the authors again for their previous response and (tentatively) raise my score.

---

> > > ### Author Response · Authors · 2025-04-07
> > >
> > > We thank the reviewer for their thoughtful and constructive comments, which have helped us improve the clarity and scope of the manuscript, as well as for their appreciation of the experimental section.
> > >
> > > >**Lipschitz continuity of the reward function:**
> > >
> > > We agree that assuming a tighter Lipschitz condition can be restrictive in practice. To clarify, we no longer assume continuity with respect to actions, as our setting considers a discrete action and state space. In this discrete case, the weaker assumption — such as a bounded reward function with Lipschitz constant $|r|_\infty$ — is sufficient for our analysis. We will make this distinction clearer in the revised version of the paper.
> > >
> > > >**Mean-field sampling extension:**
> > >
> > > We appreciate the reviewer’s interest in the applicability of our framework to mean-field and mean-field sampling settings. We have added a discussion in the appendix comparing our approach with mean-field sampling and specifically to [1]. As our techniques are agnostic to the specific type of mean-field approximation (including those based on sampling), we believe they can be extended to such settings beyond cooperative games. We agree that formalizing this connection further is a valuable direction for future work.
> > >
> > > >**Verification of Assumptions 3 and 4:**
> > >
> > > We thank the reviewer for highlighting the importance of checking these assumptions in practice. Assumption 4 is satisfied whenever the initial distribution $\nu$ assigns strictly positive probability to all states — a condition that can typically be enforced in implementation. Assumption 3 can be made without loss of generality: if it does not hold globally, our results still apply within each recurrent class separately. Transient states, by definition, do not affect the long-term behavior and thus have no impact on the mean-field mass. We will include an explicit clarification of this in the paper, as well as a brief algorithmic discussion on how to detect recurrent classes in the state space.

---

### Decision · Program_Chairs · 2025-05-01

**Decision:**

Accept (poster)

**Comment:**

This paper develops Trust Region Policy Optimization (TRPO) algorithms for Mean Field Games. It establishes finite-sample theoretical analysis for exact and stochastic versions under relaxed assumptions compared to prior work. While initial reviews raised concerns regarding assumptions and the lack of experiments, the authors' rebuttal effectively addressed major technical points for most reviewers. This leads to a consensus supporting acceptance among the reviewers.